# Combinatorial protein engineering identifies potent CRISPR activators with reduced toxicity

Marla Giddins [1,2,3,19], Alexander F. Kratz[2,4,19], Mark B. De Los Santos [2,5,6], Antoine Forget[7,8], Richa Tiwari [7,8], Gwendolyn Jang [7,8], Tomasz Blazejewski[3,4], Chuyan Qin[9,10], Yiming Huang [3,4], Yeh-Hsing Lao [11,12], Thomas Falconer [2,13], Kam. W. Leong[3,11], Nevan Krogan [7,8,14], Max Staller [15,16,17], Harris Wang[2,3], Lai Wei [2,18,20] ✉ & Alejandro Chavez [2,18,20] ✉

Current protein engineering methods are inadequate to explore the combinatorial potential offered by nature's vast repertoire of protein domains–limiting our ability to create optimal synthetic tools. To overcome this barrier, we develop an approach to create and test thousands of chimeric proteins and employ it to probe an expansive combinatorial landscape of over 15,000 multi-domain CRISPR activators. Our findings indicate that many activators produce substantial cellular toxicity, often unrelated to their capacity to regulate gene expression. We also explore the biochemical features of activation domains and determine how their combinatorial interactions shape activator behavior. Finally, we identify two potent CRISPR activators, MHV and MMH, and demonstrate their enhanced activity across diverse targets and cell types compared to the gold-standard MCP activator, synergistic activation mediator (SAM).

Laboratory-engineered proteins like high-fidelity DNA polymerases, CRISPR base and prime editors, and chimeric antigen receptors have transformed our ability to manipulate and enhance biological systems[1–6]. To craft these tools, researchers combine domains with distinct functions in hopes that the resulting chimeras will retain the properties of their constituent parts[7,8]. Yet, in practice, most protein domains do not behave as independent modules, compromising the effectiveness of predictive design[9,10]. In addition, the low-throughput approaches traditionally employed to create synthetic proteins are insufficient to explore complex combinatorial landscapes and limit our

[1]Department of Microbiology and Immunology, Columbia University Irving Medical Center, New York, NY, USA. [2]Department of Pathology and Cell Biology, Columbia University Irving Medical Center, New York, NY, USA. [3]Department of Systems Biology, Columbia University Irving Medical Center, New York, NY, USA. [4]Integrated Program in Cellular, Molecular, and Biomedical Studies, Columbia University Irving Medical Center, New York, NY, USA. [5]Neurobiology and Behavior Program, Columbia University Irving Medical Center, New York, NY, USA. [6]Department of Genetics and Development, Columbia University Irving Medical Center, New York, NY, USA. [7]Quantitative Biosciences Institute (QBI), University of California, San Francisco, San Francisco, CA, USA. [8]Department of Bioengineering and Therapeutic Sciences, University of California San Francisco, San Francisco, CA, USA. [9]Department of Biochemistry, University of Cambridge, Tennis Court Road, Cambridge, UK. [10]The Gurdon Institute, University of Cambridge, Tennis Court Road, Cambridge, UK. [11]Department of Biomedical Engineering, Columbia University, New York, NY, USA. [12]Department of Pharmaceutical Sciences, University at Buffalo, The State University of New York, Buffalo, NY, USA. [13]Department of Biomedical Informatics, Columbia University Irving Medical Center, New York, NY, USA. [14]Gladstone Institutes, San Francisco, CA, USA. [15]Department of Molecular and Cell Biology, University of California Berkeley, Berkeley, CA, USA. [16]Center for Computational Biology, University of California Berkeley, Berkeley, CA, USA. [17]Chan Zuckerberg Biohub-San Francisco, San Francisco, CA, USA. [18]Department of Pediatrics, University of California San Diego, La Jolla, CA, USA. [19]These authors contributed equally: Marla Giddins, Alexander F. Kratz. [20]These authors jointly supervised this work: Lai Wei & Alejandro Chavez. ✉e-mail: larrywei@health.ucsd.edu; chavez2@health.ucsd.edu

ability to grasp the rules governing the design of chimeric proteins (e.g., which domains work best together, which domain orders are optimal, benefits of fusing multiple copies of the same domain, etc.).

Previous state-of-the-art CRISPR activators, including the tripartite activator VP64-P65-RTA (VPR) and the Synergistic Activation Mediator (SAM), have established the benefit of combining multiple activation domains (ADs) into a single complex for improved transcriptional regulation[11–15]. While VPR and SAM proved successful in a plethora of in vitro and in vivo applications, neither activator exhibits uniform activity across targets and cellular contexts[16–18]. Furthermore, reports that these tools produce toxicity within cellular systems limit their utility[19,20].

Here, we set out to create an improved CRISPR activator and uncover the principles guiding combinatorial AD assembly. As an initial step, we individually test a set of 230 ADs across three target genes, uncovering several potent domains and identifying multiple biochemical features associated with strong activators. To overcome the arduous task of assembling all possible combinations of our strongest ADs one by one, we develop a strategy for cloning and simultaneously barcoding thousands of combinatorial variants *en masse*. Our method tailors the CombiSEAL framework, designed for creating mutants of a single protein, to complex multi-domain protein assemblies and implements unique molecular identifiers (UMIs) to improve data quality[21–24]. We employ this system to test 625 two-part (bipartite) and 15,625 three-part (tripartite) CRISPR activators and leverage our findings to elucidate general principles of activator biology and function within complex arrays. We further characterize cellular toxicity as a prominent feature among activators–which, while often overlooked, can profoundly impact a tool's utility. Finally, we identify two potent CRISPR activators, MHV and MMH, and demonstrate their improved performance across multiple targets and cellular contexts over the gold-standard MS2 coat protein (MCP) activator, SAM[12].

## Results

### Identification of potent activation domains for combinatorial assembly

We set out to use high-throughput protein engineering to uncover the rules governing combinatorial AD assembly and to develop improved CRISPR activators. As an initial step, we compiled a series of 230 ADs spanning 16 to 564 amino acids intended to serve as parts for downstream combinatorial libraries. Our set consisted of full-length proteins or protein domains associated with gene activation, primarily sourced from humans and viruses (Supplementary Fig. 1). It also featured a selection of homologs of popular ADs (e.g., VP16) expected to contain a high proportion of strong activators (Supplementary Data 1, see "Methods"). Each member within this list was C-terminally fused to endonuclease-null Cas9 (dCas9) downstream of a flexible glycine-serine linker and individually transfected into human embryonic kidney 293 T (HEK293T) cells[25]. To gauge activator generalizability across targets, we tested all fusions against two endogenous genes with different baseline expression levels, *EPCAM* and *CXCR4*, as well as a plasmid-borne synthetic reporter (Fig. 1a, and Supplementary Data 2).

While some activators exhibited a broad ability to modulate all three target genes, we observed a wide range in activation magnitudes across targets. Endogenous genes *EPCAM* and *CXCR4* yielded lower levels of induction, up to 13- and 19-fold, respectively than the synthetic reporter, which achieved up to 62-fold induction (Fig. 1b–d, Supplementary Fig. 2a–d). Of all fusions, 47, 24, and 48 produced at least two-fold activation on *EPCAM*, *CXCR4*, and the synthetic reporter, respectively. The higher resistance to activation exhibited by *CXCR4* compared to *EPCAM* might derive from its increased baseline expression in HEK293T cells. This finding aligns with prior studies in which CRISPR activators proved more effective at activating lowly expressed genes than highly expressed ones[11,12,16]. We observed the highest levels

of activity against the plasmid-borne synthetic reporter, whose absence of chromatin marks upon entering the cell and negligible baseline expression likely minimize its resistance to synthetic modulation. Overall, activators showed greater concordance between *EPCAM* and *CXCR4* than between either endogenous target and the synthetic reporter (Supplementary Fig. 3a). Given that researchers have historically relied on synthetic targets for modulator discovery, these results may explain the challenges frequently encountered upon applying previously developed CRISPR activators to endogenous genes[11,17,26–31].

Of all tested activators, the VP16 family of ADs derived from multiple herpesviruses proved most potent and target-generalizable (Fig. 1e, and Supplementary Figs. 2b–d and 3a). Interestingly, the classically used VP16 homolog from human herpes virus type 1 (HHV1_VP16) did not produce the highest overall levels of activation within this family. Rather, VP16 variants derived from other mammalian herpesviruses, such as cercopithecine alphaherpesvirus type 2 (CercAHV2_VP16), leporid alphaherpesvirus type 4 (LAHV4_VP16), and fruit bat alphaherpesvirus type 1 (FBAHV1_VP16) performed best across the majority of targets. Outside of VP16 variants, two additional constructs, termed generalists, ranked among the top 10 performers on all three genes (Fig. 1b–d and f). The first, HS_MLLx3, is comprised of three copies of a fragment from the human proto-oncogene MLL, which binds to the essential coactivator CBP[32]. CBP regulates the expression of hundreds of target genes via its histone acetyltransferase activity, likely driving the cross-target plasticity exhibited by HS_MLLx3 in our assays[33]. In line with this finding, several ADs that bind to a paralogous acetyltransferase, p300 (HS_FAM22F, HS_C3ORF62, HAVF41_E1A, etc.), also produced strong activation across targets[34]. The second non-VP16 generalist AD, HHV8_VIRF2, is derived from the human herpesvirus 8 protein, VIRF2, and is responsible for modulating the host immune response[35]. Despite originating from the same family of viruses that harbors the classically used VP16 activator (*Herpesviridae*), HHV8_VIRF2 exhibits no significant sequence or predicted structural homology to the herpes VP16 AD, suggesting that these domains deploy distinct mechanisms to modulate transcription.

The remaining top-performing non-VP16 domains ranked among the top 10 performers for either one or two target genes (Fig. 1b–d and f). Viral ADs, particularly BEL1 and ICP4 variants, dominated on *CXCR4*, potentially reflecting the known capacity of a number of human-infecting viruses, including the *Herpesviridae* family from which ICP4 is derived, to modulate *CXCR4* activity (Fig. 1c, f)[36]. An assortment of human- and viral-derived domains showed strong activity on *EPCAM* and the plasmid-borne synthetic reporter, with no single class of domains showing preferential enrichment (Figs. 1b, d, and f).

We next explored whether homologs of the same AD display conserved or variable behavior. Overall, homologs showed consistent activation profiles (Supplementary Fig. 3b). For example, BEL1 variants yielded similar activation across targets, and E1A homologs exhibited a bias towards activating *EPCAM* over *CXCR4* and the synthetic reporter. However, our probing of multiple homologs elucidated variants with unique properties or generally increased potency. These results align with recent studies, such as those surveying KRAB domains to develop stronger dCas9 repressors, and highlight the benefits of evaluating multiple homologs of the same AD when looking to maximize tool potential[27].

### Examination of the properties underlying AD potency

Previous works have established the association of a number of biochemical traits, such as acidity and hydrophobicity, with activator function (Fig. 2a)[37–43]. However, these associations have not been widely explored in the context of dCas9 fusions. Examining the amino acid composition of all tested activators, we found that hits, ADs that produced two-fold or greater activation on at least one target, were composed of more negatively charged residues than misses (Fig. 2b,

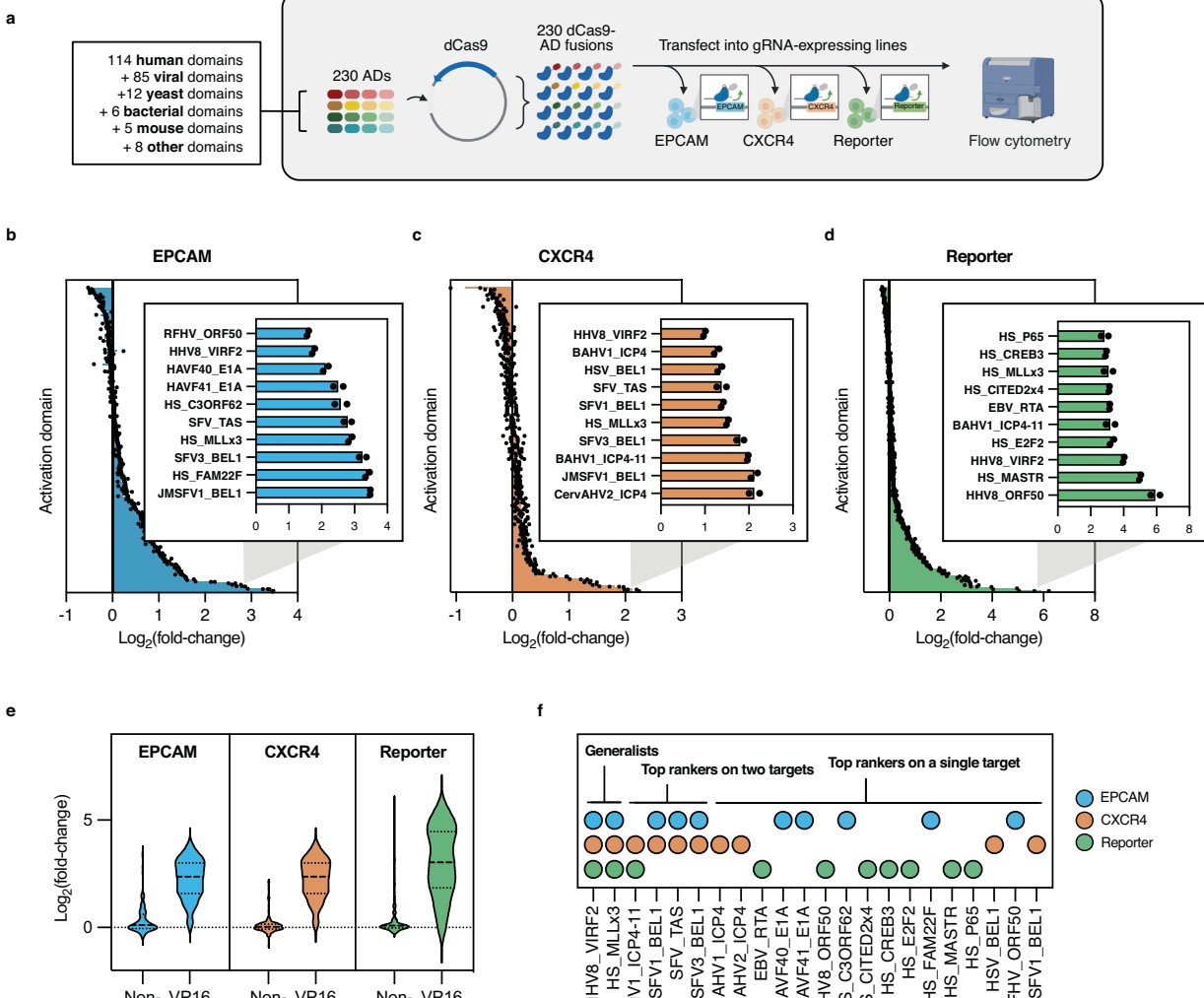

**Fig. 1 | Activator survey and cross-target performance. a** Schematic depicting experimental setup for activation domain testing. 230 activation domains (ADs) were cloned downstream of dCas9 and transfected into HEK293T cells expressing MS2-hairpin-containing gRNAs targeting *EPCAM*, *CXCR4*, or a synthetic reporter plasmid. Flow cytometry was used to measure target activation. *dCas9* endonuclease-null Cas9; *AD* activation domain; *gRNA* guide RNA (*Created in BioRender. Giddins, M.* (https://BioRender.com/ok3u8lb). **b**–**d** Log$_2$(fold-change) in *EPCAM*, *CXCR4*, and fluorescent reporter expression, respectively following targeting by 230 dCas9 fusions. Inset shows top 10 activators. VP16 variants are excluded from the plots for clarity. Black dots depict the results of two independent transfections. **e** Distribution of log$_2$(fold-change) in *EPCAM*, *CXCR4*, and

fluorescent reporter expression for non-VP16-derived ADs and VP16 variant ADs. Data represent distributions of mean values from two independent transfections ($n = 2$ biological replicates). **f** ADs were ranked by activity on each target, with those in the top 10 performers across all targets (generalist), two targets (top rankers on two targets), or one target (top rankers on a single target) shown. Blue, orange, and green circles designate performance on *EPCAM*, *CXCR4*, and the synthetic reporter, respectively. Rankings are based on mean fold-change values across two independent transfections ($n = 2$ biological replicates). For (**b**–**e**) fold-change was calculated as the ratio of median fluorescence intensity (MFI) of cells transfected with each dCas9-AD fusion to the MFI of cells transfected with dCas9 alone (negative control). Source data are provided as a Source Data file (Source Data.zip).

$p < 0.0001$). This finding underscores the importance of acidity for activation and demonstrates that this principle generalizes to dCas9 scaffolds. We did not observe significant differences in the amount of hydrophobicity or the fraction of disorder-promoting residues between hits and misses, challenging the commonly held notion that these properties are essential for AD function (Fig. 2c, d). Interestingly, Omega, a metric that quantifies the degree of mixing between four hydrophobic residues (W, F, Y, and L) and acidic residues (D and E), revealed greater acidic-hydrophobic mixing in hits than misses (Fig. 2e, $p < 0.05$)[44]. These data suggest that, while hydrophobicity alone does not drive activation, the mixing of acidic and hydrophobic residues plays a critical role in AD efficacy. Finally, Kappa, a metric reflecting the degree of mixing between positively and

negatively charged amino acids, showed no differences between hits and misses (Supplementary Fig. 4a)[44]. As a whole, our results both highlight the importance of acidity for activator function and support the Acidic Exposure Model, in which potent activators require an intermixing of acidic residues with hydrophobic residues to ensure hydrophobic motifs remain exposed to coactivators[45].

To determine whether a deep learning model could have predicted the activity of all 230 dCas9-AD fusions, we turned to a state-of-the-art convolutional neural network, PADDLE, capable of predicting strong or medium strength ADs from protein sequence[31]. While PADDLE showed a high sensitivity for identifying hits using either its strong or medium scoring threshold (Supplementary Data 3; sensitivity = 0.66 and 0.88, respectively), it misclassified a high proportion of inactive

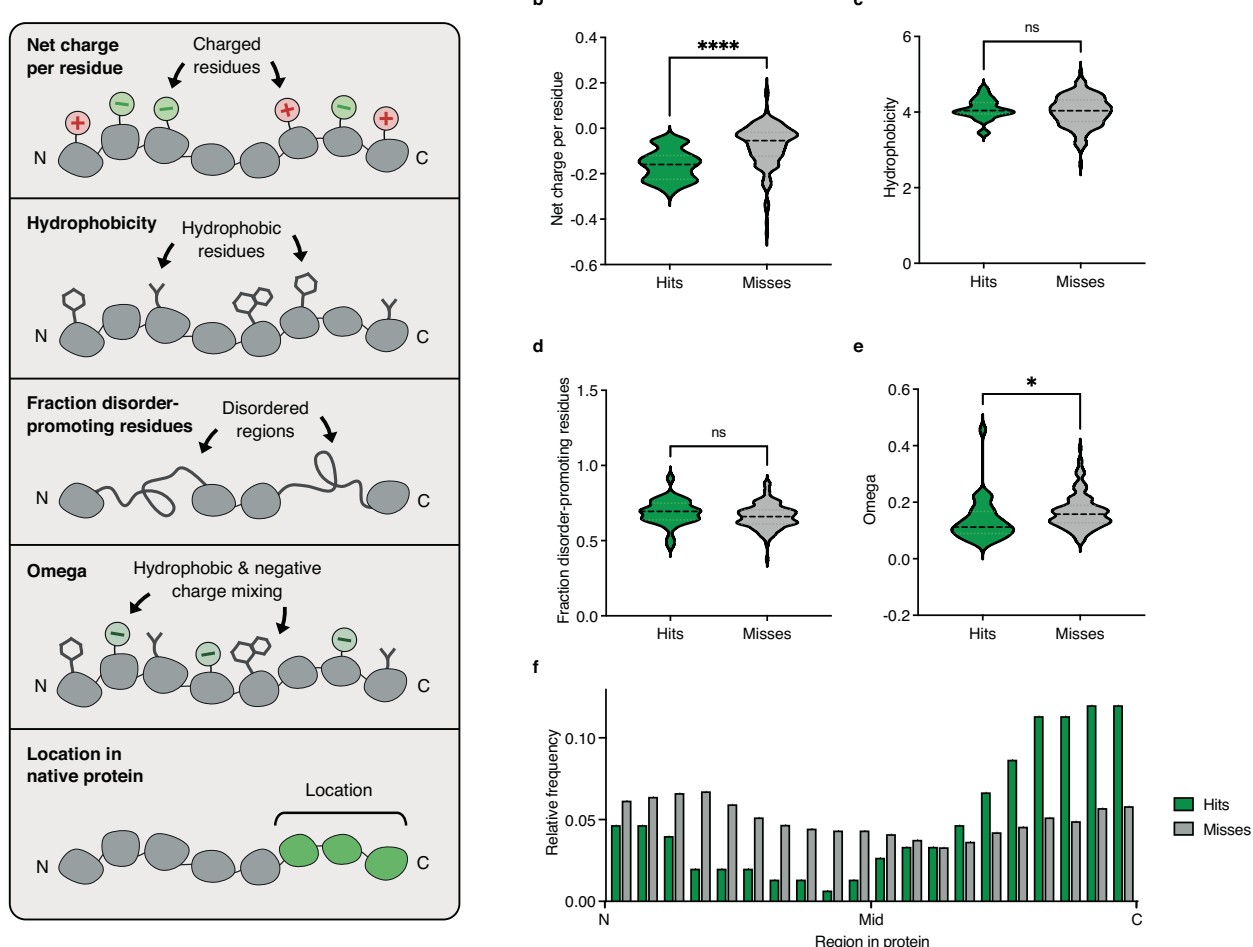

**Fig. 2 | Strong activators contain acidic residues and are located within the C-termini of their native proteins. a** Schematic depicting five AD properties explored using activation data from experimental testing of 230 ADs: the effect of net charge per residue, hydrophobicity, fraction of disorder-promoting residues, Omega, and location in native protein on AD activity (*Created in BioRender. Giddins, M. (2025)* https://BioRender.com/fbepyrg). **b**–**e** Net charge per residue (lower net charge per residue = more acidic), hydrophobicity, fraction of disorder-promoting residues, and Omega (degree of mixing between four hydrophobic residues (W, F, Y, and L) and acidic residues (D and E), lower = greater mixing), respectively, for hits (green) and misses (gray). Each point represents one AD ($n$ = 230 ADs,

measured as mean values from two independent transfections in HEK293T cells). The significance of the difference in values between hits and misses was assessed via unpaired two-sided t-test. For ****, *, and ns, $p < 0.0001$, 0.0161, and not significant, respectively. **f** The relative position of each of 230 tested ADs within its native protein is shown (see "Methods" for details). N, Mid, and C refer to the N-terminus, middle, and C-terminus, respectively, of each AD's native protein. Relative frequency indicates how often hits (green) or misses (gray) are derived from each region. We observe an enrichment of hits derived from the C-terminus of their native proteins. Source data are provided as a Source Data file (Source Data.zip).

fusions as strong or medium strength ADs (precision = 0.39 and 0.36, respectively). These findings indicate that relying exclusively on computational approaches for activator discovery will result in overlooking potent fusions and misclassifying many inactive fusions as functional, underscoring the importance of empirical testing[46].

We next aimed to investigate how the position of an AD within its native protein influences its functionality in synthetic environments. Our data show that hits were more likely to be located within the C-termini of their native proteins, while misses did not show a strong bias in derivation (Fig. 2f, and Supplementary Fig. 4b; $p < 0.05$). Although further studies are needed to support a definitive interpretation of these data, our findings suggest that accounting for the position of a biological part within its native context may help predict its function when grafted onto a synthetic scaffold.

## High-throughput screening of thousands of AD combinations
In natural systems, gene expression is regulated by multiple transcription factors (TFs) working in concert to recruit transcriptional machinery[47]. Several groups have attempted to mimic this process by

building CRISPR tools composed of multiple ADs[11–15]. Although these multi-domain activators have proven transformative in a range of applications, their efficacy varies widely across gene targets and cell types[16]. Furthermore, researchers often fail to screen constructs for cellular toxicity, an oversight that can ultimately limit their utility.

Previous efforts to create multi-domain activators employed labor-intensive protocols that permitted the investigation of only a small subset of AD combinations. To overcome this barrier, we developed a method of combinatorial assembly that extends the CombiSEAL framework to protein domains and enables the generation, barcoding, UMI labeling, and subsequent testing of thousands of activator variants at a time (Supplementary Fig. 5a, b). We set out to apply our combinatorial approach to 25 high-value ADs identified from our experimental testing. Although VP16 and VP64 ADs generally yielded the greatest levels of activation across targets, only the six strongest variants were taken forward to avoid biasing the libraries towards an individual activator family. We also included the top 10 non-VP16 and non-VP64 activators across our three tested target genes, four ADs derived from previously published tools (e.g., P65),

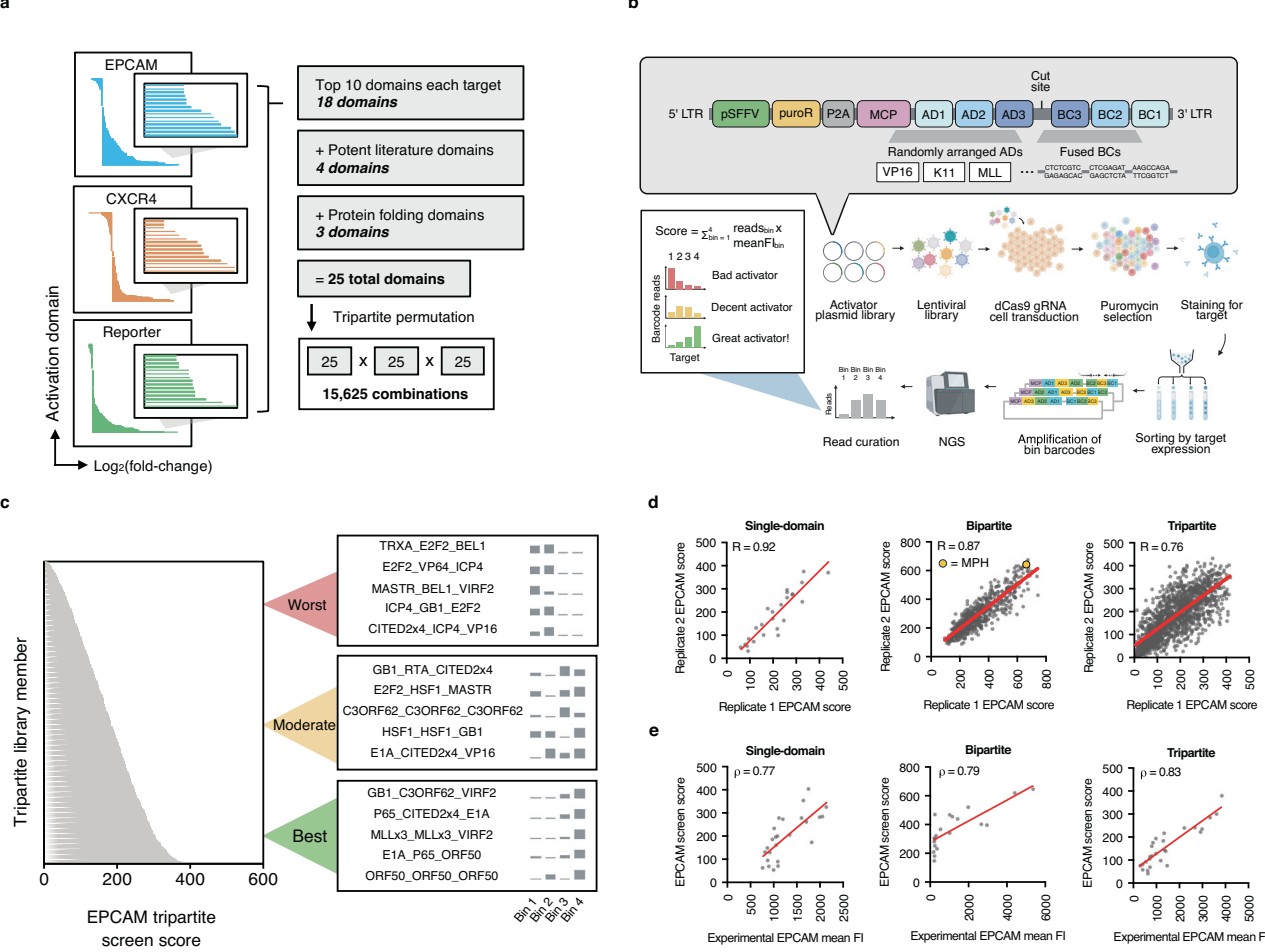

**Fig. 3 | High-throughput combinatorial screening of CRISPR activators.**
**a** Schematic depicting method for choosing 25 domains to combinatorially assemble and screen. **b** Schematic depicting approach for screening MCP activators. Domains were combinatorially fused downstream of MCP in all possible single, bipartite, and tripartite combinations (a representative tripartite construct, VP16-K11-MLL, is shown at top, in gray). Activators were then transduced into HEK293T cells, and cells were sorted based on their levels of target gene expression. All constructs within each sorted bin were sequenced, and these data were used to determine the potency of each activator. *dCas9* endonuclease-null Cas9; *gRNA* guide RNA; *MCP* MS2 coat protein; *AD* activation domain; *BC* barcode; *LTR* long terminal repeat; *pSFFV* spleen focus-forming virus promoter; *puroR* puromycin resistance gene; *NGS* next-generation sequencing; *FI* fluorescence intensity (*Created in BioRender. Giddins, M.* (https://BioRender.com/r7iqxqo). **c** *EPCAM*

screen scores of all tripartite library members, along with examples of the abundances of select activators within bins 1-4. Data are shown as the mean from two independent screen replicates (*n* = 2). **d** Correlations between *EPCAM* screen scores across two biological replicates for single-domain, bipartite, and tripartite screens. Yellow dot (bipartite panel, middle) indicates performance of MCP-P65-HSF1 (MPH, MCP component of SAM). Correlations were calculated using Pearson correlation coefficient (*R*). **e** Correlations between *EPCAM* screen scores and *EPCAM* expression derived from manual testing of individual activators isolated from the single-domain, bipartite, and tripartite libraries. Experimental data (*x* axis) are shown as the mean (*n* = 2 independent transduction replicates). Screening data (*y* axis) are shown as the mean (*n* = 2 independent screen replicates). Correlations were calculated using Spearman correlation coefficient (*ρ*). *FI* fluorescence intensity. Source data are provided as a Source Data file (Source Data.zip).

and three protein folding (PF) domains (e.g., GB1), intended to aid the solubility and stability of activator assemblies, in our set (Supplementary Table 1, Fig. 3a). The 25 aforementioned parts were C-terminally fused to MCP downstream of a glycine-serine linker using our high-throughput approach to create 25 single-domain fusions, along with 625 bipartite and 15,625 tripartite AD combinations (Supplementary Fig. 5c). To enable efficient delivery of our libraries into cells, ADs were fused to MCP (118 amino acids), which can be recruited to the dCas9 complex by binding to MS2-hairpin containing gRNAs[48]. This design bypassed the need to package dCas9 (1,368 amino acids) with each activator, likely reducing biases in delivery that arise as constructs approach the lentiviral packaging limit.

In order to evaluate the cross-target activation potential of each synthetic activator, libraries were tested against our initial panel of targets, *EPCAM*, *CXCR4*, and a synthetic reporter (Fig. 3b). For each target, cells were sorted into four bins of expression, and barcodes in

each bin were sequenced. After sequencing data were processed (Supplementary Note 1 and "Methods"), we assigned each library member an activation score based on its barcode distribution across bins (Supplementary Data 4). Our scoring method relies on the assumption that activators showing barcode enrichment in the higher target expression bins (e.g., bins three and four) are more potent than those showing barcode enrichment in the lower target expression bins (e.g., bins one and two) (Fig. 3c). Pearson correlations across biological replicates were high for all screens (Fig. 3d, and Supplementary Fig. 6a; *R* = between 0.92-0.97, 0.86-0.87, and 0.61-0.76 for single-domain, bipartite, and tripartite screens, respectively). We observed moderate correlations in activator scores across target genes. (Supplementary Fig. 6b; *R* = 0.50-0.80).

To evaluate the fidelity of our screen in ranking activators across a broad range of activity levels, we tested all 25 constructs from our single-domain library, and randomly selected 20-30 constructs from

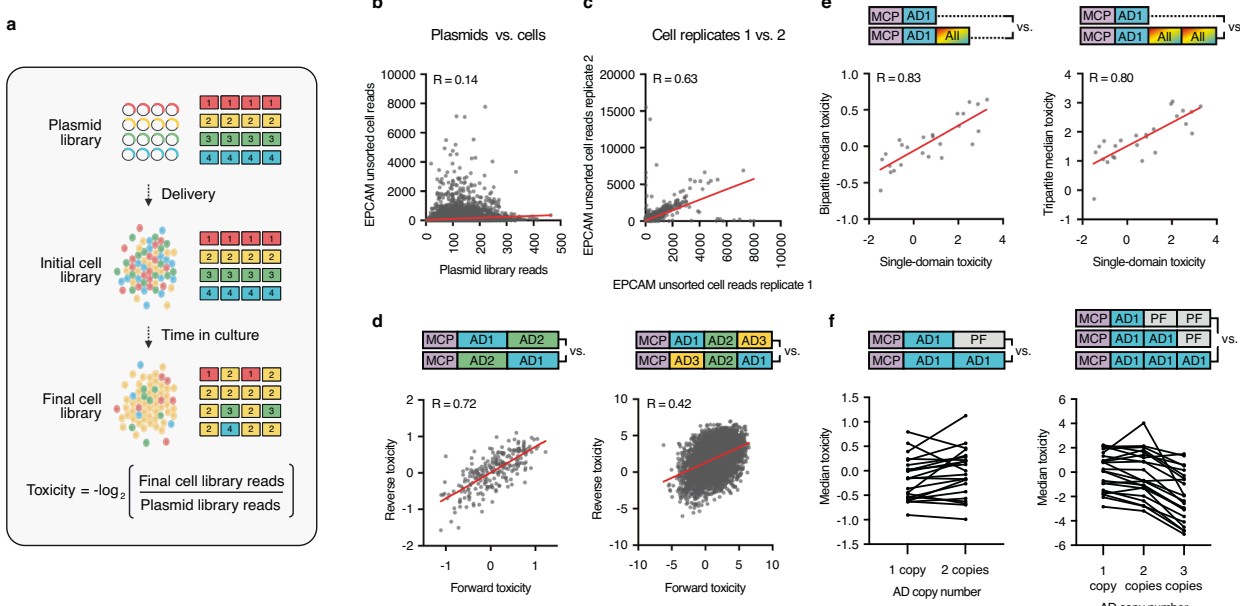

**Fig. 4 | Domain toxicity remains constant independent of surrounding context.**
**a** Schematic depicting toxicity-driven changes in library distribution from the initial plasmid mix to cells. At transduction, library member abundances in cells are expected to match the plasmid mix. Toxicity in host cells associated with certain activators leads them to grow more slowly than others. The toxicity of each activator is calculated as the negative $\log_2$ of the ratio of its number of reads in the final cell library to its number of reads in the plasmid library (*Created in BioRender. Giddins, M. (https://BioRender.com/s014rxu*). **b** Correlation between reads in the tripartite plasmid library and reads in cells expressing the tripartite library, along with dCas9 and an *EPCAM*-targeting gRNA. Cell data (*y* axis) are shown as the mean from *n* = 2 independent screen replicates; plasmid data (*x* axis) derive from one replicate. **c** Correlation between reads from two independent transductions of the tripartite library into dCas9-, *EPCAM* gRNA-expressing cells. Data shown represents cells harvested 11 days post-infection, following three passages in 0.5 μg/mL

puromycin. **d** Correlations between toxicity scores of activators in forward (AD1-AD2) and reverse (AD2-AD1) orientations in bipartite and tripartite screens. **e** Correlations between toxicity of individual ADs within the single-domain and bipartite screen (left) and single-domain and tripartite screen (right). For each AD, values were derived by calculating the median toxicity across all constructs containing that AD. **f** Median toxicity scores of 22 unique ADs (*n* = 22 ADs), calculated across all constructs where the AD was present at the indicated copy number. Inert PFs occupy any position not filled by an AD. For (**c**, **d**) data are shown as the mean (*n* = 2 independent screen replicates). For (**e**, **f**) median values were calculated across constructs based on mean screen scores (*n* = 2 independent screen replicates). For (**d**–**f**) lower (more negative) values correspond to lower toxicity. For (**b**–**e**) correlations were calculated using Pearson correlation coefficient (R). *MCP* MS2 coat protein; *AD* activation domain; *PF* protein folder. Source data are provided as a Source Data file (Source Data.zip).

each of our bipartite and tripartite libraries–spanning low, intermediate, and high screen scores–via one-by-one transduction against the original screening targets. Activator scores derived from one-by-one experimental testing correlated well with screen-produced scores, verifying the accuracy of our high-throughput approach (Fig. 3e, and Supplementary Fig. 7; *ρ* = 0.54-0.83 for all screens).

### Synthetic activators display pervasive toxicity independent of inter-domain interactions

Upon examining our screening data, we found that only 10.1%, 10.7%, and 7.9% of tripartite library members met the read threshold cut-offs implemented to accurately quantify their function for the *EPCAM*-, *CXCR4*-, and fluorescent reporter-targeting screens, respectively. Furthermore, in most instances, the distribution of library members within cells correlated poorly with the distribution of library members in the plasmid mix (Fig. 4a, b, Supplementary Fig. 8; *R* = 0.02-0.15, 0.09-0.47, and 0.14-0.15 for single-domain, bipartite, and tripartite screens, respectively). Conversely, independent cell populations transduced with the same library showed similar abundances for most members–suggesting that the observed discrepancies between plasmid and cell distributions arose from biological factors rather than poor sampling. (Fig. 4c, and Supplementary Fig. 9; *R* = 0.99, 0.66–0.99, and 0.54-0.83 for single-domain, bipartite, and tripartite screens, respectively). In light of these findings, we hypothesized that many activators elicit toxic effects that restrict cell growth and promote library skew (Fig. 4a).

To quantify toxicity within our screening data, we assigned each library member a toxicity score based on the ratio between its read count in the plasmid library and its read count in the transduced cell pool (Supplementary Data 5, Fig. 4a, see "Methods"). The high correlation between toxicity scores derived from cell populations expressing *EPCAM*- and *CXCR4*-targeted gRNAs ruled out the gRNA or target gene as drivers of toxicity and enabled us to average toxicity scores across targets for all subsequent analyses (Supplementary Fig. 10; *R* = 0.92, 0.77, 0.75 for single-domain, bipartite, and tripartite screens, respectively).

In order to experimentally validate our toxicity scores, we created stable cell lines out of 27 members of our tripartite library. We then mixed each stable line at a 1:1 ratio with a cell line expressing a control protein, MCP fused to GFP (MCP-GFP), and passaged the mixture two times (Supplementary Fig. 11a). With every split, the proportion of GFP-positive cells in each mix was quantified. We rationalized that, in cell mixes expressing toxic activators, MCP-GFP-expressing cells would quickly overtake the population. Most activator populations experienced reductions in size, up to 80%, over the course of the experiment (Supplementary Fig. 11b). As anticipated, decreases in activator populations correlated with screen-produced toxicity scores (Supplementary Fig. 11c, *ρ* = 0.58). These findings, paired with our observations that toxicity correlated only weakly with construct length (Supplementary Fig. 12, *ρ* = 0.17, 0.36, and 0.14 for the single-domain, bipartite, and tripartite screens, respectively) confirm that activator depletion from cells in our high-throughput screens reflects true

biological toxicity rather than deficiencies in lentiviral packaging or delivery of constructs to cells.

To determine whether toxicity remained consistent across cell lines and species, we delivered a small library composed of 56 randomly isolated tripartite activators into HEK293T (human), HeLa (human) and N2A (mouse) cells and passaged each mix four times before harvesting the cells and quantifying each construct's toxicity. Toxicity scores correlated well across all cell lines (Supplementary Fig. 13, R = 0.59, 0.82, 0.66 for HEK293T vs. HeLa, HEK293T vs. N2A, and HeLa vs. N2A, respectively), suggesting that the perturbations to cellular fitness induced by toxic activators are consistent across models and species, and that the underlying mechanisms driving this feature are conserved across different cellular environments.

We next investigated the relationship between toxicity and squelching, which occurs when strong activators sequester essential coactivators along with core transcriptional components. This phenomenon can perturb overall gene expression, leading to reduced cell fitness[49–53]. We hypothesized that our strongest activators, which presumably recruit the transcriptional machinery most efficiently, would produce the greatest levels of toxicity. However, toxicity did not show positive correlations with activator potency and even exhibited negative correlations with activation in some instances (Supplementary Fig. 14; $\rho$ = -0.12 to -0.63, -0.18 to -0.40, and 0.00 to 0.07, for the single-domain, bipartite, and tripartite screens, respectively). These data suggest that the underlying driver of toxicity extends beyond activator strength and likely involves mechanistically distinct processes–not always linked to transcriptional output.

We found that toxicity scores of multi-partite constructs could be predicted by summing the toxic effects of their individual parts (Supplementary Figs. 15 and 16a, b; R = 0.64 and 0.49 for the bipartite and tripartite screens, respectively). In line with these findings, our analyses revealed that AD pairs in forward and reverse orientations (e.g., AD1-AD2 vs. AD2-AD1) yielded similar toxicity scores in both bipartite and tripartite contexts (Fig. 4d; R = 0.72 and 0.42 for the bipartite screen and tripartite screens, respectively). ADs appeared to exert consistent effects on cell growth independent of their location within the bipartite or tripartite array (Supplementary Fig. 17a). Furthermore, the relative toxicity trends between component ADs remained consistent across screens of single-, bi-, and tripartite fusions, suggesting that, in general, each domain's toxic effects are only minorly impacted by its surrounding context. (Fig. 4e, and Supplementary Fig. 17b; R = 0.83, 0.80, and 0.82 for single-domain vs. bipartite, single-domain vs. tripartite, and bipartite vs. tripartite correlations, respectively). Finally, we determined the degree to which domain copy number affects toxicity by evaluating the toxicity scores of constructs consisting of one, two, or three copies of the same AD (where PFs, determined to be inert (Supplementary Fig. 18, Supplementary Note 2) filled the remaining positions in the fusion when necessary; see "Methods"). Overall, adding more copies of the same AD onto the MCP scaffold elicited only minor changes in toxicity. However, in tripartite libraries, three AD copies yielded consistently reduced toxicity levels, possibly as a result of the instability and turnover of these repetitive constructs (Fig. 4f).

In line with previous reports demonstrating an association between phase-separated condensates and cellular toxicity, we observed a positive correlation between each activator's toxicity and its fraction of disorder-promoting residues across all screens (Supplementary Fig. 19a, $\rho$ = 0.48, 0.43, and 0.27 for the single-domain, bipartite, and tripartite screens, respectively)[54–59]. Interestingly, toxicity and hydrophobicity exhibited a clear negative correlation. (Supplementary Fig. 19b, $\rho$ = -0.44, −0.34, and −0.20 for the single-domain, bipartite, and tripartite screens, respectively). These findings might stem from the propensity of hydrophobic activators to misfold within the cell, a process which, in turn, would prevent them from engaging in toxic behavior[60,61].

Overall, our data indicate that toxicity, in contrast to activation (as discussed below), does not depend strongly on interactions between ADs or the spatial arrangement of their toxic binding partners. Furthermore, our findings that increases in AD copy number do not enhance toxicity suggest that the molecular mechanism driving this property rapidly saturates. To explain these results, we propose a simple model in which the binding of ADs to other proteins, regardless of the 3D configuration of these interactions or effects of neighboring domains, is sufficient to perturb cellular fitness.

## Domain-domain interference increases with activator complexity

We next leveraged our screening data to address key questions about combinatorial activator assembly and its impact on activation strength. First, we employed our bipartite screening results to investigate pairwise AD interactions. While the strength of bipartite tools could generally be predicted by summing the activation potentials of their individual domains (Supplementary Fig. 20, R = 0.67, 0.50, and 0.61 for screens targeting *EPCAM*, *CXCR4*, and the reporter, respectively), we observed notable exceptions to this rule (Fig. 5a, and Supplementary Fig. 21a). The human HSF1 AD performed inconsistently across target genes when tested alone, yet proved a particularly good partner in the bipartite screen. This AD derives from a key regulatory protein involved in the heat shock response[62,63]. HSF1's strong cooperativity in our screen may stem from its natural ability to regulate a plethora of diverse targets, each possessing a unique mixture of regulators with which HSF1 must effectively engage. Unlike HSF1, CITED2x4, composed of four copies of the human CITED2 AD, poisoned activator function when fused adjacent to MCP but showed the expected additive behavior when placed in the most C-terminal position of the fusion[64]. Because CITED2x4 contributed similarly to activator toxicity independent of its fusion position, we hypothesize that, when adjacent to MCP, CITED2x4 does not cause instability of the synthetic construct but rather interferes with other aspects of MCP's function, such as its ability to bind to its gRNA target. As expected, PF domains, the B1 domain of the streptococcal protein G (GB1) and the *Escherichia coli* thioredoxin protein (TRXA), produced no activation as pairings with each other and high activation only when paired with potent ADs[65,66]. Interestingly, potent ADs paired with PFs often performed better than potent ADs paired with other ADs (Supplementary Fig. 21b). These results, which are in line with prior work, highlight that many AD-AD interactions are antagonistic and indicate that the indiscriminate combination of strong domains will not always lead to improved tools[28,67].

Building on this, we investigated the effect of copy number on AD strength. Prior studies have assumed that recruiting more copies of the same AD to a target locus leads to greater activity[13,15,19,25,68–70]. For instance, researchers typically rely on VP64, four copies of a fragment of the herpesvirus activator protein, VP16, to modulate target gene expression, rather than using VP16 on its own. Evaluating the potency of constructs consisting of multiple copies of the same domain revealed that the number of AD copies fused to MCP does not affect activator strength, except within some tripartite assemblies, where copy number effects vary depending on the target gene (Fig. 5b, and Supplementary Fig. 22a). In line with our findings on activator toxicity, these results suggest that, in MCP contexts, a single AD tends to saturate interactions with its targets such that additional copies of the same motif do not further improve gene activation.

Finally, we explored the effects of domain interactions and spatial arrangements on activator performance. We found that the activation potential of constructs in forward and reverse orientations correlated better for bipartite than for tripartite activators (Fig. 5c, and Supplementary Figs. 22b and 23a; R = 0.72-0.86 and 0.21-0.32 for bipartite and tripartite screens, respectively). To verify that tripartite order effects did not stem from assay noise, we focused our analyses on the subset

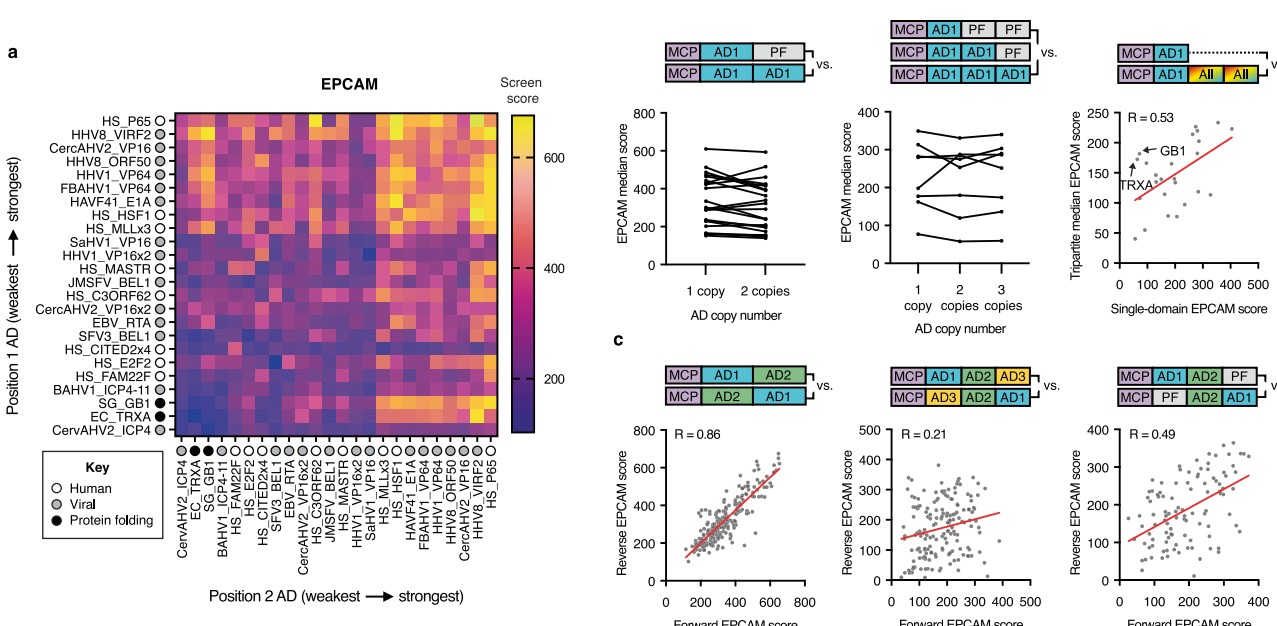

**Fig. 5 | Domain-domain interference affects activator performance. a** *EPCAM* bipartite screen scores produced by each of 25 ADs in position one, the position closest to MCP (*y* axis), upon pairing with each of 25 ADs in position two, the position most distal to MCP (*x* axis). Purple and yellow coloring highlights lower- or higher-performing combinations, respectively. ADs are ordered from weakest- to strongest-performing within the single-domain *EPCAM* screen. Each activator is annotated based on the species it derives from or its functionality as a human AD (white dots), viral AD (gray dots), or protein folder (black dots). **b** Median EPCAM screen scores of 22 unique ADs (*n* = 22 ADs) present at the indicated copy number within bipartite and tripartite screens. Median values were calculated across all constructs containing the AD at the listed copy number, with inert PF domains occupying any remaining position. **c** Correlations between *EPCAM* screen scores of

activators in forward (AD1-AD2) and reverse (AD2-AD1) orientations in bipartite and tripartite screens (left and middle), or in the tripartite screen (right) excluding any activator that does not contain one PF. **d** Correlations between *EPCAM* screen scores of individual ADs within single-domain and tripartite screens. Tripartite values were produced by calculating the median *EPCAM* screen scores of all constructs containing a given AD. Arrows point to PFs (GB1 and TRXA). For (**a**, **c**) data are shown as the mean (*n* = 2 independent screen replicates). For (**b**, **d**) median values were calculated across constructs based on mean screen scores (*n* = 2 independent screen replicates). For (**c**, **d**) correlations were calculated using Pearson correlation coefficient (*R*). *MCP* MS2 coat protein; *AD* activation domain; *PF* protein folder. Source data are provided as a Source Data file (Source Data.zip).

of tripartite activators containing one inert PF domain. We postulated that these tripartite constructs, which contain only two ADs, should behave similarly to bipartite activators (e.g., PF-AD1-AD2 vs AD2-AD1-PF, see "Methods"). As expected, activation scores of constructs in forward and reverse orientations correlated better for tripartite combinations containing two ADs and one PF than those containing three ADs (Fig. 5c, and Supplementary Fig. 22b; *R* = 0.49–0.63 for two AD-containing tripartite constructs within all screens). In line with this finding and the aforementioned high degree of AD-AD antagonism observed among many activators, the activation behavior of component ADs displayed a higher correlation between the single-domain and bipartite screen than between the single-domain and tripartite screen (Fig. 5d, and Supplementary Fig. 23b; *R* = 0.71-0.82 and *R* = 0.38-0.53, respectively). These data suggest that, unlike toxicity, an activator's transcriptional potency within tripartite contexts depends on the interactions and spatial arrangements of neighboring domains and their respective cofactors.

**Cofactor binding is sufficient for toxicity but not for activation**
Our high-throughput screening data supports a model wherein the binding of an AD to other proteins can perturb cellular fitness irrespective of adjacent domains. This loose requirement for inducing toxicity stands in contrast to the more stringent demands of transcriptional activation, where we hypothesize that interactions with essential cofactors must not only take place, but also lead to the formation of productive and appropriately arranged complexes to drive target gene expression. To address this hypothesis, we investigated patterns in activator binding by performing affinity purification-mass

spectrometry (AP-MS) on cells expressing a panel of activators with diverse activation and toxicity scores (Fig. 6a, and Supplementary Data 6).

As expected, PCA analysis of our mass spectrometry data performed using fold change enrichment of proteins identified across pull downs revealed that all activators composed of the same ADs in different orders (e.g., AD1-AD2 vs. AD2-AD1)–even those with three domains that produced vastly different activation scores–displayed a high degree of clustering (Fig. 6b). In addition, analysis of binding partner enrichment across constructs showed extremely high correlations among both bipartite and tripartite activators composed of the same domains in different orders (Fig. 6c, and Supplementary Fig. 24; *R* = 0.89-0.97).

To further investigate inter-domain interference within tripartite activators, we compared the binding profiles of bipartite and tripartite activators that differed by only one domain (e.g., AD1-AD2 vs. AD1-AD2-AD3). We hypothesized that, if inter-domain interference were affecting binding, tripartite activators would associate with the same or fewer proteins than their bipartite counterparts. Instead, our data show that tripartite constructs consistently interact with more proteins than their associated bipartite counterparts (Fig. 6d).

Overall, our findings support a model where ADs bind consistently to their respective cofactors, regardless of their order or neighboring domains. The role of domain order likely lies in allowing these bound cofactors to adopt the appropriate spatial orientation for engendering robust transcriptional activation. In this scenario, the addition of a third domain likely increases the complexity of potential binding geometries and exacerbates inter-domain interference.

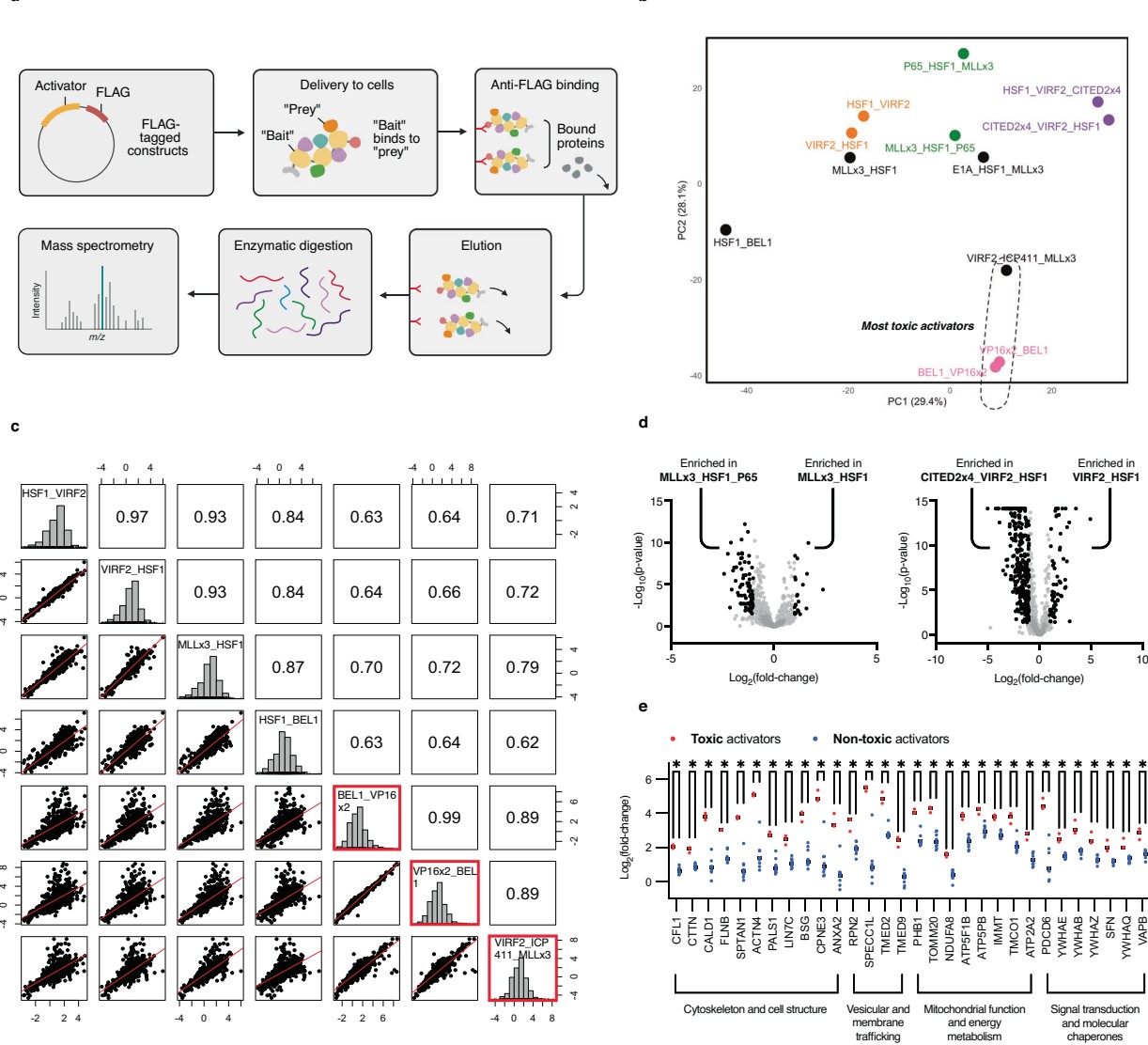

**Fig. 6 | Domain spatial arrangement does not influence activator binding.**
**a** Workflow for evaluating activator binding. Twelve activators were C-terminally FLAG-tagged and transfected into HEK293T cells expressing dCas9 and a *CXCR4*-targeting gRNA. Cells underwent protein extraction followed by anti-FLAG affinity purification, and resulting protein complexes were enzymatically digested and analyzed by mass spectrometry. Each construct was tested in $n = 4$ independent biological replicates (*Created in BioRender. Giddins, M.* (https://BioRender.com/l3knchv). **b** Principal Component Analysis (PCA) of binding profiles for each activator. Forward-reverse activator pairs are represented by identically-colored dots. Toxic activators are encircled by an ellipse. Axis labels indicate the percentage of total variance explained by each principal component. **c** Pairwise comparisons of binding profiles across all bipartite activators and a toxic tripartite activator. The grid layout shows comparisons between constructs on the diagonal. Each off-diagonal plot (left of diagonal) and R value (right of diagonal) compares the construct in its row (x axis) against its column (y axis). Dots represent individual proteins' log₂(enrichment in binding over the unfused MCP control) for the two

constructs being compared. Only proteins detected in all conditions are shown. Histograms along the diagonal show each construct's distribution of log₂(enrichment in binding over the unfused MCP control). Toxic activators (BEL1_VP16x2, VP16x2_BEL1, and VIRF2_ICP4-11_MLLx3) are boxed in red. Correlations were calculated using Pearson correlation coefficient ($R$). **d** Volcano plots showing differential protein binding between MLLx3_HSF1 vs. MLLx3_HSF1_P65 and VIRF2_HSF1 vs. CITED2x4_VIRF2_HSF1, with log₂(fold-change) in protein binding (x axis) vs. -log₁₀(p value) (y axis). Proteins significantly enriched in specific constructs are highlighted in black. **e** Log₂(fold-change) in binding for toxic (red) and non-toxic (blue) activators. Proteins are grouped by cellular function. The significance of the difference in values between toxic and non-toxic activators was assessed via unpaired two-sided t-test. Only interactions with $p < 0.0001$ are shown (exact p-values range from < 0.000001 to 0.000024). Two top-performing, low-toxicity activators, MHV and MMV (described below), are included in the non-toxic group. For (**b**–**e**) data represent mean values from $n = 4$ biological replicates. Source data are provided as a Source Data file (Source Data.zip).

We next set out to better understand the biological mechanism driving activator toxicity, which remained constant independent of domain order or neighboring ADs. The binding profiles of toxic activators showed a high degree of clustering within our PCA analysis and a high correlation in binding partner enrichment, even among those activators composed of entirely different domains (Fig. 6b, c and Supplementary Fig. 24; $R = 0.89$-$0.99$ for all toxic activators). Proteins strongly enriched among toxic activators include those involved in the

cytoskeleton and cell structure (e.g., CFL1, CTTN), signal transduction (e.g., YWHAE), and mitochondrial function (e.g., TOMM20, NDUFA8) (Fig. 6e). These findings, paired with the absence of a consistent relationship between activation and toxicity within our high-throughput screens (Supplementary Fig. 14), challenge the commonly held notion that activator toxicity derives exclusively from squelching of transcriptional cofactors and suggest that this property results from broad disruptions to multiple cellular pathways.

## MHV and MMH activators show enhanced activity across targets and cell lines

Finally, we set out to identify activators with improved performance compared to current state-of-the-art CRISPR tools. Based on our screening results, we identified 25 activators from each of our combinatorial libraries that performed well across all three targets (Supplementary Figs. 25a, b). Bipartite and tripartite constructs were stably integrated into cells and targeted to four target genes, *CD45*, *EPCAM*, *EGFR*, and *CXCR4*. Consistent with our observation of strong domain-domain interference in fusions containing three ADs, tripartite activators generally produced similar or lower activation levels than their bipartite counterparts (Fig. 7a, Supplementary Figs. 26a-c).

From these validation studies, we selected two promising activators based on their high activity across targets, low toxicity profiles (Supplementary Data 5), and distinct domain compositions: MCP fused to the human HSF1 AD and the human herpesvirus 8 VIRF2 AD (MHV) and MCP fused to three copies of the CBP-recruiting fragment from the human MLL protein and the human HSF1 AD (MMH). MHV proved our strongest overall activator, displaying robust activation across target genes. MMH, a construct composed entirely of human domains, presented potential advantages in therapeutic settings where immunogenicity would pose a concern. To evaluate the performance of MHV and MMH against the current state-of-the-art, we opted to test them against the gold-standard MCP activator, SAM, composed of dCas9-VP64 paired with an MCP-P65-HSF1 (MPH) fusion (Fig. 7b and Supplementary Fig. 27)[12].

Each activator was tested against 17 targets in HEK293T cells—five surface protein genes (*CD2*, *CD45*, *EPCAM*, *EGFR*, and *CXCR4*) using a protein-based (flow cytometry) readout and 12 non-surface protein genes (*TTN*, *HBG1*, *RHOXF2*, *NEUROD1*, *MIAT*, *ACTC1*, *ASCL1*, *IL1RN*, *IL1B*, *ZFP42*, *LIN28A*, and *IL1R2*) using an RNA-based (RT-qPCR) readout. MHV showed improved activation over SAM against most target genes, while MMH performed more similarly to SAM, with smaller and less frequent gains (Figs. 7c, d and 7g, Supplementary Fig. 28).

Unlike most synthetic TFs, such as zinc fingers and TALEs, CRISPR-based transcriptional effectors can upregulate multiple loci at once with high precision, rendering them particularly attractive for complex cell engineering and therapeutic endeavors[71]. In order to address the multiplexing potential of our tools, each activator was tested with a mix of gRNAs targeting the surface protein genes *CD45*, *EPCAM*, *EGFR*, and *CXCR4*. MHV and MMH performed better than SAM across the majority of target genes, underscoring their enhanced utility for gene regulation in multiplexed settings (Fig. 7e).

Having optimized our activators in the commonly used HEK293T cells, we sought to determine the plasticity of our tools across additional biomedically relevant human and mouse cell lines. We individually tested each MCP activator against a panel of human surface protein target genes (*CD2*, *CD45*, *EPCAM*, *EGFR*), human non-surface protein target genes (*TTN*, *HBG1*, *RHOXF2*, *NEUROD1*, *MIAT*, *ACTC1*, *ASCL1*, *IL1RN*, *IL1B*, *ZFP42*, *LIN28A*, and *IL1R2*), and mouse non-surface protein target genes (*Ttn*, *Hbb-bh1*, *Actc1*, *Neurog2*, *Ins2*, *Sim1*, and *Mef2d*) within HeLa and HCT116 or N2A lines, respectively. In HeLa cells, both MHV and MMH produced modestly better activation than SAM across protein targets but performed similarly on RNA targets. In HCT116 and N2A cells, MHV showed stronger activation across most targets, whereas MMH performed similarly to SAM (Fig. 7f, g and Supplementary Figs. 29–31).

Off-target gene modulation poses a serious concern for any CRISPR activator technology, given previous findings that dCas9 can bind promiscuously throughout the human genome[72,73]. To ensure that MHV and MMH do not produce unintended effects on transcription, we performed unbiased RNA-sequencing on cells expressing our activators targeted to the *TTN* or *HBG1* genes. Gene expression data from lines expressing each activator displayed a high correlation with gene expression data from the negative control condition consisting of MCP not fused to any protein for both targets (Fig. 7h and Supplementary Figs. 32–34; $R = 0.96$-$0.98$ for MHV vs. MCP, MMH vs. MCP, and SAM vs. MCP for both targets). After filtering away genes with low baseline expression (< 1 TPM), no activator induced significant upregulation (≥2-fold change in expression) in more than four genes for either target, suggesting that, overall, these tools are highly specific.

## Discussion

Here, we develop an efficient approach for creating and functionally evaluating thousands of combinatorial protein variants at once. Our system, which circumvents the need to manually create and test constructs one by one, reduces the time, costs, and labor of searching through large combinatorial libraries. Unlike other high-throughput approaches, our platform is not constrained by size limitations imposed by oligo chip synthesis, enabling combinatorial assembly to be performed with full-length domains or entire proteins. Our DNA barcoding technique enables facile identification of any library member through sequencing of a short DNA barcode. This approach bypasses the need for long-read sequencing to link each barcode with its associated variant, a cumbersome requirement common in alternative methods[74–77]. In addition, our method makes use of UMIs on each generated construct, which improve data quality by distinguishing between multiple cells expressing the same activator[78–81] (Supplementary Fig. 35). Finally, while we apply this approach to identify more potent CRISPR activators, our platform can be used to engineer a variety of biomedically relevant proteins, like chimeric antigen receptors, synthetic sense and response modules, and designer polymerases.

We employ our method to rapidly explore an expansive combinatorial landscape of multi-domain CRISPR activators—elucidating several valuable insights into activator biology and cooperativity. In our initial, one by one testing of 230 activators, many constructs exhibited differences in behavior between endogenous targets and the synthetic reporter. This finding underscores the shortcomings of previous studies, which designed CRISPR activators based exclusively on their performance on a single, often artificial target, and suggests that such an approach may mislead researchers regarding the generality of their tools[11,17,26–31]. Furthermore, we noticed that many constructs were only able to modulate a subset of target genes. We propose that this AD target selectivity has arisen as a natural consequence of the inherent nature of TFs, whose small and flexible binding motifs allow them to bind to more than 25,000 sites throughout the genome[82]. By ensuring that TF binding is necessary but not sufficient for regulating transcription, cells impose an additional layer of regulation that limits the expression of nearby genes to appropriate cellular contexts. This stringent control confers a level of selectivity to the intrinsically promiscuous interactions between TFs and DNA. Further elucidation of this complex regulatory grammar is needed to enable more precise synthetic modulation of gene expression.

Our data identified activator toxicity as a prominent driver of library dropout in cells. Unlike activation, the toxicities of multi-domain activators—even those containing three ADs—were well approximated as the sums of the toxicities of their individual components. We also found that toxicity remained relatively unaffected by domain order across all screens, a finding which stands in contrast to our observation that AD order significantly affects target gene activation in tripartite contexts. The impact of inter-domain interference on activation as compared to toxicity establishes a distinction between the molecular underpinnings of these properties. We hypothesized that, while toxicity can arise when an activator simply binds to other proteins, activation requires bound transcriptional cofactors to adopt a precise spatial configuration that enables the assembly of a functional transcription complex at the target gene. Our mass spectrometry analyses corroborate this model, confirming that tripartite

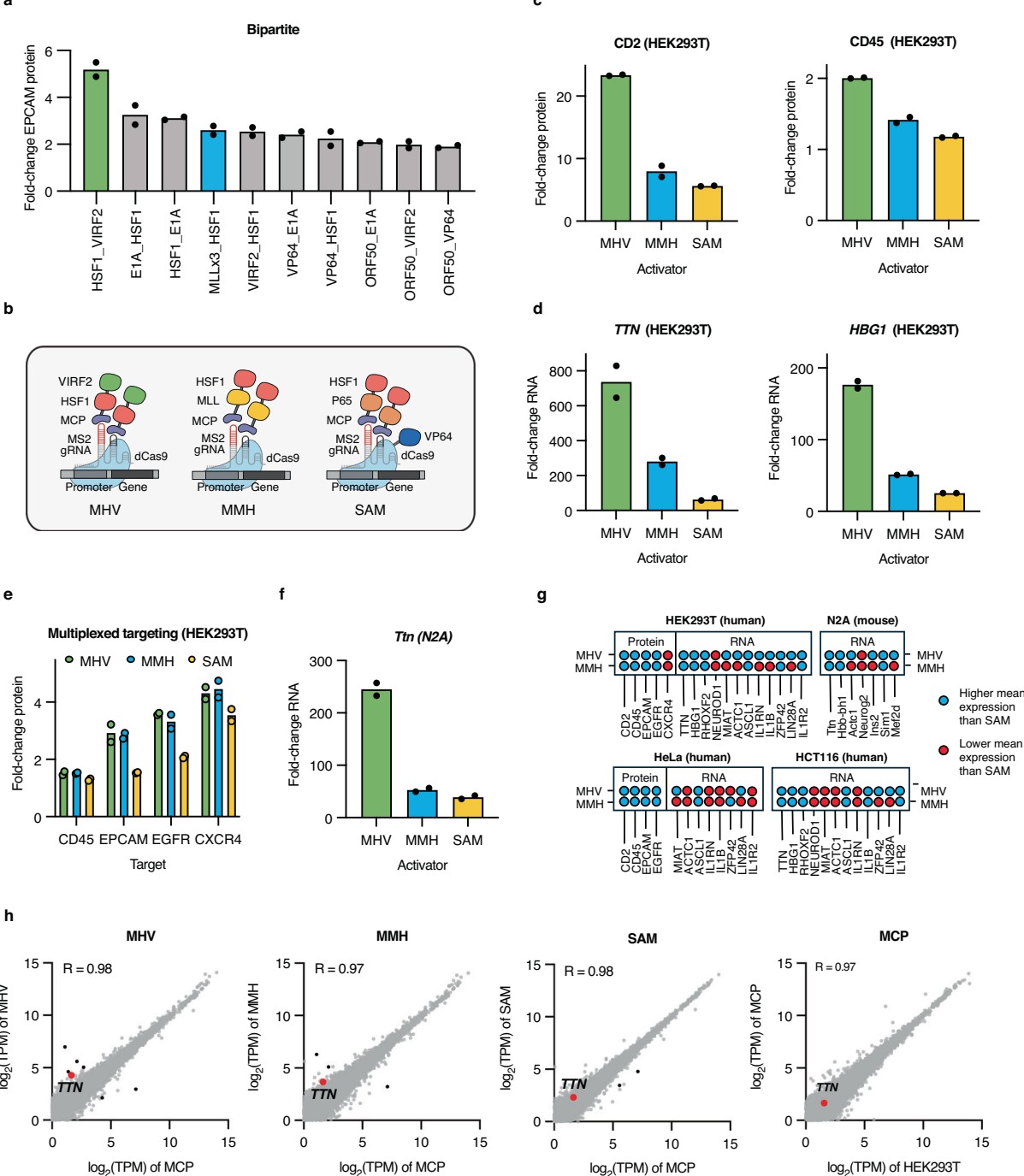

**Fig. 7 | MHV and MMH outperform SAM in multiple contexts. a** Manual valida-tion of best-performing bipartite hits. Fold-change values for the top 10 activators targeting *EPCAM* are shown. Highly potent activators, MCP-HSF1-VIRF2 (MHV) and MCP-MLLx3-HSF1 (MMH), are highlighted in green and blue, respectively. **b** Schematic depicting composition of activators taken forward for further workup: our recently identified activators, MHV and MMH, and the gold-standard MCP activator, SAM. *dCas9* nuclease-deactivated Cas9; *gRNA* guide RNA; *MCP* MS2 coat protein (*Created in BioRender. Giddins, M.* (https://BioRender.com/lgdoc71). **c**, **d** Testing MHV-, MMH-, and SAM-induced activation in HEK293T cells across multiple targets measured by flow cytometry (**c**) and RT-qPCR (**d**). **e** Multiplexed targeting of surface protein genes in MHV-, MMH-, and SAM-expressing HEK293T cells. **f** Testing activation by MHV, MMH, and SAM within mouse N2A cells. **g** Summary of activator performance across targets and cell types. Each circle represents the performance of MHV or MMH relative to SAM for a specific target

gene and cell line. Blue circles indicate targets where the activator yielded higher mean expression than SAM and red circles indicate targets where the activator yielded lower mean expression than SAM. **h** Quantifying off-target transcriptional perturbations in MHV-, MMH-, SAM-, and MCP-expressing HEK293T cells trans-fected with a *TTN*-targeting gRNA. Transcriptional aberrations driven by MCP alone were evaluated by comparing gene expression in the MCP-expressing cell line to wildtype HEK293T cells transfected with a *TTN* gRNA. *TTN* (on-target gene) is shown in red, differentially expressed genes are shown in black (defined as |log₂ fold change | > 2 and adjusted $p < 0.05$) and all other genes are shown in gray. Correlations were calculated using Pearson correlation coefficient (R). For (**a**, **c**–**f**, and **h**) data are shown as the mean of $n = 2$ independent biological trans-fections. All data were normalized to a negative-control line expressing MCP not fused to any protein. Black dots represent two independent transfection replicates.

forward-reverse pairs–even those with different activation potentials–bind to similar sets of proteins. These results underscore that binding alone is insufficient to determine activation potency and point to the critical role of 3D configurations in modulating transcriptional output.

Our protein interaction studies also highlighted distinct sets of proteins significantly more likely to be bound by toxic constructs than non-toxic constructs. These proteins play roles in broad-ranging cellular pathways and, notably, are largely unrelated to the core transcriptional machinery. These observations, paired with the distinct clustering of toxic activator interactions within our PCA analysis and absence of positive correlation between activation and toxicity, challenge the presumed link between these properties. Future studies are needed to understand the basis of activator toxicity and how these effects might be reduced to improve tool safety. Overall, our findings highlight the two-pronged functionality of our screen, which, in addition to quantifying activator strength, negatively selected against toxic tools. To our knowledge, this work provides the most comprehensive characterization of CRISPR activator toxicity to date. As a growing number of researchers strive to harness CRISPR for therapeutic applications, our observations that a large proportion of activators induced significant detriments to cell fitness highlight the need for caution when considering these tools for use in clinical contexts[83,84].

The absence of strong copy number effects within our data stands in contrast to the field's perception that using multiple copies of the same AD (e.g., VP64 instead of VP16, SunTag for recruitment of multiple VP64s, etc.) improves activation[13,15,19,25,68–70]. This result might stem from our use of MCP, which binds as a dimer to its two corresponding RNA hairpins in the gRNA, as a fusion partner[48]. This mechanism allows up to four copies of a given AD to reside on each dCas9 complex and could obviate the need to fuse more AD copies directly onto MCP.

Our comprehensive evaluation of over 15,000 activators eliminated a large proportion of suboptimal AD combinations, and showcases the need for high-throughput approaches for quickly navigating through sparsely populated landscapes. Moreover, our results provide the field with valuable negative data that will prevent future researchers from pursuing less effective designs. Our screening data yielded important insights into combinatorial activator design, such as the impact of domain-domain interactions on activator performance, the relationship between toxicity and activation potential, and the biochemical properties of effective ADs. This information is valuable for the broader field of synthetic biology and transcriptional engineering, and lays a foundation for more targeted and efficient activator construction. Future optimizations of our screening approach might include integrating machine learning to predict promising combinations of activators or expanding the parameter space to include factors such as linker lengths or additional ADs.

Although previous CRISPR activators are deployed today in broad-ranging in vitro and in vivo applications, they display inconsistent performance across targets and cellular contexts, limiting their utility. Our high-throughput screens elucidated two powerful activators, MHV and MMH. While MHV is not universally superior to SAM, it outperformed SAM in most comparisons across multiple targets and cell types. Importantly, our data point to a context-dependent component to activator performance, where gains by MHV and MMH are often substantial but not uniform across all targets or cell types. Understanding the determinants of this variability remains an important direction for future work. Beyond their performance profiles, MHV and MMH offer several advantages over existing CRISPR activators. MHV, our most potent activator, is more compact than the MCP component of SAM by 210 bp (729 bp versus 939 bp). This reduced size could facilitate easier packaging into viral vectors for gene therapy applications. MHV also induced lower levels of toxicity than the MCP component of SAM, ranking in the bottom

9th vs. 17th percentile of toxicity, respectively, in our high-throughput screen (Supplementary Data 5)–potentially enhancing its suitability for long-term expression or use in sensitive cell types. It is worth noting that VIRF2, a key component of MHV, has been recently uncovered in other studies, highlighting the convergent discovery of this potent AD[29,85]. MMH, while less potent than MHV, still achieved similar or higher levels of activation across most tested targets compared to SAM and produced relatively low toxicity, ranking in the bottom 33rd percentile of toxicity within our screen. Importantly, MMH contains all-human domains, which might lead to reduced immune responses in in vivo settings, a crucial consideration for therapeutic applications. Both MHV and MMH systems do not include the VP64 component used in many existing CRISPR activators, including SAM, which simplifies the overall system design and potentially reduces production complexity. These features, combined with their enhanced activation potential, position MHV and MMH as promising tools for a wide range of applications. That said, the incremental nature of these gains–despite extensive screening–highlights the inherent difficulty of surpassing well-optimized systems like SAM. Our results indicate that progress in this space may come through steady, context-aware refinements. We view this as a valuable insight that helps recalibrate expectations and inform future tool development.

## Methods

### Activation domain mining
We searched for trans-activation sequences in the Swissprot dataset, retrieved January 21, 2019. 1,111 sequences with regions annotated with the term "activation" were selected and manually evaluated to remove spurious results. This mining resulted in a set of 867 sequences, which were then clustered to 50% identity using CD-HIT. In order to maximize the chances that our isolated sequences would function within human systems, we selected human sequences as the representative of the clustered sequences when human sequences were available. We also extended the border of annotated regions by three amino acids on both the N-terminal and C-terminal boundaries to account for small potential errors in sequence annotation. The clustering step resulted in 201 sequence representatives which were then further filtered down through manual evaluation to a set of 123 sequences. We supplemented this set with an additional 107 ADs or full-length proteins, either sourced directly from literature or selected based on their homology to known ADs using NCBI's protein BLAST function–bringing our collection to 230 total sequences.

In assembling our list of supplemental 107 ADs, we prioritized viral domains, following the observation that a high proportion of literature activators are comprised of domains derived from viral origin (e.g., VPR, SAM, SunTag). Furthermore, we included two to three orthologues of seven domains deemed most prominent in the activator literature or most powerful upon initial testing, VP16, VP64, P65, BEL1, E1A, RTA, and HSF1, to more thoroughly explore protein space likely to be enriched for functional activators. Finally, we included double or triple copies of domains showing evidence of copy-number driven activation potential (e.g., VP16x2) in prior studies. Supplementary Data 1 shows the completed list of 230 domains taken forward for testing. Sequences were codon-optimized for human expression and modified to remove instances of homopolymers and other motifs that would hinder gene synthesis. Constructs were also modified to remove internal restriction enzyme sites that would hinder our cloning strategy.

### Vector use and design
AD sequences were ordered as TWIST BioSciences gene fragments or IDT eBlocks. All sequences were designed to contain identical upstream and downstream landing pads, enabling facile amplification via PCR and digestion for combinatorial cloning.

Manual testing on our initial set of 230 fusions (dCas9-AD fusions) was carried out in a nourseothricin resistant PiggyBac transposon vector (backbone derived from System BioSciences) expressing a CAG promoter-driven dCas9 appended with a BPSV40 NLS, along with a glycine-serine linker containing restriction sites for AD insertion and terminator sequence.

Pooled combinatorial AD testing, screen validation, and top hit work-up (MCP-AD fusions) were carried out in a pLex307 lentiviral expression vector (Addgene #41392) carrying a UCOE element to prevent transcriptional silencing followed by an SFFV promoter-driven puromycin resistance cassette, P2A skipping peptide, and BPSV40 NLS fused onto the N-terminus of the MS2 coat protein (MCP), followed by a glycine-serine linker containing restriction sites for combinatorial AD insertion (see Combinatorial activator library assembly for high-throughput screening Methods).

The dCas9-expressing transposon construct and MCP-expressing lentiviral construct were assembled using Gibson assembly via the Gibson Assembly HiFi DNA Assembly Master Mix (NEB, Cat. #E2621L). For one-by-one AD insertion into the dCas9 vector, traditional restriction cloning was employed. For *en masse* cloning into the MCP-expressing vector, our pooled combinatorial cloning protocol was employed (see Combinatorial activator library assembly for high-throughput screening Methods).

gRNAs for endogenous human and mouse gene activation or synthetic reporter activation were previously described or designed to bind between one and 200 bp upstream of the transcriptional start site (TSS) of a given gene. gRNAs were expressed either via transient transfection or via integration into target cell genomes through lentiviral delivery of pSB700 plasmids expressing puromycin resistance (single-domain testing) or blasticidin resistance (high-throughput screening). gRNAs used with MCP activators were designed to contain MS2 stem loops for MCP recruitment. gRNA sequences are listed in Supplementary Table 2.

### Individual testing of dCas9-AD fusions
All ADs were digested with BsiWI-HF (NEB, Cat. #R3553L) and NheI-HF (NEB, Cat. #R3131L) and ligated into our BsrGI-HF- (NEB, Cat. #R3575L) and AvrII (NEB, Cat. #R0174L)-digested vector downstream of dCas9. 100 ng dCas9-fused ADs were transfected in biological duplicate into cells expressing gRNAs targeting *EPCAM* or *CXCR4*, along with 10 ng eBFP2 plasmid (to enable identification of transfected cells) and 140 ng pBR322 inert stuffer plasmid (to achieve a sufficient DNA:transfection reagent ratio for optimal complexing) using FuGENE HD (Promega, Cat. #E2311). For reporter targeting, 100 ng dCas9-fused ADs were transfected in biological duplicate into HEK293T cells, along with 60 ng tdTomato reporter plasmid, 100 ng reporter-targeting gRNA, and 10 ng eBFP2 plasmid using FuGENE HD. dCas9 alone was included in all experiments for negative control purposes with either 150 ng or 170 ng pBR322 for endogenous targets or the reporter, respectively, to ensure the amount of DNA remained the same for all transfections. After 48 hours, cells were harvested, washed with PBS (Thermo Fisher. Cat. #14190144), stained with target-specific antibodies for one hour, resuspended in PBS, and prepared for flow cytometry. Median target fluorescence intensity of each AD was divided by median target fluorescence intensity of the dCas9 alone control to determine degree of target upregulation (fold-change).

### Analyses of biological patterns among dCas9-AD fusions
Using LOCALCIDER, we measured the levels of a set of biochemical traits previously associated with transactivation domains within each of 224 tested ADs (six PFs were excluded from the analysis). In addition to hydrophobicity, acidity (measured by net charge per residue), and fraction of disorder-promoting residues, we measured two compositional effects, Kappa and Omega, which represent the degree of mixing between different classes of amino acids. Kappa measures the degree

of mixing between positively and negatively charged amino acids, while Omega measures the degree of mixing between hydrophobic (W, F, Y and L), and negatively charged (D and E) amino acids.

We used UniProt to identify the location of each AD within its native protein. For a domain of length Ld appearing within a protein of length Lp at the Nth amino acid, start position was defined as (N-1)/Lp, and end position was defined as (N-1+Ld)/Lp. Using these definitions, a domain starting from the first amino acid in a protein is assigned a start value of 0, and a domain running to the last amino acid in a protein is assigned an end value of 1. Because the length of each activator's native protein is different, we first normalized each protein by dividing its length into 20 sections. We then count how frequently hits and misses are sourced from domains which overlap with each of the 20 sections. Domains can span multiple sections if every section they touch is counted. For three domains, the sequence we used in our screen contained either a mismatch or single amino acid indel compared to the canonical sequence found in UniProt. These instances were handled explicitly by applying the above algorithm to the sequence present in UniProt.

### Identifying 25 ADs for combinatorial permutation
Although VP16/VP64 ADs generally yielded the largest activation potentials across targets, six variants that performed best across *EPCAM*, *CXCR4*, and the synthetic reporter were taken forward to avoid biasing the combinatorial libraries towards an individual activator family. We also took forward a set of non-VP16/VP64 ADs that included the top 10 performs across our three targets, four ADs derived from previously published tools (e.g., P65) which either did not pass the aforementioned criteria (HHV1_VP64, HS_P65, EBV_RTA) or were not included in the manual testing (HS_HSF1), and three protein folding (PF) domains intended to aid the solubility and stability of activator assemblies (SG_GB1, HS_MBP, and EC_TRXA) in our set (Supplementary Table 1, Fig. 3a).

### Combinatorial activator library assembly
The pLex307 MCP cloning vector was designed to contain a BsrGI site and an AvrII site downstream of the glycine-serine linker at the 3' end of MCP to enable insertion of in-frame AD fusions.

AD sequences were each flanked downstream by a unique 8 bp barcode (to differentiate each AD) followed by a unique 6 bp unique molecular identifier (UMI) (to differentiate each individual AD transduction event upon delivery into cells). Each domain was designed to harbor a BsiWI restriction site upstream of the AD ORF sequence and a NheI restriction site downstream of the domain's barcode, along with a BsrGI site and an AvrII site between the 3' end of the AD ORF sequence and the start of the domain's barcode (Supplementary Fig. 5b).

Restriction digestion of the MCP cloning vector with BsrGI and AvrII alongside restriction digestion of AD inserts with BsiWI and NheI creates two compatible overhangs: the BsrGI-digested 3' overhang of the vector with the BsiWI-digested 5' overhang of the AD insert, and the AvrII-digested 5' overhang of the vector with the NheI-digested 3' overhang of the AD insert. Ligation of the digested AD insert with the digested MCP vector forms scars on either end of insert, each constituting fusions of the two original restriction sites, which cannot be digested by any of the enzymes in use (Supplementary Fig. 5b).

The BsrGI and AvrII sites between the ORF sequence and barcode of the inserted domain enable subsequent rounds of in-frame AD insertions. After cloning, the short barcode region situated downstream of the inserted domain(s) marks the composition of the fusion for short-read sequencing.

In order to create the barcoded single-domain activator library (25 members), 25 chosen AD sequences were individually PCR amplified using Q5 polymerase (NEB, Cat. #M0491L) and isolated on a 1% agarose gel using a Zymoclean Gel DNA Recovery Kit (Zymo, Cat. #D4002). Purified inserts were each digested with BsiWI-HF and NheI-HF and

further purified using a DNA Clean and Concentrator (Zymo, Cat. #D4004). In tandem, the MCP cloning vector was digested with BsrGI-HF and AvrII followed by desphosphorylation with rSAP (NEB, Cat. #M0371L) and purification using a DNA Clean and Concentrator. Each digested AD insert was ligated using T4 DNA Ligase (NEB, Cat. #M0202T) with the digested MCP vector and transformed into NEB Stable Competent *E. coli* (NEB, Cat. #C3040I). Colonies from each AD transfection were scraped to generate UMI diversity and miniprepped using a QIAprep Spin Minipep Kit (Qiagen, Cat. #27106). The 25 mini-preps were normalized and pooled to produce our single-domain library. Barcodes were sequenced to assess library member representation and UMI diversity.

In order to create the barcoded bipartite activator library (625 members), the single-domain library was digested with BsrGI-HF and AvrII followed by desphosphorylation with rSAP and purification using a DNA Clean and Concentrator. The digested single-domain library was separated into 25 separate aliquots and ligated with each of 25 digested AD inserts. Individual ligations were pooled together and concentrated using a DNA Clean and Concentrator. The concentrated ligation was electroporated into NEB 10-beta Electrocompetent *E. coli* (NEB, Cat. #C3020K). 5,000,000 colonies were scraped to maximize plasmid library coverage and UMI diversity. Scraped colonies were miniprepped using a QIAprep Spin Minipep Kit. Barcodes were sequenced to assess library member representation and UMI diversity.

In order to create the barcoded tripartite activator library (15,625 members), the bipartite library was digested with BsrGI-HF and AvrII followed by desphosphorylation with rSAP and purification using a DNA Clean and Concentrator. The digested bipartite library was separated into 25 separate aliquots and ligated with each of 25 digested AD inserts. Individual ligations were pooled together and concentrated using a DNA Clean and Concentrator. The concentrated ligation was electroporated into NEB 10-beta Electrocompetent *E. coli*. 15,000,000 colonies were scraped to maximize plasmid library coverage and UMI diversity. Scraped colonies were miniprepped using a QIAprep Spin Minipep Kit. Barcodes were sequenced to assess library member representation and UMI diversity.

### Assessment of lentiviral recombination

To evaluate the potential impact of lentiviral recombination on our screening results, we performed an experiment using two barcoded MCP activator constructs–one with an intact blasticidin resistance gene and one with a non-functional version. These constructs were mixed, packaged into lentivirus, and transduced into HEK293T cells at various MOIs. Barcode abundance was quantified after selection and passaging. Full experimental details and results are provided in Supplementary Note 3.

### High-throughput screening

HEK293T cell expressing dCas9 and an *EPCAM*- or *CXCR4*-targeting gRNA were transduced with 4 mL unconcentrated single-domain virus, 600 uL concentrated bipartite virus, or 600 uL concentrated tripartite virus per T75 flask for the single-domain, bipartite, and tripartite screens, respectively. Transduction volumes used were intended to achieve a multiplicity of infection (MOI) of 0.1 based on previous titration of lentivirus on the target cell line. Two transduction replicates (one T75 flask each, two T75 flasks each, or seven T75 flasks each for the single-domain, bipartite, and tripartite screens, respectively) were utilized per target. Cells were passaged two to four times in 0.5 ug/mL puromycin. 7,500,000, 9,000,000, or 21,000,000 infected cells per replicate per target gene were harvested for the single-domain, bipartite, and tripartite screens, respectively, followed by washing with PBS, staining with EPCAM- or CXCR4-specific antibodies for one hour, and resuspending in PBS + 2% FBS. Cells were sorted into four bins of EPCAM or CXCR4 protein expression. Each bin consisted of 12.5% of the population and covered only the extremes of the

population (two bins covering the lower extremes, two bins covering the higher extremes). 300,000, 600,000 or 1,400,000 cells were sorted into each bin for the single-domain, bipartite, and tripartite screens, respectively.

After the two to three passages, a widget of each replicate of the library-expressing cells targeting *EPCAM* or *CXCR4* was transfected with a GFP reporter plasmid harboring a constitutively active iRFP transfection marker to enable comparisons of activator activity on both chromatinized (*EPCAM* and *CXCR4*) and non-chromatinized (GFP reporter plasmid) targets. After 48 hours, 7,500,000, 9,000,000, or 21,000,000 transfected cells per replicate for the single-domain, bipartite, and tripartite screens, respectively were harvested, washed with PBS, and resuspended in PBS + 2% FBS. A population of iRFP-positive cells constituting 25% of the population was gated prior to GFP binning to confine our analysis to cells receiving similar amounts of reporter plasmid (as measured by iRFP level, which is expressed from a non-dCas9 regulated promoter). Sorting by GFP expression was carried out following the same protocol employed for EPCAM- and CXCR4-based sorting.

Genomic DNA was extracted from each bin and the unsorted population for both replicates of all three targets (*EPCAM*, *CXCR4*, and GFP reporter) by resuspending cells in 100 uL QuickExtract DNA Extraction Solution (Biosearch Technologies, Cat. #QE09050), followed by a 15-minute shaking incubation at 65 °C and a five-minute incubation at 95 °C. Activator barcodes were PCR-amplified from the gDNA of all bins (48,000, 480,000, and 1,344,000 cells per bin were estimated to have been sampled for the single-domain, bipartite, and tripartite screens, respectively), appended with Illumina adapters, and sequenced on a NextSeq 500 system (Illumina, Cat. #SY-415-1002).

### NGS analyses, UMI filtration, and elucidation of screen activation scores

NGS reads were processed by employing a computational script that extracted sequence features in specific positions between cloning scars. These features consisted of one, two, or three activator barcodes for the single-domain, bipartite, or tripartite screen, respectively, along with at least one 6 N UMI sequence intended to distinguish independent transduction events. We discarded any reads that did not contain all expected features. Any reads containing a negative control-associated barcode (see Built-in screening controls Methods) were dropped before proceeding to scoring.

For each screen, we applied a series of filtering processes (see Supplementary Table 3 for details). As an initial step, UMIs representing a high proportion of total read counts within any condition (sorted bin or unsorted population) were discarded to minimize the effects of potential PCR amplification bias or amplicon contamination.

Single-domain and bipartite screens (25 and 625 members, respectively) achieved relatively high coverage, making it feasible to separate reads for all activators into groups based on the first 3 bp or 2 bp of their UMIs (i.e., 64 UMI-groups or 16 UMI-groups, respectively). The tripartite screen (15,625 members) achieved lower coverage due to toxicity-driven library skewing, limiting our ability to effectively score multiple UMI groups per activator. Given this lower coverage, we aggregated all UMIs for each activator together before proceeding with downstream analyses.

Next, we normalized grouped counts to an average of 100 reads within each sorted condition to account for the different numbers of reads that each bin received during NGS. Before scoring the UMI groups, we discarded any whose sum of normalized read counts across sorted bins did not surpass designated thresholds to remove spurious signals from our screening data. (see Supplementary Table 3 for filtering parameters). We set this threshold higher for the tripartite analysis, in which all UMIs for a given activator were aggregated into a single group, than for the single-domain and bipartite analyses, in which UMIs were separated into multiple groups per activator.

We then assigned an activation score to each sufficiently-covered UMI group, defined as follows:

$$\sum_{bin=1}^{bin=4} normalized\ reads_{bin} \times mean\ fluorescence\ intensity_{bin} \quad (1)$$

Finally, we scored each activator by calculating the average of the activation scores for all of its UMI groups.

## Built-in screening controls

Beyond the 25 sequences selected for combinatorial permutation, we incorporated barcoded control constructs designed to assess the impact of three potential screen confounders. The first, intended to estimate the proportion of sequencing reads deriving from unpackaged plasmid in the media, lacked the sequence features required for it to be packaged into lentivirus (supernatant control). The second, intended to estimate the frequency of multiple lentiviral transduction events, contained a premature stop codon in the puromycin resistance gene (multi-copy control). The third, intended to estimate the degree of inaccurate activator scoring, expressed a premature stop codon in the MCP gene (mutant MCP control). Each of the three control constructs contained the same number of barcodes as the rest of the activator constructs in the given screen (i.e., for the bipartite screen, the control constructs expressed two barcodes, in line with the other members of the bipartite library). Single-domain controls were spiked into the library at ~3.6% of the total (the same concentration as all other members), and bipartite and tripartite controls were spiked into corresponding libraries at ~1.5% of the total (a higher concentration than other members) to ensure their behavior could be confidently evaluated. Supplementary Note 1 outlines the performance of all built-in screening controls.

## Analysis of screen-derived toxicity

To calculate toxicity, we compared sequencing results from the non-sorted cell population to sequencing results from the plasmid library. We began, as described in the NGS analyses, UMI filtration, and elucidation of screen activation scores Methods, by removing control barcodes and outlier UMIs from the non-sorted bins. The plasmid libraries did not contain outlier clonal-identifiers, so we did not apply this step to plasmid library reads. We then combined all UMIs into a single group, normalized the read counts between conditions to a mean of 100 reads, and assigned each activator a toxicity score defined as follows:

$$log_2\left(\frac{Normalized\ reads_{unsorted\ cells} + 1\ pseudoread}{Normalized\ reads_{plasmid\ mix} + 1\ pseudoread}\right) \quad (2)$$

We added one pseudo-read to each activator's normalized count in both the plasmid and cell libraries to prevent zero-values from being scored as infinitely toxic.

The scores from the two biological screening replicates targeting *EPCAM* and the two biological screening replicates targeting *CXCR4* were averaged together to produce a final toxicity score. Because the cell populations used for the synthetic reporter screens were derived from either the *EPCAM*- or *CXCR4*-targeting transductions, these reads were excluded from the toxicity score calculation to avoid double-counting toxicity effects derived from the same cell population.

## Experimental validation of screen-derived activation

In order to experimentally validate our high-throughput screening data, all 25 single-domain library members, 20 clones randomly isolated from the bipartite library, and 27 clones randomly chosen from the set of tripartite library members whose activation was quantified for screens targeting all three target genes, were individually made into lentivirus and transduced in 24-well plates into HEK293T cells

stably expressing dCas9 and a gRNA targeting *EPCAM* or *CXCR4* in biological duplicate (bipartite and tripartite activators) or singlicate (single-domain activators). After 48 hours, cells were split 1:5 into 0.5 ug/mL puromycin. Activator-expressing cells were passaged 1:5 at least 3 times to ensure stable activator expression within the population. Cells were then harvested, washed with PBS, stained with EPCAM- or CXCR4-specific antibodies for 1 hour, and run on the flow cytometer to measure target protein expression. For bipartite and tripartite activators, a widget of either *EPCAM*- or *CXCR4*-targeting activator cell lines was transfected with 25 ng of our GFP reporter plasmid harboring an iRFP transfection marker along with 10 ng of reporter-targeting gRNA and harvested as above, followed by flow cytometry for iRFP+ cells to evaluate reporter expression. We calculated Spearman correlation coefficients between each activator's mean target fluorescence intensity and its screen score against the given target to evaluate the degree to which our screen accurately quantified activator strength.

## GFP competition experiment to validate screen-derived toxicity scores

In order to experimentally determine whether library member dropout from screened cell populations reflected activator-derived fitness defects, we mixed dCas9-expressing *CXCR4*-targeting cell lines expressing 27 randomly chosen tripartite activator clones (generated above) at a 1:1 ratio with a puromycin-resistant HEK293T cell line stably expressing dCas9, a gRNA targeting *CXCR4*, and GFP fused C-terminally to MCP in 24-well plates in biological duplicate. A construct containing a mutated MCP protein, in which the sixth amino acid in the protein, glutamine, is changed to a TAG stop codon, was included as a control. Mixes were cultured in 0.5 ug/mL puromycin to sustain MCP expression and passaged 1:8 every three days. At every passage, we harvested activator-GFP mixes as above and measured GFP expression using flow cytometry. Activator populations were quantified by calculating the percentage of GFP-negative cells in each mix. Activator populations were normalized to their baseline levels at each passage to evaluate overall changes in population size. After two passages, we calculated the Spearman correlation coefficients between each activator's average decrease in population size across two biological replicates and its average screen-produced toxicity score.

## Mini-library experiment to evaluate toxicity in different cell types

In order to determine the degree to which activator toxicity generalized to different cell types, we randomly isolated 56 tripartite library members and mixed them together in equal ratios. This mini-library was then made into lentivirus and transduced in T75 flasks into HEK293T, HeLa, and N2A cells stably expressing dCas9 and a gRNA targeting *CXCR4* in biological duplicate. After 48 hours, cells were split 1:5 into 0.5 ug/mL puromycin. Activator-expressing cells were passaged 1:5 a total of four times to ensure stable activator expression and potentiate manifestation of toxicity within the population. After cell harvesting and washing with PBS, genomic DNA was extracted by resuspending cell from each flask in 100 uL QuickExtract DNA Extraction Solution, followed by a 15-minute shaking incubation at 65 °C and a five-minute incubation at 95 °C. Activator barcodes were PCR-amplified from the gDNA of all bins, appended with Illumina adapters, and sequenced on an Illumina NovaSeq X Plus platform (Illumina, Cat. #20012850) with paired-end mode (2x150bp). Toxicity scores were calculated as described in the Analysis of screen-derived toxicity Methods and compared across cell types using Pearson correlation coefficient (*R*).

## Transfections to evaluate activator binding patterns

Twelve constructs exhibiting a range of activation and toxicity scores, along with an unfused MCP plasmid, were each appended

with a 3X FLAG-tag. Each FLAG-tagged construct was transfected into HEK293T cells expressing dCas9 and a *CXCR4*-targeting gRNA in quadruplicate (*n* = 4 biological replicates) using FuGENE HD in 100 mm dishes. After 48 hours, media was removed and cells were washed with PBS. Cells were then scraped into PBS, washed twice with cold PBS, and stored at -80 °C until further processing.

## Sample lysis and purification

Cell pellets were lysed with 650 uL of ice-cold IP lysis buffer containing 0.1% NP40, 0.1% Tween 20, 10% glycerol, 100 mM KCl, 5 mM MgCl2, 20 mM Tris-HCl pH 8, 1× protease, and phosphatase inhibitor cocktail tablets (cOmplete mini EDTA-free protease and PhosSTOP phosphatase inhibitor cocktails (Roche, Cat # 11836170001)) and sonicated before centrifugation at 13,000 × *g* for 30 minutes at 4 °C. 2 mg total protein lysate in 1 mL was used for affinity purification on a KingFisher Flex (KFF) Purification System (Thermo Scientific, Cat. #5400610) at 4 °C. Briefly, 20 uL of Pierce Anti-DYKDDDDK Magnetic Agarose beads (Thermo Scientific, Cat. #A36798) was incubated with 1.0 ml cell lysate for two hours and protein-bound beads were washed three times with 1.0 mL IP lysis Buffer and then once with 1.0 mL IP buffer (50 mM Tris-HCl, pH 7.4 at 4 °C, 150 mM NaCl, 1 mM EDTA) before eluting twice in 50 μL of 0.1 mg/ml 3xFLAG peptide, 0.05% RapiGest SF (Waters, Cat. #186001861) in IP Buffer for 30 minutes each at 23 °C.

## Mass spectrometry sample preparation, acquisition and analysis

Denaturation of 50 μL of FLAG-AP elution was performed at 37 °C for 30 minutes in 50 mM Tris-HCl pH 8.0, 2 M urea buffer complemented with 1 mM DTT. Samples were then alkylated at room temperature in the dark for 45 minutes by the addition of 3 mM iodoacetamide and quenched for 10 minutes with 3 mM DTT. Trypsin (1.5 μg; Promega, Cat. #VA9000) was added and incubated at 37 °C, overnight with shaking at 1,000 rpm. Peptides were acidified with TFA (0.5% final, pH < 2.0) and desalted at room temperature using a BioPureSPE Mini 96-well plate (20 mg PROTO 300 C18; The Nest Group) as per the manufacturer's instructions and dried under vacuum centrifugation (CentriVap Concentrator, Labconco).

For mass spectrometry (MS) acquisition, dried samples were resuspended in 50 uL of 0.1% formic acid before filtering through 0.45 uM filter and injection onto the Thermo Scientific Vanquish Neo HPLC platform on-line with a Thermo Exploris 480 Orbitrap Mass Spectrometer. Peptides were separated using a Bruker 15 cm long 150 uM ID PepSep column packed with 1.5 uM BEH particles, over a 45 minute gradient with mobile phase A composed of 0.1% formic acid in water and mobile phase B composed of 0.1% formic acid in 80% acetonitrile. The chromatographic gradient ran at a 600 nL/minute flow rate throughout. The gradient started at 4% B before increasing to 28% B over 30 minutes, followed by an increase to 45% B over five minutes, and finally finishing with a wash of 95% B for nine minutes. Full scans were collected at a resolution of 120,000, 350–1,250 m/z scan range with a normalized AGC target of 100% and the maximum injection time set to "Auto". Data-dependent scans were collected at a resolution of 15,000 with a normalized AGC target of 200% and maximum injection time set to "Auto". Precursors were selected for sequencing based on an allowed charge state of 2–6, and a dynamic exclusion after two sequencing events for 20 seconds of precursors within 10 ppm. The total cycle time of the full scan and all dependent (MS2) scans was one second.

Raw MS data were searched against the Uniprot canonical isoforms of the human proteome (downloaded 04 April 2023) using the default settings in MaxQuant (version 4.8.7), with a match-between-runs enabled (PMID: 19029910). Peptides and proteins were filtered to 1% FDR in MaxQuant, and identified proteins were then subjected to PPI scoring. Detailed MS acquisition and MaxQuant search parameters are provided in Supplementary Data 7.

To quantify the enriched interactions amongst HEK293T expressing C-terminally tagged activator constructs and unfused MCP, we used a label-free quantification approach in which statistical analysis was performed using MSstats (version 4.14.0)(PMID: 24794931) from within the artMS Bioconductor (version 1.18.0) R package. Protein spectral counts, as determined by MaxQuant search results, were used for PPI confidence scoring using SAINTexpress (version 3.6.3) (PMID: 24513533) (PMID: 24794931). High-confidence interactors were nominated as having a SAINTexpress BFDR ≤ 0.05.

R with the psych package (version 2.4.3) was used to generate correlation matrices. ChatGPT (GPT-4, June 2023 version) was used to assist in grouping proteins significantly enriched in toxic activators into broad functional categories (e.g., mitochondrial, cytoskeletal, chaperone). Protein names were provided as input, and functional characterizations were based on literature-derived annotations. These outputs were manually reviewed and curated.

## Statistics and reproducibility

Initial experiments tested 230 individual ADs to identify strong performers. From these, 25 domains were selected for combinatorial assembly into bipartite and tripartite constructs–balancing the goal of exploring a broad combinatorial space (25³ = 15,625 tripartite constructs) with the practical requirements of maintaining sufficient library representation in cell culture. Three targets (*EPCAM*, *CXCR4*, and a synthetic reporter) were chosen to gauge target specificity while keeping the screening effort tractable. Screening and manual validations were performed in duplicate (*n* = 2 independent biological replicates) due to the large scale and practical constraints of these experiments, and mass spectrometry experiments were performed in quadruplicate (*n* = 4 independent biological replicates).

We applied predefined filtering steps to exclude low-quality or spurious reads. Reads were discarded if they lacked expected sequence features, contained negative-control-associated barcodes, or did not meet minimum read count thresholds (see NGS data processing and filtering Methods). These exclusions ensured reliable quantification of construct performance.

To analyze the relationships between datasets, we employed either Pearson's correlation coefficient (*R*) or Spearman's rank correlation coefficient (*ρ*), depending on the nature of the comparison in question. Pearson's correlation coefficient (*R*), which quantifies the degree of linear correlation between variables, was employed for comparisons involving variables with assumed linear relationships (e.g., activation scores yielded by activators across two biological screening replicates). Spearman's rank correlation coefficient (*ρ*), a non-parametric measure that evaluates the monotonic relationship between two variables, was employed for comparisons involving variables with no assumed linear relationships (e.g., screen produced-activation scores and activator-elicited mean fluorescence intensities after target staining and flow cytometry).

No randomization or blinding was used. No statistical method was used to predetermine sample size. Experimental designs and replicate numbers were chosen to balance reproducibility with practical feasibility. Replicate experiments consistently produced concordant results, supporting the reproducibility of our findings.

## Analyses of AD order, fusion position, copy number, and dependence on adjacent ADs for activation and toxicity

To evaluate the effects of AD order on bipartite or tripartite activation and toxicity, we calculated the Pearson correlation coefficient (*R*) between the activation or toxicity scores of order-reversed pairs (e.g., AD1-AD2 and AD2-AD1). Only bipartite or tripartite constructs containing two or three ADs, respectively, were included in the analysis. To evaluate the effects of AD order on tripartite activators containing two ADs, only constructs containing exactly two ADs (and exactly one PF) were included in the analysis.

To evaluate the effect of AD fusion position on bipartite or tripartite activator activation and toxicity, we calculated the median activation or toxicity score of all constructs containing a given AD at the listed position, and compared these values across positions. We only evaluated the positional effects of the 22 ADs (and not the two PFs as these were considered to be inert) in these analyses.

To evaluate the effect of AD copy number on bipartite or tripartite activator activation and toxicity, we calculated the median activation or toxicity score of all constructs containing a given AD at the given copy number with either of two inert PFs (A23 or A25) in the remaining positions. For example, to measure the activation of one copy of AD1 in the bipartite screen, we calculated the median of AD1-A23, AD1-A25, A23-AD1, and A25-AD1 activation scores. We only evaluated the copy number effects of the 22 ADs (and not the two PFs) in this analysis.

To evaluate the dependence of AD activation and toxicity on surrounding ADs in bipartite or tripartite contexts, we calculated the median activation or toxicity score of all constructs containing a given AD in each screen (single-domain, bipartite, and tripartite), and calculated the Pearson correlation coefficient ($R$) between values across all screens. Both ADs and PFs were included in the analysis to highlight that median PF-produced activation scores are driven up upon pairing with AD partners.

## Evaluating activation of high-scoring MCP activators

Activators were ordered from the most potent (first) to the least potent (last) according to their average screen scores on each target gene. Any tripartite activator whose activity was not quantified for all three target genes was excluded from the analysis. Next, the rank of each activator across the three targets was summed. Finally, the 25 bipartite and tripartite constructs yielding the lowest summed ranks, presumed to be the most potent overall, were identified and rederived. These constructs were made into lentivirus and transduced in 24-well plates into dCas9-expressing HEK293T in biological duplicate cells followed by selection with 0.5 ug/mL puromycin.

In order to evaluate the strength of each activator, cell lines were transfected with 10 ng of a mix of gRNAs targeting four surface protein genes, *CD45*, *EPCAM*, *EGFR*, and *CXCR4*, along with 10 ng eBFP2 plasmid and 230 ng pBR322, followed by mixed target staining and flow cytometry in separate bipartite and tripartite groups.

In order to further confirm the relative strengths of top-performing bipartite and tripartite hits, the five bipartite and five tripartite activators which performed best upon the above manual testing were made into fresh lentivirus, used to generate dCas9-expressing HEK293T cells, and tested on the same four target genes, as above.

In order to evaluate the relative targeting strengths of MHV, MMH, SAM, and MCP-only, we created stable HEK293T cell lines expressing each construct, as above. Of note, the P65-HSF1 portion of the original SAM plasmid was first cloned into a plasmid equivalent to those of MHV and MMH to focus our comparisons on the activator portion of each tool and eliminate confounding MCP and plasmid backbone effects. While the lines created for our initial manual hit workup were engineered to stably express both dCas9 and the MCP activator, lines used for subsequent experiments, intended to compare activation produced by our recently identified activators to activation produced by SAM, were engineered to express the MCP activator only. This setup enabled facile dCas9 component swapping via transient transfection of either dCas9 (transfected into MHV, MMH, and MCP-only lines) or dCas9-VP64 (transfected into MPH line to create the full SAM complex).

Each MCP activator-expressing cell line was transfected with 100 ng of its dCas9 partner (MHV, MMH, and MCP-only: dCas9; MPH: dCas9-VP64 for SAM complex formation), and 10 ng of one of nine gRNAs—five of which target cell surface protein genes (*CD2*, *CD45*, *EPCAM*, *EGFR*, and *CXCR4*) and four of which target non-cell surface

protein genes (*TTN*, *HBG1*, *RHOXF2*, and *NEUROD1*)–along with 10 ng eBFP2 plasmid and 130 ng pBR322 for cell surface protein-targeting transfections, or 140 ng pBR322 only for non-cell surface protein-targeting transfections. After 48 hours, individual target staining and flow cytometry was carried out for cell surface protein targets and RNA extraction, cDNA synthesis, and qPCR was carried out for non-cell surface protein targets.

For multiplexed targeting, MHV, MMH, SAM, and MCP-only-expressing cell lines were transfected with 100 ng of their dCas9 partner and 10 ng of a mix of four gRNAs targeting surface protein genes *CD45*, *EPCAM*, *EGFR*, and *CXCR4*, along with 10 ng eBFP2 plasmid and 130 ng PBR322, followed by mixed target staining and flow cytometry.

In order to determine the activation potentials of MHV, MMH, SAM, and MCP-only in additional cell lines, constructs were made into lentivirus and transduced into HeLa, HCT116, and N2A cells followed by selection with 0.5 ug/mL puromycin. Each cell line was then individually transfected with 100 ng of its dCas9 partner and 10 ng each of a panel of gRNAs against either a human surface protein gene (*CD2*, *CD45*, *EPCAM*, *EGFR*, *CXCR4*), a human non-surface protein gene (*TTN*, *HBG1*, *RHOXF2*, *NEUROD1*, *MIAT*, *ACTC1*, *ASCL1*, *IL1RN*, *IL1B*, *ZFP42*, *LIN28A*, *IL1R2)* or a mouse non-cell surface protein gene (*Ttn*, *Hbb-bh1*, *Actc1*, *Neurog2*, *Ins2*, *Sim1*, *Mef2d*), along with 10 ng eBFP2 plasmid and 130 ng pBR322 for HeLa transfections, or 140 ng pBR322 only for HCT116 and N2A transfections. After 48 hours, individual target staining and flow cytometry or RNA extraction, cDNA synthesis, and qPCR was carried out for HeLa targets and RNA extraction, cDNA synthesis, and qPCR was carried out for HCT116 and N2A targets.

The DNA sequences of dCas9, dCas9-VP64, MHV, MMH, and MPH (the MCP component of SAM) are listed in Supplementary Data 8.

All plotted activation data were normalized to a negative control condition consisting of MCP not fused to any AD. This control condition was tested concurrently with all experimental conditions to account for background signal and ensure accurate quantification of activation potential.

## Testing off-targeting in top activators

In order to address the potential off-target transcriptional effects of MHV, MMH, SAM, and MCP-only, cell lines expressing each construct were transfected with 100 ng of their dCas9 partner and 10 ng of a *TTN*- or *HBG1*-targeting gRNA, along with 140 ng pBR322. HEK293T cells were transfected with the same components (dCas9 and not dCas9-VP64 was used) as a control. After 48 hours, RNA was extracted and cells were prepared for RNA-sequencing. Off-target effects produced by each activator were assessed by comparing the mRNA expression profiles of each MCP-activator-expressing cell line with that of the MCP-only control. Off-target activity produced by MCP-only was assessed by comparing the mRNA expression profiles of the MCP-only-expressing cell line with that of wildtype HEK293T cells. The differentially expressed genes between each comparison was analyzed by BioJupies using raw TPM as the input after filtering out the genes with median TPMs < 1 in all conditions. The genes with adjusted $P$ values < 0.01 and $\log_2$(fold change) > 2 or < -2 were considered as significantly differentially expressed genes.

## Cell line engineering

Cell lines expressing gRNAs targeting *EPCAM* or *CXCR4* (for dCas9-AD testing) were generated by lentivirally transducing an *EPCAM*- or *CXCR4*-targeting gRNAs into HEK293T cells in 24-well plates in biological duplicate followed by selection with 1 ug/mL puromycin.

Cell lines expressing dCas9 were generated by co-transfecting HEK293T cells with 500 ng dCas9 PiggyBac transposon vector and 100 ng PiggyBac transposase vector (Systems BioSciences, Cat. #PB351A-1) in 24-well plates in biological duplicate, followed by selection using 450 ug/mL nourseothricin.

Cell lines expressing dCas9 and MS2 gRNAs targeting *EPCAM* or *CXCR4* (for MCP activator library screening, manual MCP activator testing, GFP competition experiments, and mini-library experiments) were generated by lentivirally transducing three MS2 *EPCAM*- or three MS2 *CXCR4*-targeting gRNAs into dCas9-expressing HEK293T, Hela, HCT116, or N2A cells in 24-well plates in biological duplicate followed by selection with 10 ug/mL blasticidin.

Single-domain, bipartite, and tripartite clone-expressing lines (for experimentally validating screening data) and our MCP-GFP-expressing line (for our cell competition experiment intended to experimentally validate screen-produced toxicity scores) were generated by lentivirally transducing MCP constructs into HEK293T cells expressing dCas9 and an *EPCAM*-targeting gRNA (library members only) and dCas9 and a *CXCR4*-targeting gRNA (library members and MCP-GFP) in 24-well plates in biological duplicate followed by selection with 0.5 ug/mL puromycin.

Top hit- or MPH- expressing lines were generated by lentivirally transducing MCP constructs into dCas9-expressing HEK293T cells (for preliminary testing of 50 top hits) or unengineered HEK293T, HeLa, HCT116, or N2A cells (for final testing of two best hits and SAM) in 24-well plates in biological duplicate followed by selection with 0.5 ug/mL puromycin.

## Lentivirus production
Constructs were made into lentivirus by co-transfecting HEK293T cells with 450 ng transfer vector, 600 ng psPAX2 viral packaging vector, and 150 ng pMD2.G viral envelope vector in 6-well plates using Lipofectamine 3000 (Thermo Fisher, Cat. #L3000015). Lentiviral production was scaled up to increase virus production when necessary (e.g., bipartite and tripartite libraries). After 48 hours, virus-containing supernatant was collected, centrifuged at 1,500 x $g$ for five minutes, and re-aliquoted to remove cellular debris. For viruses requiring increased titer (bipartite and tripartite libraries), one volume of Lenti-X Concentrator (Takara, Cat. #631232) was mixed with three volumes of viral supernatant, incubated at 4 °C for two hours, and centrifuged for 45 minutes at 4 °C. The virus-containing pellet was then resuspended in 1/100th the original volume of Dulbecco's Modified Eagle Medium (DMEM, Thermo Fisher Cat. #11995065). All viruses were separated into 100 uL-aliquots and stored at -80 °C.

## Lentiviral transductions
For transductions of individual constructs, cells were transduced with lentivirus in 24-well plates (24 hours prior to transfection, 50,000 cells were seeded into each well). For transductions of activator libraries, cells were transduced with lentivirus in T75 flasks (24 hours prior to transfection, 2,000,000 cells were seeded into each flask). Cells were supplemented with lentivirus containing supernatant and split 1:5 or 1:8 into media containing a selection agent. Cells were passaged every 2–5 days. Lentiviral titers were determined by transducing cells with six different volumes of lentivirus and splitting transduced cells into drug-free and drug-containing media followed by assessment of the number of surviving cells in both conditions after complete selection.

## Mammalian cell culture
HEK293T cells, Hela cells (both gifts of P. Mali, UCSD, San Diego, CA), HCT116 cells (ATCC CCL-247), and N2A cells (ATCC CCL-131) were maintained in DMEM supplemented with 10% FBS (Thermo Fisher Cat. #A5670801) and 1 % penicillin-streptomycin (Thermo Fisher, Cat. #15140122) in an incubator set at 5% CO2 and 37 °C. Cell lines were verified for being free from mycoplasma at intake and yearly afterwards.

## Mammalian transfections
Transfections for individual activation experiments were carried out in 24-well plates (24 hours prior to transfection, 50,000 cells were seeded into each well). DNA was aliquoted into individual tubes prior to transfection. For single-domain dCas9-AD testing, carried out in stable gRNA-expressing HEK293T cells, 100 ng of dCas9 plasmid was transfected into each well. For manual testing of stably expressed MCP activators in dCas9-expressing HEK293T cells, 10 ng gRNA or total gRNA mix was transfected into each well. For manual testing of stably expressed MCP activators in non-dCas9-expressing HEK293T cells, 10 ng gRNA or total gRNA mix was transfected into each well along with 100 ng dCas9 component. For experiments utilizing flow cytometry as a readout, eBFP2 plasmid was co-transfected at 10 ng per well (subsequently, to remove untransfected cells from the analysis, target fluorescence was analyzed only in cells showing > $10^3$ EBFP2 expression). Plasmid pBR322, employed as an inert stuffer plasmid, was co-transfected in variable amounts to achieve sufficient DNA:transfection reagent ratios for optimal complexing when necessary and to ensure all conditions received the same total amount of DNA.

For high-throughput screening of activator library-expressing cells against our synthetic reporter, 2 ug of GFP reporter plasmid was transfected along with 800 ng reporter-targeting gRNA and 17.2 ug pBR322 in T75 flasks.

HEK293T and HCT116 cells were transfected with FuGENE HD or Lipofectamine 3000. HeLa and N2A cells were transfected with Lipofectamine 3000. For transfections using FuGENE HD in 24-well plates, individual aliquots of DNA were each mixed with 25 uL of Opti-MEM (Thermo Fisher, Cat. #31985062). A stock solution of 25 uL Opti-MEM and 1 uL FuGENE per ug DNA per transfection was made followed by inversion. 25 uL of the Opti-MEM/FuGENE solution was then added to each Opti-MEM/DNA mix, followed by inversion and centrifugation at 100 x $g$ for one minute. The mixtures were incubated for 15 minutes and added to cells. For transfections using Lipofectamine 3000 in 24-well plates, individual aliquots of DNA were each mixed with 25 uL of Opti-MEM followed by addition of 2 uL per ug DNA p3000. A stock solution of 25 uL Opti-MEM and 1 uL Lipofectamine 3000 per transfection was made followed by inversion. 25 uL of the Opti-MEM/Lipofectamine 3000 solution was then added to each Opti-MEM/DNA/p3000 mix, followed by inversion and centrifugation at 100 x $g$ for one minute. The mixtures were incubated for 15 minutes and added to cells. For larger-scale transfections, volumes were scaled up proportionately to cell numbers.

For all transfections, we calculated Pearson correlations between signals across biological duplicates to ensure high data quality.

## Antibody staining and flow cytometry
Flow cytometry experiments were performed using a BD LSR II instrument (BD Biosciences, Cat. #347545). Data acquisition and analysis were conducted with standard protocols. Forward scatter area (FSC-A) and side scatter area (SSC-A) were used to gate live cells, and single cells were gated based on side scatter height (SSC-H) and SSC-A. Transfected cells were identified based on their expression of BFP, detected in the Pacific Blue channel. Fluorescence intensity of the protein of interest was measured specifically in transfected cells, with mean or median fluorescence intensity (MFI) used as the quantitative readout. Supplementary Fig. 36 shows an example of our gating strategy.

The following antibodies were used. Each was diluted 1:100, and 100 μL of the diluted antibody solution was used to resuspend each cell pellet: anti-human EPCAM-PE (clone VU-1D9, Invitrogen, Cat. #A15782), anti-human CXCR4-PE (clone 12G5, BioLegend, Cat. #306506), anti-human CD2-APC/Cyanine7 (clone RPA-2.10, BioLegend, Cat. #300219), anti-human CD45-PerCP-Cy5.5 (clone QA17A19, BioLegend, Cat. #393419), anti-human EPCAM-FITC (clone VU-1D9, Invitrogen, Cat. #A15755), anti-human EGFR-Alexa Fluor 647 (clone AY13, BioLegend, Cat. #352918), and anti-human CXCR4-Alexa Fluor 594 (clone 44716, R&D Biotechne, Cat. #FAB172T-100UG).

For each antibody, the target protein was synthetically activated and stained, and the resulting signal was compared to a non-activated control to confirm antibody specificity and functionality. All antibodies were used according to the manufacturers' instructions.

## RNA extraction and qPCR analysis

Cells were harvested for RNA 48 hours post-transfection. RNA was extracted using the Direct-zol RNA Miniprep Kit (Zymo, Cat. #R2072). Synthesis of cDNA was carried out using PrimeScript RT Mastermix (Takara, Cat. #RR036A) using 500 ng RNA per reaction. All qPCR reactions were performed using the KAPA SYBR Fast Universal 2x quantitative PCR kit (Roche, Cat. #KK4651) using 1 µl of cDNA per reaction in a 20 µl total reaction volume. Gene expression was normalized to the expression of internal housekeeping gene *ACTB* for human cell lines and to the mouse ortholog *actb* for mouse cell lines. The following cycling conditions were used: 95 °C for 10 minutes and 40 cycles of 95 °C for 15 seconds and 60° for one minute (the plate read was taken after each cycle). All qPCR primer sequences are listed in the supplementary data. qPCR primer sequences can be found in Supplementary Table 4.

## RNA-seq library prep, sequencing, and analysis

Purified RNA was prepared for sequencing using the Stranded mRNA Prep kit (Illumina, Cat. #20040534). Libraries were sequenced on an Illumina NovaSeq X Plus with paired-end mode (2x150bp). RTA (Illumina) was used for base calling and bcl2fastq2 (version 2.20) for converting BCL to FASTQ format. Raw reads were then processed by Cutadapt (version 2.1) with the following parameters: "--minimum-length 24:24 -u 5 -U 5 -q 20 --max-n 0 --pair-filter=any" to remove low-quality bases and Illumina adapters. Pseudoalignment was performed against the index created from human transcriptomes (GRCh38) using Kallisto (version 0.44.0), and transcripts per million (TPM) was calculated to quantify gene expression levels. Genes were filtered to include only those with TPM > 1. The average of two biological replicates are shown. Differential expression analysis was conducted using BioJupies (RRID:SCR_016346) with raw TPM values as input after filtering for genes that produced TPMs of > 1 in at least one condition. Differential expression was defined by an adjusted $P$ value of < 0.01 and $\log_2$(fold-change) of > 2.

## Reporting summary

Further information on research design is available in the Nature Portfolio Reporting Summary linked to this article.

## Data availability

The next-generation sequencing datasets generated and analyzed in this study are available in the NCBI Sequence Read Archive (SRA) under BioProject accession number PRJNA1065930. Plasmids expressing MHV, MMH, MPH, dCas9 and dCas9-VP64 have been deposited to Addgene (https://www.addgene.org/browse/article/28247691/). The mass spectrometry proteomics data have been deposited to the ProteomeXchange Consortium via the PRIDE partner repository with the dataset identifier PXD068296. Source data are provided with this paper.

## Code availability

The code supporting the findings of this study is available under the MIT license, on Zenodo at (https://doi.org/10.5281/zenodo.10962794) (https://doi.org/10.5281/zenodo.10962793).

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

## Acknowledgements

We thank the Harris Wang, Sally Amundson, David Brenner and Kam Leong labs and other members of the Chavez lab for helpful discussions. We thank M. Staller for his insights on activation biology. We thank S. Resnick, P. Krishna, and B. Mayfield for technical assistance. We thank K. Beiswenger for project management expertise. Research reported in this publication was performed in the CCTI Flow Cytometry Core, supported in part by the Office of the Director, National Institutes of Health under awards S10OD020056. The content is solely the responsibility of the authors and does not necessarily represent the official views of the National Institutes of Health. This research was funded in part through the NIH/NCI Cancer Center Support Grant P30CA013696 and used the Genomics and High Throughput Screening Shared Resource. Figure graphics were created using BioRender.com. This work was supported by the Defense Advanced Research Projects Agency (HR0011-19-2-0009). H.H.W. acknowledges additional relevant funding from NIH (1R01EB031935). A.C. was supported by the NIH Grant DP2NS131566-01 and a Career Award for Medical Scientists from the Burroughs Wellcome Foundation. N.K. was supported by the NIH Grant U54 CA274502. A.C.

has a series of CRISPR technology-related patents that are managed by Harvard and Columbia University, and the authors have submitted a provisional patent related to this work.

## Author contributions

M.G., A.K., L.W. and A.C. conceived of the study and designed experiments. M.G., A.K., L.W., M.D.L.S., C.Q., Y.H.L. and T.F. performed experiments. M.G., A.K. and L.W. analyzed data. T.B. computationally mined for putative ADs. A.F., R.T. and G.J. performed AP-MS experiments and related analysis. Y.H. performed RNAseq library loading and analysis. M.S. interpreted data related to activator biology. H.W., N.K., and K.W.L. provided project guidance and secured project funding. A.C. and L.W. supervised the study. M.G., A.K. and A.C. wrote the manuscript with support from L.W. and other authors.

## Competing interests

Columbia University has filed a provisional patent application with the United States Patent and Trademark Office pertaining to the CRISPR activators MHV and MMH described in this work (U.S. Provisional Application No. 63/892,477). The remaining authors declare no competing interests.
