## [Transparent Peer Review file · Nature Communications]

Combinatorial Protein Engineering Identifies Potent CRISPR Activators with Reduced Toxicity

Corresponding Author: Dr Alejandro Chavez

Version 0:

Reviewer comments:

Reviewer #1

(Remarks to the Author)

In the present manuscript, Giddins et al. have developed an efficient approach for creating and evaluating complex combinatorial landscapes of multi-domain CRISPR activators, uncovered the principles of combinatorial AD assembly, and identified an improved activator. The authors first individually tested a set of 230 ADs at three targets, and uncovered several highly potent domains. The authors further employed a high-throughput combinatorial approach for 25-high value ADs to test 625 bipartite and 15,625 tripartite CRISPR activator variants at a time. Based on screening results, the authors found that two selected promising activators showed enhanced activity against CRISPR-SAM across genome targets and cell lines, and untested off-targets at TTN gene. In addition, the authors explored the biochemical features of activation domains, determined how their combinatorial interactions shape activator behavior, and found that many activators produce substantial cellular toxicity independent of their ability to regulate gene expression. Overall, the topic is interesting, and this study contributes through providing researchers with increased options for using CRISPR activator in their research. However, some data are robust and not enough. Here are some comments and suggestions for the authors to improve the quality of the work.

1. The authors state that two potent activators MHV and MMH outperform SAM in multiple contexts. However, in Fig.6e-g and Fig.S25, MHV activity at these 9 targets was higher at some sites, equal at some sites, and lower at some sites compared with SAM in HEK293T. Similar situation is showed in N2A and Hela cells. Similarly, at most of targets, MMH showed comparable activity to SAM. To demonstrate the improved activity of both MHV and MMH, data are not far enough and more targets in those cell lines are needed. The negative control is also needed.

2. For off-target gene regulation, should add at least one more target to perform RNA-seq analysis of MHV, MMH and SAM.

3. According to current verified data and comparison data to SAM, among vast of activators variants, A22_A08 MHV and A03_A22 (MMH), showing the top high activity among these thousands of ADs, showed enhanced or comparable activity to SAM. This high-throughput work looks very cool, but for this combination designs results out only 1-2 variants with high activity better than SAM. The cost performance is too low, what is the defect of the current screening, or is there a more optimized design scheme. Please discussion this.

4. The authors selected two promising activators for further workup. What is the principle of selection. It's obvious not random. But if choose based on the activity, the bipartite A16_A22 is higher activity than A03_A22 (MMH) in EPCAM, CD45, EGFP and CXCR4 and not chosen in Fig. 6a-b and Fig.S23. Please explain the considerations.

5. In addition to activity advantages, the authors could describe the other advantages of both MHV and MMH, such as size.

6. Among these CRISPR activator variants, whether the SAM combination is present, and what about the level of activity at three screening targets.

7. The authors found that increases in AD copy number do not enhance toxicity suggest that the molecular mechanism driving this property rapidly saturates. But why the increased toxicity of 3 copies than that of 1 or 2 copies, in Fig. 4f.

8. Line400, the authors state the MHV and MMH yielded relatively low cell toxicity. How to define the standard of cell toxicity. In this cellular toxicity evaluation system, what is the level of SAM?

9. In Fig.6e-g and Fig.S25, compared to SAM, MHV displayed the opposite activity at CXCR4 target from single site and four multiplexed targets. Would you be able to explain why this is the case?

10. In general, the biological replicates are 3 or over. Thus, it is recommended that for activity comparison of among MHV, MMH and SAM, three biological replicates are set.

(Remarks on code availability)

Reviewer #2

(Remarks to the Author)

Giddins and Kratz et al. describe a CRISPRa protein variant library platform for identification and optimization of proteins that can activate gene expression. The authors have some nice insights in the manuscript such as that activator protein activity does not correlate with toxicity and that indiscriminate combinations of domains is unlikely to result in better tools. A major goal of this manuscript is identification of a new best in class gene activator but problematically it is not clear that the MHV activator nominated by their screens is meaningfully better than the SAM system across cell types.

Figure 1 and associated supplemental figures.

Initially the authors screen 230 domains in an arrayed format using transient transfection to identify domains that modulate activation of 2 endogenous genes and a reporter. Please clarify whether Supplementary Figure 2A shows all 230 domains? Their activation results are very much dominated by VP16 variants which are already very well established as strong activators and thus this screen doesn't open up much new biology or tools space.

The authors should graphically present show how many unique domains are present in the library vs variant and homologs.

Figure 2 and associated supplemental figures.

The authors examine properties of hits from Figure 1 and arrive at the conclusion that acidic domains tend to be activators and intermixing acidic and hydrophobic domains tends to promote activation. Both observations support known features of activation domains and thus this is not particularly novel nor do these results provide insight into how to better design tools.

Please clarify for Figure 2F – were ADs predicted with PADDLE and then plotted across the length of the protein? Or were these minimal 80AA tiles identified as ADs from prior literature and then plotted across the length of each protein?

Figure 3 and associated supplemental figures.

The authors combine 25 domains into all pairwise and three way combinations in a pooled lentiviral library and then screen these combinations as activators across 2 endogenous genes and a reporter to identify proteins that potently activate gene expression.

Please clarify for Figure 3E and Supplementary Figure 6 which of the 20-30 proteins tested in validation experiments would have been called hits in the screening data.

It is known that lentiviral recombination during reverse transcription is a problem for using barcodes to track elements within a library. Please quantify via long read NGS how much recombination occurs for constructs with this domain/barcode library design after construct integration by lenti into the genome. Intra- and inter- molecular recombination could result in deletions or in scrambling of the domains or in decoupling of barcodes from their assigned domains. This is noted by the authors in Figure 4F as a problem and so should be quantified to enable others to use such a platform approach in a robust manner.

Figure 4 and associated supplemental figures.

The authors note that for the tripartite library many library members did not meet the read thresholds required for to accurately quantify their function. Please state clearly what fraction of the single domain, bipartite and tripartite library meets the read threshold in the main text. The authors comment that the distribution of constructs present in plasmid DNA is poorly correlated with the distribution in cells and attribute this to toxicity. If the authors believe this to be true they should state clearly what fractions of the single domain, bipartite and tripartite library are toxic in the main text.

It is noted that disordered domains have increased toxicity but the correlation isn't extremely strong leaving it a bit unclear what drives toxicity. It is known that activators can be highly toxic and so although this data is useful it doesn't provide enough mechanism to design tools that avoid toxicity. Basically the authors are saying everything needs to be tested empirically for activity and toxicity. Is there a more constructive message that can be delivered or do the authors plan to make this library cloning strategy very easy to use for the community if empirical testing is always required?

What time point post infection is displayed in Figure 4C?

For Supplemental Figure 10 the authors should determine experimentally whether these constructs toxicity is dCas9 dependent or independent by expressing the MCP fusion proteins in the absence of dCas9 and then tracking cell populations over time. The toxicity of such a dCas9/MCP-fusion/sgRNA-MS2 system can relate to protein expression ratios for dCas9/MCP. Knowing if the toxicity is associated with the DNA binding protein is highly useful and would help interpret their claims in Supplemental Figure 12 around squelching.

Figure 5 and associated supplemental figures.

In this figure the authors show that activation domains effects are context specific in the tripartite but not bipartite context- from a biological perspective why would this be true? They support this by analyzing the tripartite constructs that have a stabilization domain and 2 activation domains to show in that context the 2 AD domains in the tripartite context behave more

like the bipartite proteins. One simple explanation could be that the tripartite domains are not stable proteins. The authors could test this by measuring protein levels for some number of bifunctional and trifunctional proteins (with 3 activation domains or 2 activation domains and a stabilization domains) to determine the effect of protein levels on activity.

Figure 6 and associated supplemental figures.

A major goal of this effort is to identify better gene activators for the research community.

As such the authors choose 25 bipartite and tripartite hits from their screens and then validate these in an arrayed format. The authors claim the bipartite hits generally perform comparably to the tripartite but in fact for EGFR and CXCR4 the tripartite hits perform substantially more poorly than the bipartite hits on average (Supp Figure 23). Please state this in the text.

Combined analysis of Figure 6C&D and Supplemental Figure 25 shows that MHV more robustly activates gene expression of 6/9 genes relative to SAM in HEK293s (4/6 differences are substantial and the other 2/6 are significant but not substantial). The authors then compare MHV and SAM in 2 other cell lines and claim that MHV is better than SAM for 3/6 genes however of the 3/6 genes where MHV is statistically better than SAM in N2A or Hela cells only 1 gene (TTN) is substantial whereas the other two genes show statistically significant but similar levels of activation (e.g. for CD2 and EPCAM in Hela all 3 constructs activate expression by 2-3 fold and it is not clear the difference between 2 vs 3 fold is meaningful biologically). So basically the entire claim that MHV is substantially better than SAM rests on HEK293 data or on a single gene activation measurement in one other cell type. It would be very problematic to nominate a "best in class activator" that in fact is really a "best in class HEK293 cell type specific activator" as this will waste a substantial amount of other lab's time and not fulfill the goal of identifying a new best in class gene activation construct. Please show MHV is better than SAM in other cell types more convincingly.

It seems like the fold changes for TTN as measure by qpcr and rna-seq in 6D versus 6H are not concordant. If the qpcr readouts are not trustworthy then this calls key data in figure 6 and supplemental figure 25 into question. Please comment on this?

(Remarks on code availability)

Reviewer #3

(Remarks to the Author)

The authors set out to establish a CRISPR-based platform that enables high-throughput exploration of combinatorial activator constructs, leveraging a modified CombiSEAL system to bypass individual synthesis of activator domain (AD). Additionally, the authors make an effort to characterize activator potency on endogenous genes as opposed to exclusively a synthetic reporter, addressing the shortcoming of generality of current state-of-the-art CRISPR activators across target contexts. They identify trends in the features of strong activators and in how ADs can be composed in combinations. The authors develop and characterize two potent activators, MHV (HSF1-VIRF2) and MMH (MLLx3-HSF1), which outperform the CRISPR activator SAM across several sgRNAs targeting endogenous genes in a handful of cell types (mouse and human) in terms of activation potency. Importantly, the authors claim that these new activators display reduced cytotoxicity compared to SAM. They characterize activator cytotoxicity in multipartite constructs, and show trends with disorder and hydrophobicity. Further, the authors find multipartite AD cytotoxicity approximates to the sum of individual AD toxicities, irrespective of AD order in the overall construct. This contrasts with their somewhat surprising finding that activator potency is highly dependent on the order of ADs within their tripartite (but not bipartite) constructs.

I feel the largest impact will be from the MHV tool, which looks stronger than SAM and more limited data suggests is also less toxic. They also set the laudable goal of uncovering rules for composing ADs, but I feel their success here is somewhat less confirmed. They observed several relevant trends, but it is not proven that these can now be taken as 'rules' to guide, for example, the more successful design of the next library of ADs. For example, it seems that MHV is selected because of its performance in the screen, not because they applied new knowledge learned about AD composition principles. They don't seem to comment on whether MHV is representative of the principles they learned. I appreciate the way they highlight the importance of toxicity, although they do not mine their data in a way that explains the toxicity phenomenon further (perhaps their biggest impact on the toxicity question is they push back on the theory that squelching is the driver).

Main comments

1. The figures are neat and often have very helpful clear schematics. However, they mostly show overall trends and it is difficult to learn much specifically for the activator domains because of the anonymized way they present the data. For example, in Figure 3-6 they usually label genes with a code ID instead of a more easily interpretable gene name. I understand the names of triple combos can get long, but I urge the authors to consider ways to make more information-rich visualizations of their datasets. For example, very few dots in the plots are labeled with gene names. It is not clear where negative controls (e.g. dCas9- or MCP-only) or positive/well-known activators (e.g. VP16, VP64, p65, HSF1) fall in the plots. Given the importance of VIRF2 and MLL in the final figure, they could also label them in more of their plots.
2. Why remove the directly fused VP64, as in SAM, when making the MHV/MMH constructs? Could they outperform SAM even more often if they also had the VP64 component? This could be tested, or the reasoning better explained.
3. I think the investigation of toxicity is a strength of the paper, although ideally there would be firmer conclusions about toxicity. If domain toxicity "remains constant independent of surrounding context", wouldn't the slopes in D-E be along the $y=x$ line? Could you show replicates of the toxicity score? Further, is there bottlenecking in the screen that causes outliers, and is this handled by filtering (and/or the use of UMIs)? I ask because bottlenecking could be suggested by some elements with many reads in one unsorted cell replicate and 0 in the other replicate. The authors mention that toxicity has been

reported for previous activators such as VPR and SAM; where do those fall in these plots, if they (or related fusions) are present (e.g. Figure 4D)? Several results show toxicity and activation are not correlated; which is interesting and not fully explained here. Also, given how they compare to the plasmid library, it may not be possible for them to assess if “toxicity” occurs during cell library growth or during lentiviral packaging, but this would be of interest and could affect how they use the term “toxicity”. The pooled measurement of toxicity doesn’t correspond perfectly with the individual measurements in Supp Fig 10C ($\rho=0.58$); is this the explanation or is there another reason? Would the authors expect the AD to also be expressed during lentiviral packaging, such that toxicity in the packaging cell could also impact viral titer?

4. Does the C-terminal trend only hold for dCas9-AD fusions where the AD is displayed on the C-terminal end? Does the original position of the AD also correlate with how well it works in the context of the MCP fusions; e.g. do ADs from the middle of the protein work better in position 1 in the multipartite fusions?

5. Figure 5B – order has a major impact on tripartite but not bipartite activators, can this surprising result be further explained? The authors suggest activation requires a precise spatial configuration, but how is that compatible with the results from the bipartite elements? Perhaps relatedly, how long is the linker between ADs? Is it possible that the tripartite behave differently than the bipartite domains (e.g. much greater effects of domain order) due to lentiviral recombination between repetitive linkers/domains?

Minor comments

1. They could better demonstrate the impact of including UMIs, e.g. showing the data with or without UMI aware analysis.
2. Supp Fig 24 has a floating “e” near “Off target”. I also think the schematic showing what MHV and MMH look like is very useful and could be moved into a main figure.
3. Is it a stretch to call SAM “the gold standard CRISPR activator” (as in the Abstract), given the diversity of CRISPRa’s that are optimized for use in different contexts? Although it was highlighted in at least one published “bake off”, SAM is still not standardly used across all active laboratories in this field, in my experience. At one point in the manuscript the authors call SAM “the gold standard MCP activator” (emphasis my own), which seems more accurate to me.
4. The key new component of MHV, VIRF2, was also identified in a recent preprint (Carosso et al., BioRxiv) that screened for activator domains to attempt to improve CRISPRa. I imagine this is an instance of independent co-discovery. Also, Ludwig et al, Cell Systems, 2023 reports a functional characterization of VIRF2 effectors and finds that overexpression of VIRF2 causes transcriptional changes in some ISGs. The present authors also cite that VIRF2 modulates human immune genes. Do these results conflict with the seemingly unperturbed RNA-seq upon MHV expression?
5. Figure 1 legend could be more clear how the fold-change is calculated (MFI or % positive cells between two samples?). Is the normalization to dCas9-only negative control plasmid an additional step?
6. Figure 1F – the term “specialist” sounds strong to me given the definition of being within the top 10. E.g. if an activator were top 10 in two screens, but ranked #11 in the third screen, it would seem incorrect to call it a “dual-target specialist”.
7. Could the authors show how single domain lentiviral/MCP screen results correlate with the transfection/direct-fusion screen?
8. For the Figure 3B schematic- Is there a 2A peptide between PuroR and MCP?
9. I don’t think Figure 3 should be labeled “rapid” screening, as there is no information presented in the figure about the amount of time it takes to do this method. It appears to require the standard amount of time for such a combinatorial lentiviral FACS-based screen.
10. Does the 4-bin analysis add useful information beyond a 2-bin (1 vs 4, or 1+2 vs 3+4) analysis? It could be very useful to readers if the authors can comment further on their binning strategy. Perhaps relatedly, I wonder if their library construction strategy makes the singles and doubles more highly abundant than the numerous triples, and that the triples then have lower cell coverage during the screen and readout, and this explains the higher noise in the measurements of triples. As such, it seems plausible that combining Bins1+2 and 3+4 to improve cell coverage would overall improve data quality, despite lowering bin resolution.
11. Figure 6H – Could TTN be labeled in Supp Fig 26A to show if CRISPRa is working? It just shows replicates, with no comparison between conditions or labeling of on/off-targets. Would MHV vs negative control be more informative about the transcriptional perturbations of overexpressing the MHV (which do sound minimal)?
12. Is there a mistake in the title of Supp Figure 17 that gets the positive and negative reversed, relative to the plots?
13. I believe PADDLE was exclusively trained on yeast data, which may be worth mentioning when discussing how it performs on this dataset. Could they present a plot of PADDLE prediction vs measurement too?
14. Could the authors expand on the ‘built in screening controls’ from Note 1 more? They seem interesting and useful to readers. How do supernatant controls make it all the way through the cell sorting step into the final libraries? I am not familiar with supernatant controls; are they described elsewhere? Would the authors suggest not trusting screen data with a certain score from these controls?

(Remarks on code availability)

Reviewer #4

(Remarks to the Author)

(Remarks on code availability)

Version 1:

Reviewer comments:

Reviewer #1

(Remarks to the Author)

The authors have reasonably addressed my comments.

Minor :

In the Supplementary Figures 32, 33, and 34, up/down-regulated genes should be labeled as distinguishing points, including HBG1.

Referring to the recent Nature Methods-published tool CRISPR DREAM (Mahuta et al. 2023), the author used its data to justify the current analysis. But I haven't seen similar data, such as TBX5 et al. Please indicate which figure in CRISPR DREAM these data came from. Also, it'd be better if the author could make a statistical graph to clearly show the activity comparison with SAM.

(Remarks on code availability)

Reviewer #2

(Remarks to the Author)

The authors have responded with experiments, analysis and text changes to my review. My concerns related to lentiviral recombination are fully satisfied which was a major concern. My concern related to the explanation of validation data in Figure 3E remains a concern although this can be addressed as noted below. The expansion of text and data on toxicity of activators as well as AD order and context are strengths of the manuscript and will be highly useful.

I remain unconvinced that MHV and MMH (or DREAM) are a sufficiently big advance (especially across cell types as further discussed below) to convince the community to move away from SAM, VPR or other more widely used activator constructs. Specifically, MMH is similar or sometimes worse than SAM. MHV is overall not significantly more active in N2A or HCT116 across comparisons. MHV is more active than SAM in HEK293 and Hela at about half of tested examples (many of which may not be biologically meaningful differences) so there remains unexplained activator cell type specificity and overall MHV is more active than SAM at only 19 out of 48 comparisons across cell types. If the authors really want to show MHV is better than SAM they should do a genome-scale screen and show they learn something new with MHV that is missed by the SAM approach. However that said it is interesting that such a large effort has not yielded an activator that is remarkably better than SAM.

Related to Figure 1:

Please show a histogram of domain amino acid length for the 230 domain library to illustrate how the authors library would compare against more the more common approach of using 80AA tiles. Please color also

Related to Figure 3:

In Figure 3D the authors now highlight that the MCP components of the SAM system (MCP-P65-HSF1) are a very top hit in the screen (top 2) which undermines the notion that this manuscript's discovery effort will lead to new useful tools.

Figure 3E as commented on in my first round of comments it remains confusing that the authors picked random hits from the screens. The authors state "To experimentally validate our screening results, we isolated 20-30 library members from each of our single-domain, bipartite, and tripartite libraries and tested them independently via transduction against our screening targets." Perhaps stating "To experimentally validate our screening results, we randomly isolated 20-30 library members from each of our single-domain, bipartite, and tripartite libraries only a subset of which are strong hits and tested them independently via transduction against our screening targets." Please clarify.

In my second round of comments I reviewed the manuscript without first reading the response to review and wrote the whole comment below on Figure 3E so this section is clearly confusing.

In Figure 3E the authors validate a number of hit constructs from their screens. The authors show a fairly good correlation between screen and validation (0.54-0.83) but closer inspection of the data shows that many of the hits are inactive in the validation experiments. This is especially true for the bipartite hit validation with a stripe of hits at zero on the x-axis. Please plot the x-axis as \log_2 fold change over a negative control construct for all validation data instead of or in addition to the current plots (Figure 3E and Supplementary Figure 7). In Figure 1 the authors define hits as producing "at least two-fold activation on EPCAM, CXCR4, and the synthetic reporter, respectively". It seems to me that any bipartite or tripartite construct that doesn't meet this criteria is a false positive hit from the screen despite the apparent correlations. Please define your false positive rate given these issues raised.

As you can see I remained confused. Perhaps noting with different colors which tested constructs would have been called

hits will clarify this figure?

Related to Figure 6:

Did the authors include a dCas9 alone, MCP alone or dCas9 + MCP controls or any control here to help with interpretation of which proteins are binding to dCas9 + MCP + linkers and which are binding to various activation domains? I ask because if the vast majority of these human proteins were to bind to dCas9 + MCP + linker protein interfaces then this would cause the results to look more correlated than expected across different activator fusion proteins. If the authors don't have such controls this experiment is very hard to interpret especially as protein-protein interactions can stabilize or destabilize fusion proteins. Please include such controls.

Do the results in Figure 6 relate to this experiment being a transient transfection assay? Transient transfection assays result in vastly higher protein levels than are generally relevant to cellular proteins with natural activation domains.

I would recommend that the authors validate their mass-spec data with co-IP western data for constructs expressed via transient transfection and lentiviral expression so the community has a sense of validation and some more constructive context for this toxicity mass-spec analysis.

What proteins do SAM, MHV and MMH bind? I would add a figure and additional discussion on this to the manuscript. It will scare the community to highlight co-factor binding toxicity without discussing this in detail for these key highlighted constructs. Validating that SAM which is widely used and also MHV and MMH does or does not bind to proteins that are associated with toxicity is a really key part of this new data section.

How many of the proteins enriched in this mass-spec analysis are canonically localized within an lipid membrane enclosed organelle? I ask because it is surprising to see such an enrichment of mitochondrial proteins. If proteins that are localized inside the ER, golgi or mitochondria are enriched in these mass spec data it would argue that some fraction of these results are artifacts. Understanding this is important for interpretation of the authors's claims. I understand that some of the proteins identified in the mass-spec have regions of the protein exposed to the cytosol and thus could be reasonably expected to interact with activation domains localized to the cytoplasm/nucleus.

Related to Figure 7

I believe the authors have a mistake in the text related to Figure 7c-d and Supplemental Figure 28. They state MHV is significantly more active than SAM for 11/17 genes but it is actually 10/17 as 6 genes show no significant difference between MHV and SAM and 1 gene (CXCR4) shows SAM is more active.

In Supplemental Figure 29, for ITGAV it appears each of the activators is actually a repressor with around 50% knockdown? Please clarify. If all constructs tested are repressive this should be noted or removed from the claims that call MHV as more active than SAM as an activator so that comparison would be 9 out of 31 rather than 10 out of 32.

Related to Figure 7 and Supplementary Figures 28-31: I believe a more accurate way to describe the data is MHV and SAM are roughly equivalently active in N2A cells (MHV is more active in 1 out of 7 comparisons once ITGAV is excluded) and HCT116 cells (MHV is more active in 2 out of 12 comparisons) while MHV is somewhat more active than SAM in HEK293 (10 of 17 comparisons) and HeLa cells (6 out of 12 comparisons) which highlights that there is a cell type specific aspect of activators that is not fully explored here. This could be a future direction commented on in the discussion.

Overall I would temper expectations for MHV and MMH. For MHV which is the more active construct the authors show that only 19 out of 48 comparisons show MHV is somewhat more active than SAM across genes and cell types.

(Remarks on code availability)

Reviewer #3

(Remarks to the Author)

I commend the authors on a thorough revision, wherein they performed further analysis of domain order, performed a new mini screen that shows toxicity is similar across cell types, showed MHV>SAM across more target genes, analyzed PADDLE scores, clarified figure labels, better explained UMIs and spike in controls, and made other improvements. Most notably, the new AP-MS characterization of protein binders of their activator combos adds to the impact of the paper, showing clearly how domain order does not majorly affect AP-MS interactors.

I expect many readers will be especially interested in MHV as a useful new activator that sometimes outperforms SAM and is rarely worse (while dropping the VP64 component). The authors did a good job of thoroughly demonstrating this across several contexts, while not over-promising on its performance. I think MHV's advantage is sufficiently consistent, and the need for better activators so pressing, that many readers will want to try this tool. By one measure, it seems MHV performs

better than SAM at ~43% of targets and similarly at most other targets. MMH is almost always equivalent to SAM, but is notable because it is all human activators, unlike SAM which includes viral components. Further studies will be needed to assess how the set of recently published next generation activators (e.g. DREAM, NFZ, MHV) perform head-to-head and whether certain ones are better used in certain biological, protein fusion, or delivery contexts.

Major comment:

The authors responded convincingly to many of the questions about toxicity. The AP-MS Figure 6 is entirely new and raises some questions. Could they better show what are the top interactions for these activators (e.g. labeling dots in D)? It is unclear if they pulled down any co-activators (e.g. p300, Med). It is definitely of interest to the field to see which co-activators are recruited by which activator, including the new MHV. Perhaps co-activators could be included in C,D,E. If no co-activators are enriched, that would suggest these datasets are incomplete, and should be more caveated in the writing, especially given the lack of validation experiments.

In E there are some proteins that are more enriched with the toxic activators – is that due to more binding or more expression of those proteins in the cells experiencing toxicity? Why does E not include any interactors that are more specific to the non-toxic activators? (e.g. wouldn't it change interpretations if more protein folding pathway interactors are actually higher in the non-toxic activators?) Is it valid to use ChatGPT to group interactors into functional categories (vaguely described in Methods), as opposed to supervised and reproducible gene ontology/gene set enrichment methods? Are those functional categories in E statistically enriched among toxic AD interactors vs non toxic AD interactors?

Minor comments, which are not necessary to resolve for publication in my view:

1. I valued the QC of lentiviral recombination with their triple AD combos, and would have liked to see that data added to the paper, as I think it would be useful to see for others working on related methods. Hopefully it will at least be included in an available peer review file.
2. It was helpful to see RNA-seq measurement of off-targets with appropriate controls and two target genes, but I wondered if more information could be retrieved from this data. The RNA-seq plots supp 33 do seem to show some off targets, and the writing mentions up to 5 genes that are significantly upregulated. What are these genes, and are they related to the sgRNA (e.g. have a mismatched sgRNA binding site near the TSS) or not? Are they induced by expression of MHV/SAM regardless of target gene? Of course, overall the tools look quite specific, so this is not a major problem, but it could be useful information.
3. The text does a better job than Fig 7 of accurately portraying how often MHV outperforms SAM. It may be helpful to add a panel to 7 that summarizes the many barplots that are currently left in the supplement, perhaps a heatmap or a count of the frequency with which each activator is best.
4. Double check author names – is it Max Staller?
5. The “floating e” issue is still there in Supp Fig 27. Also, the Off-targeting panel seems to show “idealized data” of what the RNA-seq could look like, but with stronger activation of the targeted gene than what was actually achieved in the real data in Figure 7 and supp 32-34. This scatter plot depiction could instead just be a clearly cartoonized schematic of RNA-seq.

(Remarks on code availability)

n/a

Reviewer #4

(Remarks to the Author)

(Remarks on code availability)

Version 2:

Reviewer comments:

Reviewer #2

(Remarks to the Author)

The authors have responded to most of my concerns with revised figures, text, analysis or new data.

I remain concerned by the observation that mass-spec results show “a high correlation in binding partner enrichment, even among those activators composed of entirely different domains” (e.g. Figure 6C: VP16x2-BEL1 and VIRF2-ICP411-MLLx3 = 0.89). How do the authors explain this result given the data is normalized to an MCP control?

Is there some structural or domain similarity driving this for specific domains? Or perhaps it is more likely the explanation could come from the features of their MCP library construct design such as mislocalization or aggregation for toxic MCP fusions? The authors never examine the degree of P2A skipping in their Puro-P2A-NLS-MCP library design. It is known that P2A skipping can be context specific (PMID:). Additionally they do not examine the degree of nuclear localization for a representative toxic and non-toxic MCP activator fusion. It is known that NLS biology is also context specific. I realize the

variable domains of each protein (the AD domains are on the C terminus but is it possible that much of this toxicity phenotype seen for the library could be due to fusion proteins that are Puro-P2A-NLS-MCP fusions and/or cytosolic mislocalized proteins? Or is it possible the toxic proteins are aggregating despite the inclusion of PF domains in some constructs? Western blotting to examine protein size could address P2A skipping efficiency. Immunofluorescence or cell fractionation followed by western blotting could address mislocalization concerns. This concern is re-enforced by my prior concern that it is quite surprising that the authors see so many mitochondrial, cytoskeletal and vesicular trafficking proteins associated with MCP fusions in their mass-spec results.

(Remarks on code availability)

Reviewer #3

(Remarks to the Author)

1. I appreciate the new summary figure and would only suggest changing the color scheme to be a bit more colorblind friendly (e.g. red, gray, blue).
 2. I note they chose not to revise the AP-MS section, but did provide more nuanced interpretations in the rebuttal to both reviews. I think more future work is needed to firmly establish what causes CRISPRa toxicity (including if toxicity requires an AD interaction with any of these proteins).
 3. I appreciate their point about clarifying for the field where the baseline for activators is and their setting expectations for where to find future gains.
- I suggest this revised manuscript should be accepted for publication and commend the authors on this work.

(Remarks on code availability)

Reviewer #4

(Remarks to the Author)

(Remarks on code availability)

Version 3:

Reviewer comments:

Reviewer #2

(Remarks to the Author)

The authors have responded to my comments on AP-MS results for toxic constructs that have highly correlated pull down profiles without common MCP fusion domains with text and a slight modification to the manuscript. The authors now state this toxicity mechanism issue is beyond the scope of the current manuscript in the text of the manuscript. I think it is reasonable for this manuscript to leave open mechanistic questions around toxicity but the results do remain somewhat confusing to me which is still a bit problematic as a big message of the paper is around avoiding toxicity.

With respect to P2A/T2A biology. There is quite a bit known about this for example:
<https://www.nature.com/articles/s41598-017-02460-2>

(Remarks on code availability)

Color coding for this document: Reviewer comments in **BLACK**, our responses in **BLUE**, words and edits that are in the manuscript are in **PURPLE**. Of note, some responses were refined for clarity with the aid of ChatGPT-4o.

We extend our gratitude to the reviewers for their insightful comments on our manuscript. It is clear that the reviewers recognize the value of our work on developing an efficient approach for creating and evaluating complex combinatorial mutants and its application to CRISPR activators. The reviewers provided valuable feedback on several aspects of our study, including the need for additional data to demonstrate the improved performance of our top activators (MHV and MMH) over SAM, requests for more comprehensive off-target analyses, and inquiries about the principles guiding the selection of our most promising activators. They also highlighted areas in which the presentation of our data, usually related to the biology of activation and toxicity, could be enhanced.

In response to these comments, we have addressed each of the specific suggestions in detail below. Additionally, to enhance the robustness of our study, we have included several new analyses that were not specifically requested by the reviewers but which add additional insight into our findings. These include:

1. **AP/MS analysis:** We conducted affinity purification followed by mass spectrometry (AP/MS) to identify patterns in protein-protein interactions that drive activator strength and toxicity (see **Figure 6**, pasted below).
2. **Expanded toxicity analyses:** We extended our toxicity assessments to include different cell types to better understand the generalizability of this property (See **Supplementary Figure 13**, pasted below).

To the best of our knowledge, this work represents the only study that has both taken a systematic approach to quantifying activator toxicity across thousands of constructs as well as attempted to elucidate the potential biological basis of this toxicity.

We hope that the revisions and additional data will meet the reviewers' expectations and provide a clearer picture of the potential of our CRISPR activators, the utility of our high-throughput approach, and the significance of our biological findings. Thank you once again for your valuable feedback.

Editorial Note: Figure 6a in this Peer Review File is reproduced with permission from BioRender, *Created in BioRender. Giddins, M. (2025) <https://BioRender.com/l3knchv>*

Figure 6. Domain spatial arrangement does not influence activator binding patterns. (a) Workflow for evaluating activator binding. Twelve activators were C-terminally FLAG-tagged and transfected into HEK293T cells expressing dCas9 and a CXCR4-targeting gRNA. Cells expressing FLAG-tagged constructs were subjected to protein extraction followed by anti-FLAG affinity purification, and the resulting protein complexes were enzymatically digested and analyzed by mass spectrometry. (b) Principal Component Analysis (PCA) of activator binding profiles for each of the tested activators. Forward-reverse activator pairs are represented by identically-colored dots. Toxic activators are encircled by an ellipse. Axis labels indicate the percentage of total variance explained by each principal component. (c) Pairwise comparisons of protein binding profiles across all bipartite activator constructs and a toxic tripartite activator. The grid layout shows comparisons between constructs labeled on the diagonal. Each off-diagonal plot (left of diagonal) and R value (right of diagonal) compares the construct named in its row (x-axis) against the construct named in its column (y-axis). Dots represent individual

proteins' \log_2 (enrichment in binding over the unfused MCP control) for the two constructs being compared. Only proteins detected in all conditions are shown. Histograms along the diagonal show the distribution of \log_2 (enrichment in binding over the unfused MCP control) for each construct. Toxic activators (BEL1_VP16x2, VP16x2_BEL1, and VIRF2_ICP4-11_MLLx3) are boxed in red. Correlations were calculated using Pearson correlation coefficient (R). (d) Volcano plots showing differential protein binding between MLLx3_HSF1 vs. MLLx3_HSF1_P65 and between VIRF2_HSF1 vs. CITED2x4_VIRF2_HSF1, where \log_2 (fold-change) in protein binding is shown on the x-axis, and $-\log_{10}$ (p-value) is shown on the y-axis. Proteins significantly enriched in specific constructs (MLLx3_HSF1_P65 and CITED2x4_VIRF2_HSF1: left, MLLx3_HSF1 and VIRF2-HSF1: right) are highlighted in black. (e) \log_2 (fold-change) in protein binding for toxic (red) and non-toxic (blue) activators. Proteins are grouped by cellular function. The significance of the difference in values between toxic and non-toxic activators was assessed via unpaired two-tailed t-test for each protein. Only interactions with $p < 0.0001$ are shown.

Supplementary Figure 13. Activator toxicity does not depend on cell type. The correlations between activator toxicity scores across cell types following delivery of a 56-member “mini-library” into HEK293T, HeLa, and N2A cells are shown. Toxicity scores depict the $-\log_2$ (number of reads in the final unsorted cell library/number of reads in the plasmid library). Data were derived from two independent transduction replicates for each cell type. Correlations were calculated using Pearson correlation coefficient (R). Lower (more negative) values correspond to lower toxicity.

Reviewer #1 (Remarks to the Author):

In the present manuscript, Giddins et al. have developed an efficient approach for creating and evaluating complex combinatorial landscapes of multi-domain CRISPR activators, uncovered the principles of combinatorial AD assembly, and identified an improved activator. The authors first

individually tested a set of 230 ADs at three targets, and uncovered several highly potent domains. The authors further employed a high-throughput combinatorial approach for 25-high value ADs to test 625 bipartite and 15,625 tripartite CRISPR activator variants at a time. Based on screening results, the authors found that two selected promising activators showed enhanced activity against CRISPR-SAM across genome targets and cell lines, and untested off-targets at TTN gene. In addition, the authors explored the biochemical features of activation domains (ADs), determined how their combinatorial interactions shape activator behavior, and found that many activators produce substantial cellular toxicity independent of their ability to regulate gene expression. Overall, the topic is interesting, and this study contributes through providing researchers with increased options for using CRISPR activator in their research. However, some data are robust and not enough. Here are some comments and suggestions for the authors to improve the quality of the work.

We would like to thank Reviewer #1 for their clear summary of our paper and its significance. The reviewer acknowledged the novelty and potential impact of our work but pointed out several areas where the work could be improved, including the need for additional data to demonstrate the improved activity of our top activators, additional off-target analyses, and clarification of our rationale for selecting the most promising activators. We appreciate these insights and have addressed each comment and suggestion below.

1. The authors state that two potent activators MHV and MMH outperform SAM in multiple contexts. However, in Fig.6e-g and Fig.S25, MHV activity at these 9 targets was higher at some sites, equal at some sites, and lower at some sites compared with SAM in HEK293T. Similar situation is showed in N2A and Hela cells. Similarly, at most of targets, MMH showed comparable activity to SAM. To demonstrate the improved activity of both MHV and MMH, data are not far enough and more targets in those cell lines are needed. The negative control is also needed.

We appreciate the reviewer's careful analysis of our data and the opportunity to clarify our findings. The performance of MHV and MMH relative to SAM indeed vary across targets and cell types, and we agree that our initial statement may have been overly broad. Our revised description, intended to more accurately reflect the nuanced performance of MHV and MMH across different targets and cell lines, is pasted below.

[LINE 481]

MHV showed up to 11.6-fold increased activation over SAM against 11 of the 17 target genes (**Figures 7c-d, Supplementary Figure 28**; $p < 0.05$). For the remaining targets, MHV and SAM performed comparably, except against CXCR4, where SAM exhibited 1.3-fold increased activation over MHV. MMH demonstrated similar efficacy to SAM, producing up to a 2.0-fold improved activation on three out of 17 targets ($p < 0.05$), while the fold-changes for a number of additional targets approached but did not achieve statistical significance.

[LINE 497]

Having optimized our activators in the commonly used HEK293T cells, we sought to determine the plasticity of our tools across additional biomedically relevant human and mouse cell lines. We individually tested each MCP activator against a panel of human surface protein targets (CD2, CD45, EPCAM, EGFR), human non-surface protein targets (*TTN*, *HBG1*, *RHOXF2*, *NEUROD1*, *MIAT*, *ACTC1*, *ASCL1*, *IL1RN*, *IL1B*, *ZFP42*, *LIN28A*, and *IL1R2*), and mouse non-surface protein targets (*Ttn*, *Hbb-bh1*, *Actc1*, *Neurog2*, *Ins2*, *Itgav*, *Sim1*, and *Mef2d*) within HeLa and HCT116 or N2A lines, respectively. MHV and MMH displayed statistically meaningful improvements in activation over SAM for 10 out of 32 and two out of 32 targets, respectively (**Figures 7f-h and Supplementary Figures 29-31**; $p < 0.05$ for all increases). Several other targets showed a trend towards increased activation over SAM that approached, but did not reach, statistical significance.

As the reviewer will note from our answer above, for these revisions, we tested MHV, MMH, and SAM on additional targets and in one additional cell line. We believe that these new data points and comparisons will provide a more robust characterization of our top activators, MHV and MMH, across multiple contexts. For convenience, we have included our previous activation data along with new activation data below. Our new data are now included in the manuscript as Figure 7h and Supplementary Figures 28-31.

HEK293T

N2A

HeLa

HCT116

We have also analyzed our data in the context of another recent Nature Methods-published tool, CRISPR DREAM (Mahuta et al. 2023). The comparisons between our data and data from Mahuta et al. 2023 (representative comparisons showing the weakest, moderate, and strongest improvements over SAM) are shown below (all activation was quantified by evaluating fold-change in RNA). The improvements over SAM yielded by CRISPR-DREAM are in line with the relative improvements we report for MHV and MMH.

MHV & MMH

DREAM

It is worth noting that most next-generation activator studies, including the CRISPR-DREAM study, primarily use RNA as a readout. This approach typically yields higher target fold-changes compared to using protein as a readout. In our study, we have provided both RNA (panels shown above) and protein (panels shown below) fold-change data. As expected, our RNA fold-changes are higher than our protein fold-changes. When the CRISPR-DREAM study used protein as a readout for one of their panels, the observed fold-change was within the same range as ours. This consistency across different studies further supports the validity of our findings and sets expectations for what might be expected of a next generation activator tool.

The reviewer also suggests that the negative control data are needed. We note that all activation data are normalized to the negative control condition (MCP-empty), although this may not have been as clear as hoped. In line with the reviewer's comment, we have updated our methods to highlight how the activation data are plotted.

[LINE 1161]

All plotted activation data were normalized to a negative control condition consisting of MCP not fused to any AD. This control condition was tested concurrently with all experimental conditions to account for background signal and ensure accurate quantification of activation potential.

The figure legends also contain the following passage to help ensure our readers are aware of our use of a negative control to normalize the plotted data, for example:

[LINE 1752]

For panels a and c-h, data are normalized to a negative control line expressing MCP not fused to any protein.

Finally, should any reader want to see our raw data (including the negative controls) those are provided in the source data (“Source Data Figure 7 and Supplementary Figures 25-34.”)

2. For off-target gene regulation, should add at least one more target to perform RNA-seq analysis of MHV, MMH and SAM.

We appreciate the reviewer’s suggestion to perform our off-target experiment on one additional target gene (in addition to *TTN*). In line with the reviewer’s suggestion, we delivered an *HBG1*-targeted gRNA to cell lines expressing MHV, MMH, SAM, MCP (not fused to any domain), and un-engineered HEK293T cells in biological duplicate and performed RNA sequencing on all conditions after 48 hours. The correlation between gene expression data from lines expressing each activator and the negative control condition consisting of MCP not fused to any domain proved similar to the correlation between biological replicates. These data indicate that, against multiple targets, our activators do not stimulate wide-ranging transcriptional abnormalities. Of note, *HBG1* expression was not detectable in any condition. We have updated our off-targeting section to reflect these new insights.

[LINE 509]

To ensure that our new tools do not produce unintended effects on transcription, we performed unbiased RNA-sequencing on cells expressing our activators targeted to the *TTN* or *HBG1* genes. Gene expression data from lines expressing each activator displayed a high correlation with gene expression data from the negative control condition consisting of MCP not fused to any protein for both targets (Figure 7i, Supplementary Figures 32-34; R = 0.98 for MHV vs. MCP, MMH vs. MCP, and SAM vs. MCP for both targets; $p < 0.0001$ for all correlations). After filtering away genes with low baseline expression (<1 TPM), no activator induced significant upregulation (≥ 2 -fold change in expression) in more than five genes for either target, suggesting that, overall, these tools are highly specific.

For convenience, gene expression data from our *HBG1* off-targeting experiment, corresponding to Supplementary Figures 32, 33, and 34 in the text, are pasted below.

Supplementary Figure 32. Activators exhibit minimal off-target activity as compared to MCP not fused to any protein upon targeting *HBG1*. Quantifying off-target transcriptional perturbations induced by MHV-, MMH-, SAM-, and MCP-expressing HEK293T cells transfected with an *HBG1*-targeting gRNA. *HBG1* gene expression was below the <1 TPM cutoff in all conditions. Transcriptional aberrations driven by MCP alone were evaluated by comparing gene expression in the MCP-expressing cell line to wildtype HEK293T cells transfected with an *HBG1* gRNA. Data are shown as the mean of $n = 2$ independent transfections. Correlations were calculated using Pearson correlation coefficient (R).

Supplementary Figure 33. Activators exhibit minimal off-target activity as compared to cells not expressing MCP. Quantifying off-target transcriptional perturbations induced by MHV-, MMH-, SAM-expressing HEK293T cells transfected with a (a) *TTN*- or (b) *HBG1*-targeting gRNA compared to cells absent of any MCP protein. For all panels, genes showing <1 TPM (transcript per million) in either replicate of any construct in the given correlation were excluded before log transformation. For panel b, *HBG1* gene expression was below this cut-off, preventing it from being plotted. For all panels, data are shown as the mean of $n = 2$ independent transfections. For all panels, correlations were calculated using Pearson correlation coefficient (R).

Supplementary Figure 34. Transfection replicates show high correlations in gene expression. (a-b) Correlation between *TTN* (a) or *HBG1* (b) gene expression ($\log_2(\text{TPM})$ (transcripts per million)) in transfection replicates of activator- or control-expressing lines transfected with a *TTN*- (a) or *HBG1*- (b) targeting gRNA. For all panels, genes showing <1 TPM (transcript per million) in either replicate of any construct in the given correlation were excluded before \log transformation. Correlations were calculated using Pearson correlation coefficient (R). For panel A, red dot indicates gene expression of *TTN*. *HBG1* expression is not marked in panel B because it failed to meet the read cutoff.

3. According to current verified data and comparison data to SAM, among vast of activators variants, A22_A08 MHV and A03_A22 (MMH), showing the top high activity among these thousands of ADs, showed enhanced or comparable activity to SAM. This high-throughput work looks very cool, but for this combination designs results out only 1-2 variants with high activity better than SAM. The cost performance is too low, what is the defect of the current screening, or is there a more optimized design scheme. Please discuss this.

We appreciate the reviewer's thoughtful analysis of our work and the opportunity to address the points raised. We acknowledge that our high-throughput screening seems to yield a small number of top performers with enhanced activity compared to SAM. While we only fully worked-up MHV and MMH, our screen elucidated other activators that outperformed SAM. The MCP component of SAM, MCP-P65-HSF1 (MPH) ranked among the top 25 hits in our bipartite screen (25th out of 25), prompting its inclusion in our manual validation. In this experiment, MPH ranked 22nd, 18th, 23rd, and 25th on CD45, EPCAM, EGFR and CXCR4 respectively. These results suggest that other activators with improved activity over SAM outside of MHV and MMH exist. Furthermore, while our rate of finding potent activators may appear low given the

number of variants tested, we believe this outcome is both valuable and insightful for several reasons:

- Discovery of novel, potent activators: Despite the large number of variants screened, identifying even one or two activators that outperform the current gold standard (SAM) is a significant achievement, as exemplified by recent publications in high impact journals (Mahuta et al. 2023, Nature Methods) showcasing tools with similar performance as ours. As such, MHV and MMH represent important advances in CRISPR activator technology. Importantly, it is only through such a high-throughput approach to activator discovery that we could identify more optimal activators. Given the relatively sparse number of highly active CRISPR activators, had a one-at-a-time cloning and testing approach been employed, it most certainly would not have sufficed for identifying an improved system.
- Elimination of suboptimal combinations: By testing thousands of variants, we were able to eliminate a large number of suboptimal combinations. This "negative" data is equally valuable. It prevents future researchers from pursuing these less effective designs, saving time and resources in the field.
- Comprehensive exploration of the activator landscape: Our high-throughput approach allowed us to systematically explore a vast combinatorial space of activator designs. This comprehensive screening provides valuable insights into activator biology that go beyond just identifying top performers. It allows us to understand which combinations work, which don't, and why - information that is crucial for future activator design and to understand the fundamental biology of transcriptional activators.
- Methods development: The high-throughput screening pipeline we developed is itself a significant contribution to the field. This approach can be applied to other protein engineering challenges beyond CRISPR activators, potentially accelerating discoveries in various areas of biotechnology.

While we believe our screen performed exactly as designed, potential optimizations for future screens or approaches that one might consider are:

- Machine learning integration: Incorporating machine learning algorithms trained on our generated data could help predict promising combinations, potentially reducing the number of variants that need to be experimentally tested.
- Expanded parameter space: Future screens could explore additional parameters such as linker lengths, N- and C-terminal protein fusions, or even novel synthetic ADs.
- Context-specific screening: In cases where users have defined target(s) of interest they wish to regulate, designing screens for those unique contexts could yield activators optimized for particular applications.

In conclusion, while our screen yielded a select number of top performers, we believe the comprehensive nature of our approach, the insights gleaned, and the potential impact of the identified activators justify the high-throughput strategy. The knowledge and tools developed through this work lay a foundation for more targeted and efficient activator design in the future.

In light of the reviewer's comments, we have updated our discussion with the following passages to help highlight the utility of our approach and future routes of exploration:

[LINE 587]

This study also offers significant value beyond the novel constructs we characterized. Our comprehensive evaluation of over 15,000 activators eliminated a large proportion of suboptimal AD combinations, and showcases the need for high-throughput approaches for quickly navigating through sparsely populated landscapes. Moreover, our results provide the field with valuable "negative" data that will prevent future researchers from pursuing less effective designs. Our screening data also yielded important insights into combinatorial activator design, such as the impact of domain-domain interactions on activator performance, the relationship between toxicity and activation potential, and the biochemical properties of effective ADs. This information is valuable for the broader field of synthetic biology and transcriptional engineering, and lays a foundation for more targeted and efficient activator construction. Future optimizations of our screening approach might include integrating machine learning to predict promising combinations of activators or expanding the parameter space to include factors such as linker lengths or additional ADs.

4.The authors selected two promising activators for further workup. What is the principle of selection. It's obvious not random. But if choose based on the activity, the bipartite A16_A22 is higher activity than A03_A22 (MMH) in EPCAM, CD45, EGFP and CXCR4 and not chosen in Fig. 6a-b and Fig.S23. Please explain the considerations.

The reviewer is correct in noting that our selection was not random. The selection of MHV (A22_A08) and MMH (A03_A22) for further workup was based on a combination of factors, including but not limited to activation potency. Here are the additional considerations that guided our selection:

- Toxicity profile: Crucially, both MHV and MMH exhibited relatively low toxicity in our high-throughput screens (as noted in Supplementary Data 5 of our manuscript). This low toxicity was a key factor in their selection, as high toxicity could significantly hinder the activator's practical applicability.
- Domain composition: We prioritized activators with distinct domain compositions to explore diverse activation mechanisms. MHV combines the human HSF1 AD with the viral VIRF2 AD, while MMH is an all-human construct, composed of the CBP-recruiting fragment from the human MLL protein and the HSF1 AD, potentially offering advantages in therapeutic contexts where immunogenicity may be an important consideration.
- Reproducibility and consistency: The performance of MHV and MMH was consistently high across multiple replicates and targets, indicating robust and reliable activation.

Regarding the specific case of A16_A22 mentioned by the reviewer:

While A16_A22 did indeed show higher activity than MMH (A03_A22) on some targets, MMH was prioritized given that it was the best all human domain-containing activator and, as noted above, for contexts in which immunogenicity might prove important, we considered it a more viable option.

We acknowledge that this decision-making process was not fully explained in the original manuscript. We have now revised our manuscript with the following sentences to better articulate our selection criteria.

[LINE 467]

From these validation studies, we selected two promising activators **based on their high activity across targets, low toxicity profiles (Supplementary Data 5), and distinct domain compositions:** MCP fused to the human HSF1 AD and the human herpesvirus 8 VIRF2 AD (MHV) and MCP fused to three copies of the CBP-recruiting fragment from the human MLL protein and the human HSF1 AD (MMH). **MHV proved our strongest overall activator, showing exceptional potency across target genes. MMH, a construct composed entirely of human domains, presented potential advantages in therapeutic settings where immunogenicity would pose a concern.** To evaluate the performance of our new modulators against the current state-of-the-art, we opted to test MHV and MMH against the gold-standard MCP activator, SAM, composed of dCas9-VP64 paired with an MCP-P65-HSF1 (MPH) fusion (**Figure 7b and Supplementary Figure 27**).

5. In addition to activity advantages, the authors could describe the other advantages of both MHV and MMH, such as size.

We value the reviewer's recommendation to highlight the other advantages of MHV and MMH. In addition to the previously discussed improvements in activation potential over SAM, MHV and MMH offer several other benefits:

1. Compact size: MHV's MCP fusion is smaller than SAM's MCP fusion by 210 base pairs (729 bp vs. 939 bp) and does not require VP64 to be fused to the Cas9 protein, removing another 150 bases in overall size. This reduced size could be advantageous for packaging into viral vectors such as adeno-associated viral vectors for gene therapy applications or for facilitating easier delivery.
2. Human-origin domains: The all-human domain composition of MMH could lead to reduced immunogenicity in *in vivo* settings. This is a crucial consideration for therapeutic applications, where minimizing immune responses is essential for the safety and efficacy of gene therapies.
3. Lower toxicity: MHV exhibits lower toxicity than the MCP component of SAM, as demonstrated in our high-throughput screens. This reduction in toxicity is significant for maintaining cell viability and function, making MHV particularly suitable for applications involving long-term expression or in sensitive cell types.

We have edited the text to further highlight the advantages of MHV and MMH compared to existing tools.

[LINE 605]

Furthermore, MHV and MMH offer several additional advantages over existing CRISPR activators. **MHV, our most potent activator, is more compact than the MCP component of SAM by 210 bp (729 bp versus 939 bp). This reduced size could facilitate easier packaging into viral vectors for gene therapy applications. MHV also induced lower levels of toxicity than the MCP component of SAM, ranking in the bottom 9th vs. 17th percentile of toxicity, respectively, in our high-throughput screen (Supplementary Data 5) – potentially enhancing its suitability for long-term expression or use in sensitive cell types.** It is worth noting that VIRF2, a key component of MHV, has been recently uncovered in other studies, highlighting the convergent discovery of this potent AD. **MMH, while less potent than MHV, still achieved similar or higher levels of activation across most tested targets compared to SAM and produced relatively low toxicity, ranking in the bottom 33rd percentile of toxicity within our screen. Importantly, MMH contains all-human domains, which might lead to reduced immune responses in *in vivo* settings, a crucial consideration for therapeutic applications. Both MHV and MMH systems do not include the VP64 component used in many existing CRISPR activators, including SAM, which simplifies the overall system design and potentially reduces production complexity. These features, combined with their enhanced activation potential, position MHV and MMH as promising tools for a wide range of applications.**

6. Among these CRISPR activator variants, whether the SAM combination is present, and what about the level of activity at three screening targets.

The reviewer raises a question regarding whether SAM was present in our screened library, and if so, how it performed. To address the reviewer's query:

Presence of SAM combination: In our screen, only the MCP component of SAM (MCP-P65-HSF1) was present in our library of CRISPR activator variants. Our screen was performed using dCas9 rather than dCas9-VP64, which is used in SAM, to allow us to analyze more clearly the activity of the different transactivation domains without the confounding factor of activity from VP64.

Activity levels of SAM at the three screening targets: As noted, our screen does not contain the full SAM system but only the MCP component of SAM. When looking at the performance of the MCP component of SAM (MCP-P65-HSF1) across the three screening targets:

MCP-P65-HSF1 performed the 2nd best of all activators on EPCAM, the 60th best of all activators on CXCR4, and the 68th best of all activators on the fluorescent reporter (all ranks are out of 625 members in the bipartite screen). For comparison, our top hit MHV was ranked

3rd, 19th, and 4th, and MMH was ranked 24th, 45th, and 66th on EPCAM, CXR4, and the reporter, respectively.

7. The authors found that increases in AD copy number do not enhance toxicity suggest that the molecular mechanism driving this property rapidly saturates. But why the increased toxicity of 3 copies than that of 1 or 2 copies, in Fig. 4f.

The reviewer is correct in noting that three copies of the same domain shows distinctly different behavior than one or two copies of the same domain. We note, however, that three copies of the same domain does not increase toxicity, but rather decreases it. In our plots, more negative numbers are indicative of lower toxicity. The reason for this finding is unclear, but, as we briefly mention in the manuscript, we hypothesize that highly repetitive constructs (such as those with three copies of the same AD) may be more prone to instability and turnover, leading to reduced levels of the construct, and as such, reduced toxicity.

To help clarify the way the toxicity data are plotted we have adjusted the figure legend for 4f to explain the relationship between the y-axis value and overall toxicity.

[LINE 1648]

Median toxicity scores of ADs present at various copy numbers, with inert PFs occupying any position not occupied by an AD. **For panels d-f, lower (more negative) values correspond to lower toxicity.**

8. Line400, the authors state the MHV and MMH yielded relatively low cell toxicity. How to define the standard of cell toxicity. In this cellular toxicity evaluation system, what is the level of SAM?

Defining the standard of cell toxicity: In our study, we quantify toxicity by comparing the read count in the plasmid library and the transduced cell pool (Supplementary Data 5, Figure 4a, see Methods) for all activators. We use $-\log_2$ fold change of the read count in the cell pool compared with plasmid library as the index of toxicity. It generated a normally distributed curve among all activators.

SAM toxicity level: We apologize for not explicitly stating the toxicity level of SAM in our original manuscript. In our screen, we only included MCP-P65-HSF1, the MCP component of SAM, instead of the full version of SAM (with the dCas9-VP64 component) in our toxicity assessment. We found that MMH and MHV fall within the 33rd and 9th percentile of toxicity scores, respectively, among the 625 bipartite activators. The MCP component of SAM, MPH, falls between the two, within the 17th percentile. All three activators demonstrated low toxicity scores, appearing in the bottom third of toxicity scores for the entire library of 625 bipartite activators.

To highlight the relative toxicity levels across our newly developed activators and the MCP component of SAM, we have adjusted our discussion section as follows:

[LINE 605]

Furthermore, MHV and MMH offer several additional advantages over existing CRISPR activators. MHV, our most potent activator, is more compact than the MCP component of SAM by 210 bp (729 bp versus 939 bp). This reduced size could facilitate easier packaging into viral vectors for gene therapy applications. **MHV also induced lower levels of toxicity than the MCP component of SAM, ranking in the bottom 9th vs. 17th percentile of toxicity, respectively, in our high-throughput screen (Supplementary Data 5) – potentially enhancing its suitability for long-term expression or use in sensitive cell types.** It is worth noting that VIRF2, a key component of MHV, has been recently uncovered in other studies, highlighting the convergent discovery of this potent AD. **MMH, while less potent than MHV, still achieved similar or higher levels of activation across most tested targets compared to SAM and produced relatively low toxicity, ranking in the bottom 33rd percentile of toxicity within our screen.**

9. In Fig. 6e-g and Fig. S25, compared to SAM, MHV displayed the opposite activity at CXCR4 target from single site and four multiplexed targets. Would you be able to explain why this is the case?

Thank you for your observation regarding the differential activity of MHV at CXCR4 compared to SAM in Fig. 6e-g (now 7e-g) and Fig. S25 (now S28) (pasted below).

7e (multiplexed targeting)

S28 (individual targeting)

It is possible that the differences between the single site and multiplexed experiments stem from slight differences between how the assays are performed (e.g., exact time after splitting the cells that they are transfected into, time post transfection to harvesting the cells), differences in transfection efficiency between experiments (e.g. cells slightly more or less confluent between

studies), and/or the interplay between the CRISPR activator and the limited amounts of endogenous transcriptional resources when applied in the single gene activation vs. multiplex activation paradigms. In addition, while each of the comparisons shows statistically significant differences, the magnitude of the difference in each case between SAM and MHV at the CXCR4 locus is small and may stem from inherent assay noise. Despite our efforts to minimize noise, this variability is difficult to eliminate entirely in biological experiments of this nature. However, when considering activator performance across 25 targets in various experimental contexts and cell lines, the consistent trend of MHV outperforming SAM is what we find most compelling, rather than any individual comparison.

10. In general, the biological replicates are 3 or over. Thus, it is recommended that for activity comparison of among MHV, MMH and SAM, three biological replicates are set.

While we agree that increasing the number of biological replicates generally enhances the reliability of experimental results, we believe that the breadth of data we present across 25 targets in four cell lines supports our findings. Furthermore, the use of two replicates is supported by previous highly cited studies (Chavez et al., "Comparative Analysis of Cas9 Activators Across Multiple Species" Nat Methods, 2016; Kiani et al., "Cas9 gRNA engineering for genome editing, activation and repression" Nat Methods, 2015; Yeo et al., "An enhanced CRISPR repressor for targeted mammalian gene regulation" Nat Methods, 2018; Karasu et al., "Removal of TREX1 activity enhances CRISPR–Cas9-mediated homologous recombination" Nat Biotechnology, 2024; Jin et al., "Enhancing homology-directed repair efficiency with HDR-boosting modular ssDNA donor" Nat Communications, 2024), where two replicates were used to test tools across numerous targets, contexts, and cell lines – similar to this work.

Reviewer #2 (Remarks to the Author):

Giddins and Kratz et al. describe a CRISPRa protein variant library platform for identification and optimization of proteins that can activate gene expression. The authors have some nice insights in the manuscript such as that activator protein activity does not correlate with toxicity and that indiscriminate combinations of domains is unlikely to result in better tools. A major goal of this manuscript is identification of a new best in class gene activator but problematically it is not clear that the MHV activator nominated by their screens is meaningfully better than the SAM system across cell types.

We would like to thank Reviewer #2 for their clear summary of our paper and its significance. The reviewer acknowledged the novelty and potential impact of our work but noted the need to clarify that MHV is meaningfully better than SAM across cell types. We appreciate this valuable insight and have added additional data (discussed below) to address this concern, demonstrating MHV's efficacy across more cell types and targets. Furthermore, we would like to draw attention to additional data that we have added to this revision regarding the mechanism of activator toxicity, which provides another unique and valuable angle to our work.

Figure 1 and associated supplemental figures.

Initially the authors screen 230 domains in an arrayed format using transient transfection to identify domains that modulate activation of 2 endogenous genes and a reporter. Please clarify whether Supplementary Figure 2A shows all 230 domains? Their activation results are very much dominated by VP16 variants which are already very well established as strong activators and thus this screen doesn't open up much new biology or tools space.

The reviewer inquires about the preliminary testing we performed using transient transfection to determine which domains to use for our combinatorial screen. They are correct that the Supplementary Figure, which is now labeled 3A due to the addition of a new Supplementary Figure 1, shows all 230 domains, and express concern that VP16 derivatives strongly dominate this experiment, thus limiting the value of our study. In reply, we would like to emphasize the difference in performance of the VP16 derivatives between one-by-one domain testing and the combinatorial screens. As shown in the below figure, domains containing VP16 derivatives did not dominate the combinatorial screen. This result highlights the value of our large-scale approach, which allowed us to explore the un-intuitive properties of AD synergy, whereas the one-by-one single domain analysis alone may have misdirected us towards focusing exclusively on VP16 variants. Furthermore, we highlight that our best performing activator, MHV, does not contain a VP16-derived domain and instead is composed of the human HSF1 AD and the viral VIRF2 domain. Finally, by testing 230 domains for their ability to induce gene expression, we gained insight into the properties of strong activators. We also produced a valuable dataset on dozens of full length ADs that had not been previously tested for gene activation and represent potentially useful parts for future engineering efforts.

VP16-containing vs non-VP16 constructs, bipartite screen

The authors should graphically present show how many unique domains are present in the library vs variant and homologs.

Throughout our analysis, we divided domains into clusters based on 50% sequence identity using the uclust algorithm (Robert C. Edgar, Search and clustering orders of magnitude faster than BLAST, Bioinformatics, Volume 26, Issue 19, October 2010, Pages 2460–2461), which is widely used to cluster sequences for taxonomic analysis. The graph below shows the size and diversity of these clusters. Some families, namely VP16s, VP64s, BEL1s, and E1As, were composed of more than one cluster due to their high degrees of sequence diversity. These are grouped and colored together in the below pie chart to visually show their similarity. The largest cluster, singletons, shows the 151 tested ADs which did not cluster with any other domains at a 50% sequence identity level.

We have provided a Supplementary figure with these data below:

Supplementary Figure 1. Clustering of individually tested domains. All individually tested parts were clustered at 50% sequence identity using uclust. Families of related activators which were clustered into multiple sets were grouped and colored similarly. Clusters are labeled with their centroid member's name followed by the size of the cluster.

Figure 2 and associated supplemental figures.

The authors examine properties of hits from Figure 1 and arrive at the conclusion that acidic domains tend to be activators and intermixing acidic and hydrophobic domains tends to promote activation. Both observations support known features of ADs and thus this is not particularly novel nor do these results provide insight into how to better design tools.

We would like to thank Reviewer #2 for their feedback on Figure 2. While we agree that our findings on the importance of acidic domains and the intermixing of acidic and hydrophobic residues align with known features of ADs, we believe our study offers a number of insights and contributions to the field:

Previous works have shown that ADs are not modular protein domains functioning independently of local context. Rather, they are disordered, and their activity is sensitive to the local context. Most ADs have been screened attached to classical DNA binding domains (DBDs) such as Gal4, Tet, and zinc fingers. Our work represents the first comprehensive analysis of sequence features in the context of Cas9. Given the non-modular nature of ADs, it was not a given that their biochemical signatures would remain consistent in this new context.

While several synthetic biology groups have screened peptides for AD activity on Cas9, they did not perform detailed analyses on sequence features such as acidity, hydrophobicity, and disorder. Conversely, other groups have screened for AD activity on classical DBDs and examined sequence features. Our study combines both approaches. The fact that we observe convergence between our Cas9-based results and previous findings with classical DBDs is reassuring and suggests that ADs are perhaps more modular than previously thought.

Despite these biochemical insights, our results demonstrate that biochemical characteristics (such as acidity) alone are currently insufficient to predict activator function. Even more sophisticated machine learning models like PADDLE currently fail to accurately predict activation potential from sequence alone. This highlights a crucial gap in our understanding and underscores the continued importance of empirical testing in identifying potent ADs.

To highlight this point, we have adjusted the text as follows:

[LINE 167]

Previous works have established the association of a number of biochemical traits, such as acidity and hydrophobicity, with activator function (**Figure 2a**)³⁷⁻⁴³. **However, these associations have not been widely explored in the context of dCas9 fusions.** Examining the amino acid composition of all tested activators, we found that "hits," ADs that produced two-

fold or greater activation on at least one target, were composed of more negatively charged residues than “misses” (Figure 2b, $p < 0.0001$). This finding underscores the importance of acidity for activation and demonstrates that this principle generalizes to dCas9 scaffolds.

Please clarify for Figure 2F – were ADs predicted with PADDLE and then plotted across the length of the protein? Or were these minimal 80AA tiles identified as ADs from prior literature and then plotted across the length of each protein?

We appreciate the reviewer’s request for clarification regarding Figure 2F. The ADs plotted in Figure 2F were neither predicted by PADDLE nor based on 80AA tiles from prior literature. Instead, the plotted ADs are derived from our initial testing of 230 domains, as described in Figure 1 and Supplementary Figure 1 (now Supplementary Figure 2). We further note that, in comparison to many previous studies that use short synthetic tiles to study activator biology, our domains span a wide range of sizes from 16 to 564 amino acids. By employing this approach, we sought to better preserve domains’ native folding and function. Figure 2F shows the relationship between where in the native protein each domain was sourced from (i.e. was the domain we extracted located in the N or C-terminus of its native protein) and whether or not the domain was a “hit” or a “miss” in our activation testing (Figure 1). In this context, hits are defined as domains that activated at least one target (EPCAM, CXCR4, or the fluorescent reporter) by at least two-fold. Domains that did not meet this criteria were classified as misses. To account for variations in the lengths of the native proteins from which our ADs were sourced, we divided each protein into 20 equal regions spanning from the N-terminus to the C-terminus. Then, for each AD we tested, we determined which of the 20 region(s) it overlaps with in its native protein. Next, for each region, we counted how many of the overlapping ADs were hits and how many were misses. Examining the results of Figure 2F, we notice a clear trend where domains that were found to be hits in our screen were more often derived from the C terminus of their native proteins as compared to misses, which show a more even distribution across all regions of their native proteins. We speculate that this finding may be due to the C-terminal positioning of the transactivation domain in our fusion constructs. Domains that are naturally located at the C-terminus of their parent protein may be more likely to adopt a functional conformation when fused in this manner.

To help clarify this figure better for our readers we have made the following adjustments to the figure legend and have also added additional details to the methods section:

[LINE 1593]

(f) The relative position of each of 230 tested ADs within its native protein is shown (see Methods for details). “N,” “Mid,” and “C” refers to the N-terminus, middle, and C-terminus, respectively, of each AD’s native protein. Relative frequency indicates how often hits or misses are derived from each region. We observe an enrichment of hits derived from the C-terminus of their native proteins.

[LINE 747]

We used UniProt to identify the location of each AD within its native protein. For a domain of length L_d appearing within a protein of length L_p at the N th amino acid, start position was defined as $(N-1)/L_p$, and end position was defined as $(N-1+L_d)/L_p$. Using these definitions, a domain starting from the first amino acid in a protein is assigned a start value of 0, and a domain running to the last amino acid in a protein is assigned an end value of 1. **Because the length of each activator's native protein is different, we first normalized each protein by dividing its length into 20 sections. We then count how frequently hits and misses are sourced from domains which overlap with each of the 20 sections. Domains can span multiple sections if every section they touch is counted.** For three domains, the sequence we used in our screen contained either a mismatch or single amino acid indel compared to the canonical sequence found in UniProt. These instances were handled explicitly by applying the above algorithm to the sequence present in UniProt.

Figure 3 and associated supplemental figures.

The authors combine 25 domains into all pairwise and three way combinations in a pooled lentiviral library and then screen these combinations as activators across 2 endogenous genes and a reporter to identify proteins that potentially activate gene expression.

Please clarify for Figure 3E and Supplementary Figure 6 which of the 20-30 proteins tested in validation experiments would have been called hits in the screening data.

Figure 3E and Supplementary Figure 6 (now Supplementary Figure 7) illustrate the results of the experiments we performed to validate our screening data. In these experiments, we independently tested all 25 AD combinations for the single-domain library or 20-30 random AD combinations for the bipartite and tripartite libraries. The purpose of picking random clones (for the bipartite and tripartite libraries) was to be sure that our screen accurately quantified all members of the libraries, including those that performed poorly. Due to their random selection, these proteins do not represent the top hits identified in our screens. We used these studies to provide us with confidence that our screens, in addition to showing good correlations between biological replicates, yielded results that would match those obtained from more traditional one by one approaches. These results served as a “go-ahead” to discuss insights related to the properties of not just the most potent activators, but all activators within our pools.

For clarity, we would like to direct the reviewer to Supplementary Figure 25 and Supplementary Figure 26, where we specifically identify top hits based on their high activity in the screening data and validate them against multiple target genes. These constructs are not the same as those used for validation in Figure 3E and Supplementary Figure 7.

Choosing top hits (Supplementary Figure 25):

Supplementary Figure 25. Identification of top screening hits for additional workup. (a-b) Bipartite or tripartite top hits, respectively, chosen for additional workup are shown. Hits taken forward for additional testing (green) performed well relative to the total library (gray) across both replicates on three screening targets (EPCAM, CXCR4, and synthetic reporter).

Testing top hits (Supplementary Figure 26):

Supplementary Figure 26. MMH and MMH perform well compared to all bipartite and tripartite top hits. (a-b) Manual validation of 25 bipartite and 25 tripartite hits, respectively. Constructs were stably integrated into HEK293T cells and transfected with a mixture of gRNAs targeting

four surface-protein genes, CD45, EPCAM, EGFR, and CXCR4, followed by flow cytometry to measure activation. Each activator is annotated based on the species it is derived from or its functionality as a PF, human ADs (white dots), viral ADs (gray dots), and PFs (black dots). Positions within the MCP fusion (P1-P3) are shown, where P1 represents the position most proximal to MCP. (c) The same assay described in panels a and b was conducted on the top five bipartite and top five tripartite hits. For all panels, activator-produced fold-changes in target expression were normalized to a negative control line expressing MCP not fused to any protein. Black dots depict performance of each of two independent transfection replicates. For panels a and c, activation by MCP-HSF1-VIRF2 (MHV, green bar) and MCP-MLLx3-HSF1 (MMH, blue bar) are shown.

It is known that lentiviral recombination during reverse transcription is a problem for using barcodes to track elements within a library. Please quantify via long read NGS how much recombination occurs for constructs with this domain/barcode library design after construct integration by lenti into the genome. Intra- and inter- molecular recombination could result in deletions or in scrambling of the domains or in decoupling of barcodes from their assigned domains. This is noted by the authors in Figure 4F as a problem and so should be quantified to enable others to use such a platform approach in a robust manner.

The reviewer notes an extremely important aspect of lentivirus screens: the risk of cross-over between elements of a library. In our studies, we included a number of controls to assess whether this phenomenon was occurring. The three barcoded controls (included in our library) that we used were 1) an “MCP-mutant” control, which contains the puromycin resistance gene, but does not contain any AD fused to MCP; 2) a “Puro-mutant” control, which does not produce puromycin resistance; 3) a “Supernatant-mutant” control, which is not able to be packaged into lentivirus because it lacks viral packaging elements in the construct. The Puro-mutant control allows us to detect whether cells were either multiply infected with several viral constructs or whether there were crossovers between barcodes during packaging (the initial concern raised by the reviewer). The supernatant control allows us to detect the presence of plasmid DNA that remains in the harvested lentiviral supernatant, which can potentially confound our analysis with false signal. All controls were spiked into the viral barcoded library at an equal ratio. The dramatic difference in the abundance of the various control barcodes is shown in the graph below, where the MCP-control displayed an average of 10 times higher frequency than the Puro-mutant control. This indicates that two of the greatest concerns when performing lentiviral based screens, multiple infections per cell and crossovers within the library, were quite rare in our dataset. Finally, we note that, while the suggestion of performing long-read sequencing of the integrated constructs is an excellent one, we have found that the process of performing long-range PCR on complex libraries generates PCR induced crossover, which itself produces false positive signals, confounding the interpretation of the resulting data.

Furthermore, prior to carrying out our high-throughput screen, we tested the frequency of lentiviral recombination using two barcoded lentiviral MCP activator constructs: one with an intact blasticidin resistance gene and another with a non-functional blasticidin resistance gene. For clarity, we have pasted a schematic of the constructs below.

We mixed the constructs together at equal ratios and made virus out of the mix. Then, we transduced it into HEK293T cells (the cell type ultimately used for our screen) at different multiplicities of infection (MOIs) and cultured the cells in 5 µg/mL blasticidin. After four passages, we extracted genomic DNA from the cells, amplified the barcode region using primers that bind to common regions flanking the barcode, and sequenced the barcodes. We hypothesized that by measuring the frequency that the barcode associated with the non-intact blasticidin resistance gene appeared in our sequencing, we could estimate the frequency of barcode swapping.

Our data show that, at low MOI (“1 μ L virus” in the graph below), barcode swapping is infrequent (<5%). Our “high MOI controls,” in which multiple integration events (e.g., intact blasticidin barcode + non-intact blasticidin barcode integrated into the same cell) should have allowed the barcode associated with the non-intact blasticidin resistance gene to persist in culture, behaved as expected, showing increasing frequencies of the non-intact blasticidin barcode with increasing MOI.

We hope this explanation addresses the reviewer's concerns regarding lentiviral recombination and provides clarity on the measures taken to ensure the robustness of our platform approach.

Figure 4 and associated supplemental figures.

The authors note that for the tripartite library many library members did not meet the read thresholds required for to accurately quantify their function. Please state clearly what fraction of the single domain, bipartite and tripartite library meets the read threshold in the main text. The authors comment that the distribution of constructs present in plasmid DNA is poorly correlated with the distribution in cells and attribute this to toxicity. If the authors believe this to be true they should state clearly what fractions of the single domain, bipartite and tripartite library are toxic in the main text.

We scored all library members in both single-domain and bipartite conditions. However, analyzing the tripartite library proved more challenging due to its larger size and stronger toxicity phenotypes. Consequently, we scored 1,527 (10.1%), 1,665 (10.7%), and 1,237 (7.9%) tripartite activators within screens targeting EPCAM, CXCR4, and the fluorescent reporter, respectively. We would have preferred to quantify the activation potential of all members of the tripartite library. Yet, the significant depletion of many tripartite activators due to high toxicity levels suggests that, even if we had quantified all constructs across targets, the toxic ones would not have been of practical utility given their strong negative effects on cell fitness.

We have now modified our manuscript to include the number of constructs scored within our tripartite screens in the main text, as suggested by the reviewer.

[LINE 256]

Upon examining our screening data, **we found that only 10.1%, 10.7%, and 7.9% of tripartite library members met the read threshold cut-offs implemented to accurately quantify their function for the EPCAM-, CXCR4-, and fluorescent reporter-targeting screens, respectively.**

It is noted that disordered domains have increased toxicity but the correlation isn't extremely strong leaving it a bit unclear what drives toxicity. It is known that activators can be highly toxic and so although this data is useful it doesn't provide enough mechanism to design tools that avoid toxicity. Basically the authors are saying everything needs to be tested empirically for activity and toxicity. Is there a more constructive message that can be delivered or do the authors plan to make this library cloning strategy very easy to use for the community if empirical testing is always required?

We appreciate the reviewer's observation regarding the relationship between domain disorder and toxicity – and their emphasis on the broader challenge of predicting activator behavior *a priori*. While our study does not provide a complete mechanistic basis for activator toxicity, we believe it offers several valuable contributions:

1. We discovered that toxicity is an intrinsic property of each individual domain – and that it manifests independent of its spatial arrangement or interactions with other domains. This additive property should allow researchers to predict the toxicity of multi-domain constructs based on the toxicities of its individual components.
2. We found that toxicity is not correlated (and sometimes is even anticorrelated) with activation potential. This challenges the prevailing assumption that stronger activators are necessarily more toxic.
3. Our study revealed that toxicity positively correlates with the fraction of disorder-promoting residues and negatively correlates with hydrophobicity. This finding points to potential biological mechanisms and might serve as a filter that future tool builders could impose onto the constructs they make to help reduce their chances of producing high levels of toxicity.
4. We observed that toxicity doesn't increase linearly with the number of copies of an AD, suggesting a saturation effect. This insight is relevant for the design of multi-domain activators expressing more than one copy of the same domain.
5. Our data showed that toxicity did not depend on the target gene. This insight is particularly valuable, as it allows researchers to assess activator toxicity without testing against multiple target genes.
6. Importantly, our study revealed toxicity to be a more widespread issue among activators than previously recognized, underscoring the critical importance of testing for it.
7. We have conducted additional studies, including AP-MS experiments on activators with varying toxicity levels, to better understand the protein-protein interactions driving toxicity. We identified a number of cofactors with diverse biological functions that were

expressing FLAG-tagged constructs were subjected to protein extraction followed by anti-FLAG affinity purification, and the resulting protein complexes were enzymatically digested and analyzed by mass spectrometry. (b) Principal Component Analysis (PCA) of activator binding profiles for each of the tested activators. Forward-reverse activator pairs are represented by identically-colored dots. Toxic activators are encircled by an ellipse. Axis labels indicate the percentage of total variance explained by each principal component. (c) Pairwise comparisons of protein binding profiles across all bipartite activator constructs and a toxic tripartite activator. The grid layout shows comparisons between constructs labeled on the diagonal. Each off-diagonal plot (left of diagonal) and R value (right of diagonal) compares the construct named in its row (x-axis) against the construct named in its column (y-axis). Dots represent individual proteins' $\log_2(\text{enrichment in binding over the unfused MCP control})$ for the two constructs being compared. Only proteins detected in all conditions are shown. Histograms along the diagonal show the distribution of $\log_2(\text{enrichment in binding over the unfused MCP control})$ for each construct. Toxic activators (BEL1_VP16x2, VP16x2_BEL1, and VIRF2_ICP4-11_MLLx3) are boxed in red. Correlations were calculated using Pearson correlation coefficient (R). (d) Volcano plots showing differential protein binding between MLLx3_HSF1 vs. MLLx3_HSF1_P65 and between VIRF2_HSF1 vs. CITED2x4_VIRF2_HSF1, where $\log_2(\text{fold-change})$ in protein binding is shown on the x-axis, and $-\log_{10}(\text{p-value})$ is shown on the y-axis. Proteins significantly enriched in specific constructs (MLLx3_HSF1_P65 and CITED2x4_VIRF2_HSF1: left, MLLx3_HSF1 and VIRF2-HSF1: right) are highlighted in black. (e) $\log_2(\text{fold-change})$ in protein binding for toxic (red) and non-toxic (blue) activators. Proteins are grouped by cellular function. The significance of the difference in values between toxic and non-toxic activators was assessed via unpaired two-tailed t-test for each protein. Only interactions with $p < 0.0001$ are shown.

Supplementary Figure 13. Activator toxicity does not depend on cell type. The correlations between activator toxicity scores across cell types following delivery of a 56-member “mini-library” into HEK293T, HeLa, and N2A cells are shown. Toxicity scores depict the $-\log_2(\text{number of reads in the final unsorted cell library}/\text{number of reads in the plasmid library})$. Data were derived from two independent transduction replicates for each cell type. Correlations were calculated using Pearson correlation coefficient (R).

What time point post infection is displayed in Figure 4C?

The data shown in Figure 4C represents cells harvested approximately 11 days post-infection, after three passages in puromycin.

We have revised our figure legends (Figure 4C and Supplementary Figure 9), pasted below, to reflect this timeline and provide clarity on the experimental process.

[LINE 1639]

(c) Correlation between reads from two independent transductions of the tripartite library into dCas9-, EPCAM gRNA-expressing cells. **Data shown represents cells harvested 11 days post-infection, following three passages in 0.5 µg/mL puromycin selection.**

[LINE 1933]

Library member abundances are similar across biological replicates in cells. The correlation between numbers of reads for two unsorted cell biological replicates within EPCAM- (single-domain and bipartite), CXCR4- (single-domain, bipartite, and tripartite), and synthetic reporter- (single-domain, bipartite, and tripartite) targeting screens are shown. **For EPCAM- and CXCR4-targeting screens, data shown represent cells harvested 10, 11, and 11 days post-infection for the single-domain, bipartite, and tripartite screens, respectively, following three passages in 0.5 µg/mL puromycin selection. For the reporter-targeting screen, data shown represent cells harvested 13, 14, and 14 days post-infection for the single-domain, bipartite, and tripartite screens, respectively.** Correlations were calculated using Pearson correlation coefficient (R).

For Supplemental Figure 10 the authors should determine experimentally whether these constructs toxicity is dCas9 dependent or independent by expressing the MCP fusion proteins in the absence of dCas9 and then tracking cell populations over time. The toxicity of such a dCas9/MCP-fusion/sgRNA-MS2 system can relate to protein expression ratios for dCas9/MCP. Knowing if the toxicity is associated with the DNA binding protein is highly useful and would help interpret their claims in Supplemental Figure 12 around squelching.

We value the reviewer's suggestion regarding Supplementary Figure 10 (now Supplementary Figure 11). The goal of Supplementary Figure 11 was to verify that our approach to scoring toxicity in the high-throughput screen was reproducible when examined under the same conditions (with dCas9 present) using a traditional lower throughput approach. While testing the toxicity of MCP fusion proteins in the absence of dCas9 could provide interesting insights, it is important to emphasize that these activators are specifically designed to function with dCas9 and will always be used in this context in practical applications. Therefore, we believe that evaluating toxicity in the presence of dCas9 provides the most relevant information for real-world applications of these tools.

Furthermore, as noted in our responses to the reviewers' previous questions, we have conducted additional experiments to better understand the drivers of toxicity in our system and its universality. In future studies, outside of the scope of this current manuscript, we look forward to further elucidating the mechanism driving the strong cellular toxicity we observed.

Figure 5 and associated supplemental figures.

In this figure the authors show that activation domains effects are context specific in the tripartite but not bipartite context- from a biological perspective why would this be true? They support this by analyzing the tripartite constructs that have a stabilization domain and 2 activation domains to show in that context the 2 AD domains in the tripartite context behave more like the bipartite proteins. One simple explanation could be that the tripartite domains are not stable proteins. The authors could test this by measuring protein levels for some number of bifunctional and trifunctional proteins (with 3 activation domains or 2 activation domains and a stabilization domains) to determine the effect of protein levels on activity.

We thank the reviewer for drawing attention to one of our more surprising findings. In our manuscript, we note that, for activator constructs with two domains (bipartite constructs), the order in which those domains are fused together exerts little effect on their overall ability to activate gene expression. In contrast, for constructs with three domains (tripartite constructs), we observe a notable order effect, where one orientation often shows much stronger activation than another. We then show that for tripartite constructs that contain two ADs and one inert domain (non-AD), domain order appears to exert a diminished effect on gene activation compared to tripartite constructs that contain three ADs. To quantify this (using our EPCAM-targeting screen as an example), the correlation between forward and reverse activation scores for bipartites was $R=0.86$. For all tripartites, it was $R=0.21$. For tripartites that contain an inert domain, it was $R=0.49$ – significantly closer to the behavior of the bipartites. From these data, we conclude that, in tripartite constructs, there are more opportunities for interference between ADs in orienting the recruited transcriptional machinery in the proper configuration to drive gene expression. In other words, we hypothesize that it is not simply the binding of the transcriptional machinery that drives a tripartite activator's performance, but how the various recruited components are arranged in 3D space.

In contrast to this interpretation, however, the reviewer suggests that the reason we observe tripartite activators with one inert domain showing similar behavior in both orders is not that the inert domain is reducing the complexity of the system or causing interference between ADs, but rather that tripartite activators are unstable. Thus, by removing one of the ADs within a tripartite assembly and replacing it with an inert domain, we are making chimeric proteins that are now more stable and in turn better able to function. While we cannot rule out whether our tripartite activators that contain an inert domain are more stable, it does not detract from the main point we are making, which is that order matters more for tripartite systems. In other words, it is possible that some orientations of tripartite constructs are less stable than others, and that is what drives their differences in gene activation. But this still means that the orientation of the same domains can make a large difference in the overall protein's behavior in tripartite systems.

In addition, our toxicity data suggest that stability likely does not explain why orientation matters for the tripartite constructs, since in contrast to gene activation, activators produce consistent toxicity independent of the order of the fused domains. If a difference in stability explained why tripartite activators in one orientation displayed better activation than another, then we would

have expected to see a similar effect of order on toxicity (i.e., that the more stable orientations are more toxic), but we do not see this. Furthermore, our recent mass spectrometry data provides additional insights that suggest differences in construct stability do not underlie our results. In these data, we find that tripartite activators composed of the same set of ADs but in different orders bind similar sets of proteins at similar efficiencies (Supplementary Data 6), even in cases where there are large differences in activator potency between them. If certain tripartite orientations were less stable, we would not expect both orientations to consistently show similar binding patterns, as the less stable variant would be expected to capture fewer targets.

Finally, we have updated our discussion with the above points to help better contextualize our findings for our readers.

[LINE 560]

We hypothesized that, while toxicity can arise when an activator simply binds to other proteins, activation requires bound transcriptional cofactors to adopt a precise spatial configuration that enables the assembly of a functional transcription complex at the target gene. **Our mass spectrometry analyses corroborate this model, confirming that tripartite forward-reverse pairs – even those with different activation potentials – bind to similar sets of proteins. These results underscore that binding alone is insufficient to determine activation potency and point to the critical role of 3D configurations in modulating transcriptional output.**

Figure 6 and associated supplemental figures.

A major goal of this effort is to identify better gene activators for the research community. As such the authors choose 25 bipartite and tripartite hits from their screens and then validate these in an arrayed format. The authors claim the bipartite hits generally perform comparably to the tripartite but in fact for EGFR and CXCR4 the tripartite hits perform substantially more poorly than the bipartite hits on average (Supp Figure 23). Please state this in the text.

We appreciate the reviewer's careful analysis of our data, particularly regarding the comparison between bipartite and tripartite activators. Upon re-examination of Supplementary Figure 23 (now Supplementary Figure 26), we agree that our initial statement about the comparable performance of bipartite and tripartite hits does not accurately reflect the data for all targets, especially EGFR and CXCR4.

We will revise the text to more accurately describe these results, stating:

[LINE 462]

Consistent with our observation of strong domain-domain interference in fusions containing three ADs, tripartite activators generally produced similar or lower activation levels than their bipartite counterparts.

Combined analysis of Figure 6C&D and Supplemental Figure 25 shows that MHV more robustly activates gene expression of 6/9 genes relative to SAM in HEK293s (4/6 differences are substantial and the other 2/6 are significant but not substantial). The authors then compare MHV and SAM in 2 other cell lines and claim that MHV is better than SAM for 3/6 genes however of the 3/6 genes where MHV is statistically better than SAM in N2A or HeLa cells only 1 gene (TTN) is substantial whereas the other two genes show statistically significant but similar levels of activation (e.g. for CD2 and EPCAM in HeLa all 3 constructs activate expression by 2-3 fold and it is not clear the difference between 2 vs 3 fold is meaningful biologically). So basically the entire claim that MHV is substantially better than SAM rests on HEK293 data or on a single gene activation measurement in one other cell type. It would be very problematic to nominate a “best in class activator” that in fact is really a “best in class HEK293 cell type specific activator” as this will waste a substantial amount of other lab’s time and not fulfill the goal of identifying a new best in class gene activation construct. Please show MHV is better than SAM in other cell types more convincingly.

We appreciate the reviewer's careful analysis of our data and the opportunity to clarify our findings. The performance of MHV and MMH relative to SAM indeed vary across targets and cell types, and we agree that our initial statement may have been overly broad. Our revised description, intended to more accurately reflect the nuanced performance of MHV and MMH across different targets and cell lines, is pasted below.

[LINE 481]

MHV showed up to 11.6-fold increased activation over SAM against 11 of the 17 target genes (**Figures 7c-d, Supplementary Figure 28**; $p < 0.05$). For the remaining targets, MHV and SAM performed comparably, except against CXCR4, where SAM exhibited 1.3-fold increased activation over MHV. MMH demonstrated similar efficacy to SAM, producing up to a 2.0-fold improved activation on three out of 17 targets ($p < 0.05$), while the fold-changes for a number of additional targets approached but did not achieve statistical significance.

[LNE 497]

Having optimized our activators in the commonly used HEK293T cells, we sought to determine the plasticity of our tools across additional biomedically relevant human and mouse cell lines. We individually tested each MCP activator against a panel of human surface protein targets (CD2, CD45, EPCAM, EGFR), human non-surface protein targets (*TTN*, *HBG1*, *RHOXF2*, *NEUROD1*, *MIAT*, *ACTC1*, *ASCL1*, *IL1RN*, *IL1B*, *ZFP42*, *LIN28A*, and *IL1R2*), and mouse non-surface protein targets (*Ttn*, *Hbb-bh1*, *Actc1*, *Neurog2*, *Ins2*, *Itgav*, *Sim1*, and *Mef2d*) within HeLa and HCT116 or N2A lines, respectively. MHV and MMH displayed statistically meaningful improvements in activation over SAM for 10 out of 32 and two out of 32 targets, respectively (**Figures 7f-h and Supplementary Figures 29-31**; $p < 0.05$ for all increases). Several other targets showed a trend towards increased activation over SAM that approached, but did not reach, statistical significance.

As the reviewer will note from our answer above, for these revisions, we tested MHV, MMH, and SAM on additional targets and in one additional cell line. We believe that these new data points and comparisons will provide a more robust characterization of our top activators, MHV and MMH, across multiple contexts. For convenience, we have included our previous activation data along with new activation data below. Our new data are now included in the manuscript as Figure 7h and Supplementary Figures 28-31.

N2A

HeLa

HCT116

We have also analyzed our data in the context of another recent Nature Methods-published tool, CRISPR DREAM. The comparisons between our data and data from Mahuta et al. 2023 (representative comparisons showing the weakest, moderate, and strongest improvements over SAM) are shown below (all activation was quantified by evaluating fold-change in RNA). The improvements over SAM yielded by CRISPR-DREAM are in line with the relative improvements we report for MHV and MMH.

It is worth noting that most next-generation activator studies, including the CRISPR-DREAM study, primarily use RNA as a readout. This approach typically yields higher target fold-changes compared to using protein as a readout. In our study, we have provided both RNA (panels shown above) and protein (panels shown below) fold-change data. As expected, our RNA fold-changes are higher than our protein fold-changes. When CRISPR-DREAM used protein as a readout for one of their panels, the observed fold-change was within the same range as ours. This consistency across different studies further supports the validity of our findings and sets expectations for what might be expected of a next generation activator tool.

It seems like the fold changes for *TTN* as measure by qPCR and rna-seq in 6D versus 6H are not concordant. If the qPCR readouts are not trustworthy then this calls key data in figure 6 and supplemental figure 25 into question. Please comment on this?

The reviewer raises an important concern regarding the discrepancy between qPCR and RNA-seq measurements for *TTN* activation. This phenomenon of differing fold-change magnitudes between qPCR and RNA-seq readouts has been observed in previous studies of CRISPR activators (data excerpts from those studies are shown below, and in cases where primary data are not provided, they were estimated based on the figures presented). For instance:

- With MHV (our study), we see a ~150x difference (750-fold by qPCR vs. 5-fold by RNA-seq)
- CRISPR-DREAM (Mahata et al., Nat Methods, 2023) shows a ~100x difference (100,000-fold by qPCR vs. 1,000-fold by RNA-seq)
- SAM exhibits a ~50x difference (5,000-fold by qPCR vs. 100-fold by RNA-seq) from Chavez et al., Nature Methods, 2016

[Figure Redacted]

These discrepancies likely stem from inherent differences, including the dynamic range and sensitivity, as well as the potential biases in the measurement techniques. Importantly, while the absolute fold-changes differ, the relative performance of activators remains consistent across both readouts.

Reviewer #3 (Remarks to the Author):

The authors set out to establish a CRISPR-based platform that enables high-throughput exploration of combinatorial activator constructs, leveraging a modified CombiSEAL system to bypass individual synthesis of activator domain (AD). Additionally, the authors make an effort to characterize activator potency on endogenous genes as opposed to exclusively a synthetic reporter, addressing the shortcoming of generality of current state-of-the-art CRISPR activators across target contexts. They identify trends in the features of strong activators and in how ADs can be composed in combinations. The authors develop and characterize two potent activators, MHV (HSF1-VIRF2) and MMH (MLLx3-HSF1), which outperform the CRISPR activator SAM across several sgRNAs targeting endogenous genes in a handful of cell types (mouse and human) in terms of activation potency. Importantly, the authors claim that these new activators display reduced cytotoxicity compared to SAM. They characterize activator cytotoxicity in multipartite constructs, and show trends with disorder and hydrophobicity. Further, the authors find multipartite AD cytotoxicity approximates to the sum of individual AD toxicities, irrespective of AD order in the overall construct. This contrasts with their somewhat surprising finding that activator potency is highly dependent on the order of ADs within their tripartite (but not bipartite) constructs.

I feel the largest impact will be from the MHV tool, which looks stronger than SAM and more limited data suggests is also less toxic. They also set the laudable goal of uncovering rules for composing ADs, but I feel their success here is somewhat less confirmed. They observed several relevant trends, but it is not proven that these can now be taken as 'rules' to guide, for example, the more successful design of the next library of ADs. For example, it seems that MHV is selected because of its performance in the screen, not because they applied new knowledge

learned about AD composition principles. They don't seem to comment on whether MHV is representative of the principles they learned. I appreciate the way they highlight the importance of toxicity, although they do not mine their data in a way that explains the toxicity phenomenon further (perhaps their biggest impact on the toxicity question is they push back on the theory that squelching is the driver).

We would like to thank the reviewer for their concise summary of our manuscript. The reviewer recognizes the significance of investigating activators in the context of endogenous genes, as well as the value of our toxicity analyses, and we share their hope that MHV will prove to be an impactful development from this paper. They also helpfully raise a number of points that could benefit from more analysis, including additional insight into the mechanism of toxicity, the degree to which the trends we observed can be taken as general rules, and a number of questions about our screening technique. We appreciate these questions and have addressed them below.

Main comments

1. The figures are neat and often have very helpful clear schematics. However, they mostly show overall trends and it is difficult to learn much specifically for the activator domains because of the anonymized way they present the data. For example, in Figure 3-6 they usually label genes with a code ID instead of a more easily interpretable gene name. I understand the names of triple combos can get long, but I urge the authors to consider ways to make more information-rich visualizations of their datasets. For example, very few dots in the plots are labeled with gene names. It is not clear where negative controls (e.g. dCas9- or MCP-only) or positive/well-known activators (e.g. VP16, VP64, p65, HSF1) fall in the plots. Given the importance of VIRF2 and MLL in the final figure, they could also label them in more of their plots.

We sincerely appreciate the reviewer's thoughtful feedback on our figures and data presentation. We agree that enhancing the clarity and information content of our visualizations would greatly benefit readers in interpreting and understanding our results. We propose the following improvements to address these concerns:

- Gene and domain labeling: We have replaced the code IDs with actual gene names and AD identifiers in Figures 3-6.
- Highlighting key constructs: We clearly label and highlight the MCP component of SAM (MPH) in the activation and toxicity plots from our high-throughput screens.
- Supplementary information: We have already provided comprehensive supplementary tables, supplementary files, and source data with full details of all constructs and their performance metrics. This will allow interested readers to access the complete dataset while maintaining the clarity of the main figures.

Below, we have included select examples of our updated figures. By implementing these changes, we aim to strike a balance between providing detailed information and maintaining the clarity of our figures. We believe these improvements will significantly enhance the readability

and interpretability of our data, allowing readers to gain deeper insights into the performance of diverse ADs and their combinations.

Figure 3. (c) EPCAM screen scores of all tripartite library members, along with examples of the abundances of select activators within bins 1-4. Data are shown as the mean ($n = 2$ independent screen replicates).

Previous 3D New 3D (yellow dot is MPH)

Figure 3. (d) Correlations between EPCAM screen scores across two biological replicates for single-domain, bipartite, and tripartite screens. **Yellow dot (bipartite panel, middle) indicates performance of MCP-P65-HSF1 (MCP component of SAM).**

Previous 5A

New 5A

Figure 5. Domain-domain interference affects activator performance. (a) EPCAM bipartite screen scores produced by each of 25 ADs in position one, the position closest to MCP (y-axis),

upon pairing with each of 25 ADs in position two, the position most distal to MCP (x-axis). Purple and yellow coloring highlights lower- or higher-performing combinations, respectively. ADs are ordered from weakest- to strongest-performing within the single-domain EPCAM screen. Each activator is annotated based on the species it derives from or its functionality as a PF, human ADs (white dots), viral ADs (gray dots), and PFs (black dots).

Figure 7. (a) Manual validation of best-performing bipartite hits. Activator-produced fold-changes for the top 10 performers against one of the four targets, EPCAM, are shown. Highly potent activators, MCP-HSF1-VIRF2 (MHV) and MCP-MLLx3-HSF1 (MMH), are highlighted in green and blue, respectively.

2. Why remove the directly fused VP64, as in SAM, when making the MHV/MMH constructs? Could they outperform SAM even more often if they also had the VP64 component? This could be tested, or the reasoning better explained.

Our decision to remove the VP64 fusion from dCas9 in the MHV/MMH constructs was deliberate, as these constructs were originally discovered in this context (without the VP64 component). Furthermore, by removing the VP64 component, we reduce the overall size of the delivered construct and reduce potential added toxicity from an additional activator component in the system.

3. I think the investigation of toxicity is a strength of the paper, although ideally there would be firmer conclusions about toxicity. If domain toxicity “remains constant independent of surrounding context”, wouldn’t the slopes in D-E be along the $y=x$ line? Could you show replicates of the toxicity score?

With regards to the reviewer's comments on additional insights into toxicity, in this updated manuscript, we have included additional studies, such as AP-MS experiments on activators with a wide range of toxicity levels, to better understand the interactions driving this property. We identified a number of cofactors with diverse biological functions, including protein folding and processing, cell structure and adhesion, mitochondrial function, and membrane transport, that were significantly more likely to be bound by toxic constructs (Figure 6e, shown below for convenience). We also performed toxicity studies in two additional cell lines, human Hela cervical cancer cells and mouse N2A neuroblastoma cells (Supplementary Figure 13, shown below for convenience) and observed a similar pattern of toxicity across these contexts. These new data suggest that toxicity occurs via interference with core cellular processes, explaining its consistency across cell lines and our finding that diverse toxic activators shared similar interaction partners when examined via AP-MS.

Figure 6. (e) $\text{Log}_2(\text{fold-change})$ in protein binding for toxic (red) and non-toxic (blue) activators. Proteins are grouped by cellular function. The significance of the difference in values between toxic and non-toxic activators was assessed via unpaired two-tailed t-test for each protein. Only interactions with $p < 0.0001$ are shown.

Supplementary Figure 13. Activator toxicity does not depend on cell type. The correlations between activator toxicity scores across cell types following delivery of a 56-member “mini-library” into HEK293T, HeLa, and N2A cells are shown. Toxicity scores depict the $-\log_2(\text{number of reads in the final unsorted cell library}/\text{number of reads in the plasmid library})$. Data were derived from two independent transduction replicates for each cell type. Correlations were calculated using Pearson correlation coefficient (R). Lower (more negative) values correspond to lower toxicity.

With regards to the reviewer’s comment on the slope of the relationship between toxicity measurements for forward-reverse pairs of activators in Figure 4D, we agree that, in an idealized setting, one might expect to observe a perfect $y=x$ relationship. We believe that we do not see this relationship due to biological and technical sources of experimental noise, as a similar deviation from the $y=x$ relationship is seen in our biological replicate correlations of toxicity scores.

For convenience, we have pasted these relationships below.

Below, we show the biological replicates for our toxicity measurement along with the lines of best fit through these data.

Bipartite toxicity replicate fit line equations:

EPCAM: $y = 0.7275x + 0.0199$

CXCR4: $y = 0.6195x + 0.0345$

Tripartite toxicity replicate fit line equations:

EPCAM: $y = 0.7701x + 0.4219$

CXCR4: $y = 0.6754x + 0.6883$

With regards to Fig 4e, we are comparing toxicities between screens. While these correlate linearly, we believe it is not surprising that the relationship is not $y=x$, as these libraries are different sizes, and were passaged for slightly different amounts of time. As a result, the degree to which activators enrich or deplete should (and does) correlate, but is not the exact same.

Further, is there bottlenecking in the screen that causes outliers, and is this handled by filtering (and/or the use of UMIs)? I ask because bottlenecking could be suggested by some elements with many reads in one unsorted cell replicate and 0 in the other replicate.

As the reviewer suggests, our team applied several approaches to reduce the chances of bottlenecking the library. First, each library was transduced into $>1,000x-10,000x$ the number of cells as library members. In addition, during passaging, we split each transduced flask into 4 downstream flasks and carried all flasks forward until the end of the screen, allowing us to preserve as many independently transduced cells as possible. Finally, we sorted $>300x-3,000x$ the number of cells as barcoded variants present in each library. By scaling our screens as such, we aimed to minimize bottlenecking due to insufficient sampling and passaging of the library. Furthermore, in our data analysis, we made use of UMIs to remove barcoded variants that displayed outlier behavior or were more enriched than other UMIs encoding the same CRISPR activator (full details of data filtering and data analysis approach are available in the Materials and Methods).

The authors mention that toxicity has been reported for previous activators such as VPR and SAM; where do those fall in these plots, if they (or related fusions) are present (e.g. Figure 4D)?

The reviewer raises an important point regarding the toxicity of previously reported activators like VPR and SAM. To address this concern, we have highlighted the toxicity score of the MCP component of SAM (MPH), along with the toxicity scores of our new activators, MHV and MMH, in the context of the toxicity scores of all bipartite activators in the figure below (top figure). We also performed the same analysis for VPR fused to MCP in the context of all tripartite activators in an additional figure (middle figure). To summarize our findings:

As for the bipartites, the MCP component of SAM (MPH) exhibits a toxicity score at the bottom 17th percentile of all bipartite constructs. MHV shows a lower toxicity score than MPH, at the 9th percentile, while MMH shows a higher toxicity at the 33rd percentile.

As for VPR, MCP-VPR shows a toxicity score in the middle range for our tripartite constructs.

Our re-worked version of Figure 4D shows the consistency of toxicity measurements between forward and reverse orientations of bipartite activators, including MPH.

Figure 4. (d) Correlations between toxicity scores of activators in forward (AD1-AD2) and reverse (AD2-AD1) orientations in bipartite and tripartite screens. **Yellow dot (bipartite panel, left) indicates toxicity score of MCP-P65-HSF1 (MCP component of SAM).**

Finally, to highlight the relative toxicity levels across our newly developed activators and the MCP component of SAM, we have adjusted our discussion section as follows:

[LINE 605]

Furthermore, MHV and MMH offer several additional advantages over existing CRISPR activators. MHV, our most potent activator, is more compact than the MCP component of SAM by 210 bp (729 bp versus 939 bp). This reduced size could facilitate easier packaging into viral vectors for gene therapy applications. **MHV also induced lower levels of toxicity than the MCP component of SAM, ranking in the bottom 9th vs. 17th percentile of toxicity, respectively, in our high-throughput screen (Supplementary Data 5) – potentially enhancing its suitability for long-term expression or use in sensitive cell types.** It is worth noting that VIRF2, a key component of MHV, has been recently uncovered in other studies, highlighting the convergent discovery of this potent AD. **MMH, while less potent than MHV, still achieved similar or higher levels of activation across most tested targets compared to SAM and produced relatively low toxicity, ranking in the bottom 33rd percentile of toxicity within our screen.**

Several results show toxicity and activation are not correlated; which is interesting and not fully explained here. Also, given how they compare to the plasmid library, it may not be possible for

them to assess if “toxicity” occurs during cell library growth or during lentiviral packaging, but this would be of interest and could affect how they use the term “toxicity”.

The reviewer raises an interesting point regarding the source of the toxicity that is observed, and whether it comes from difficulties in lentiviral delivery of each activator or activator effects on cell growth. While lentiviral packaging plays a role (as evidenced by a weak correlation between toxicity and construct length in Supplementary Figure 12, shown below), our data in Supplementary Figure 11 (shown below) provide strong evidence that the observed toxicity is related to activators’ effects on cell growth. In this experiment, we mixed cells expressing individual activators with control cells and monitored their relative growth over time, demonstrating that the toxicity effects we observe are indeed occurring during cell growth.

Supplementary Figure 12. Activator toxicity correlates weakly with total construct length. Correlations between the total length of each AD or AD combination and its toxicity score for single-domain, bipartite, and tripartite screens are shown. Correlations were calculated using Spearman correlation coefficient (ρ). Lower (more negative) values correspond to lower toxicity.

Editorial Note: Supplementary Figure 11a in this Peer Review File is reproduced with permission from BioRender, *Created in BioRender. Giddins, M. (2025) <https://BioRender.com/cgc9g54>*

a

b

c

Supplementary Figure 11. Experimental validation of screen-derived toxicity scores. (a) Schematic for experimentally evaluating activator toxicity. Activators randomly chosen from the tripartite library were made into stable cell lines in dCas9- and CXCR4 gRNA-expressing HEK293T cells. Activator-expressing cell lines were each mixed at a 1:1 ratio with HEK293T cells stably expressing MCP fused to GFP (MCP-GFP). Every 72 hours, cell mixes were split 1:8, and GFP expression was quantified using flow cytometry. Non-toxic activators (blue) are expected to show smaller reductions in population size (as evaluated by the fraction of the non-GFP population, non-green) than toxic activators (pink) over time. (b) Growth of tripartite clones isolated from the tripartite library, after being mixed 1:1 with an MCP-GFP control-expressing line and passaged at a 1:8 ratio twice. Activator population sizes (y-axis, non-GFP populations) were normalized to their baseline levels at the start of the experiment. The abundance of each activator at each of two passages, P1 and P2, on the x-axis, are shown. A construct containing a mutated MCP protein (dark blue), in which the sixth amino acid in the sequence, glutamine, was changed to a TAG stop codon, was included as a control. Data are shown as the mean of two independent transduction replicates. (c) Correlation between the reduction in population size of each individually tested activator over the passaging experiment (x-axis) and its screen-produced toxicity score (y-axis). Correlations were calculated using Spearman correlation coefficient (ρ). Lower (more negative) values correspond to lower toxicity.

The pooled measurement of toxicity doesn't correspond perfectly with the individual measurements in Supp Fig 10C ($\rho=0.58$); is this the explanation or is there another reason?

We appreciate the reviewer's observation regarding the correlation between our pooled toxicity measurements and individual toxicity measurements in Supplementary Figure 10C, now Supplementary Figure 11C (shown below).

Supplementary Figure 11. (c) Correlation between the reduction in population size of each individually tested activator over the passaging experiment (x-axis) and its screen-produced toxicity score (y-axis). Correlations were calculated using Spearman correlation coefficient (ρ). Lower (more negative) values correspond to lower toxicity.

The Spearman correlation coefficient of 0.58, indicating a strong positive correlation, demonstrates that our pooled toxicity measurements are indeed largely reflective of individual activators' effects on cell growth. However, as the reviewer correctly notes, the correlation is not perfect. Several factors may contribute to this observation:

1. As noted in Supplementary Figure 12, we do observe minor effects of the length of the construct on its presence in the library. These factors, while not dominant, could contribute to differences between pooled and individual measurements.
2. Our observations in Supplementary Figure 11B indicate that toxicity tends to stabilize over time. However, for these measurements, we needed to fully select stable lines expressing each activator before beginning the experiment. This process may have resulted in an underestimation of the full toxicity effect, as the most severely affected cells might have been lost during the selection process.
3. There may be slight differences in activator copy number between the individual toxicity measurements and the screen toxicity measurements. For individual measurements, we transduced cells with each activator separately, resulting in pure populations. In contrast, the screen involved transducing cells with the whole library at low MOI, leading to a

mixed population. This difference in transduction approach could result in variations in copy number and, consequently, in the observed toxicity.

These factors likely collectively contribute to the observed differences between pooled and individual toxicity measurements. Despite these differences, the strong positive correlation supports that our toxicity scores largely capture activators' effects on cell fitness, while highlighting the complexity of measuring such effects in different experimental contexts.

Would the authors expect the AD to also be expressed during lentiviral packaging, such that toxicity in the packaging cell could also impact viral titer?

The reviewer is correct that we anticipate that the ADs are expressed within the packaging cells and, as such, likely cause toxicity and a subsequent impact on viral titer. However, we do not believe this toxicity in the packaging lines affects our results. All members of the library are delivered to the packaging cells as a pool via transfection and thus are packaged under similar conditions. Given that during transfection, cells uptake thousands of plasmid copies, we expect packaging cells to express an array of different constructs with no one construct dominating. Thus, we expect every transfected cell to experience the average toxicity of all the constructs in the library. Finally, our toxicity data are normalized to the abundance of all library members in the unsorted cell pool. Thus, any bias in packaging is controlled for and thus unlikely to affect our ability to identify potent activators.

4. Does the C-terminal trend only hold for dCas9-AD fusions where the AD is displayed on the C-terminal end? Does the original position of the AD also correlate with how well it works in the context of the MCP fusions; e.g. do ADs from the middle of the protein work better in position 1 in the multipartite fusions?

Not including the protein folding domains, our library of 22 activators has 3 domains from the N terminus, 2 from the middle (internal domains), 11 domains from the C terminus, and 6 domains which were engineered/designed and thus cannot be assigned to any natural location. A paired comparison of the performance of bipartites of the form MCP-Internal_domain-C_terminal_domain (hypothetical good orientation) vs MCP-C_terminal_domain-Internal_domain (hypothetical bad orientation) shows effectively the same performance of the two orientations. Of note, while these data suggests that the origin of a domain may not be as important in the context of MCP fusions, the small number of ADs examined <20 as compared to the 230 tested in the dCas9 contexts is a limitation of these analyses.

5. Figure 5B – order has a major impact on tripartite but not bipartite activators, can this surprising result be further explained? The authors suggest activation requires a precise spatial configuration, but how is that compatible with the results from the bipartite elements? Perhaps relatedly, how long is the linker between ADs? Is it possible that the tripartite behave differently than the bipartite domains (e.g. much greater effects of domain order) due to lentiviral recombination between repetitive linkers/domains?

We appreciate the reviewer's question regarding the difference in order effects between bipartite and tripartite activators. This is indeed a surprising result that warrants further explanation.

The difference in order effects between bipartite and tripartite activators can be better understood in light of our new mass spectrometry data. As shown in our PCA analysis of the mass spec results (Figure 6B, shown below), forward-reverse pairs for both bipartite and tripartite activators cluster closely together. This indicates that domain order does not significantly affect proteins bound for either bipartite or tripartite activators.

Figure 6. (b) Principal Component Analysis (PCA) of activator binding profiles for each of the tested activators. Forward-reverse activator pairs are represented by identically-colored dots. Toxic activators are encircled by an ellipse. Axis labels indicate the percentage of total variance explained by each principal component.

While our activation data shows that bipartite activators are relatively insensitive to domain order, tripartite activators exhibit strong order-dependent effects on activation. This apparent distinction between effects of domain order on activation vs. cofactor binding suggests that activation requires more than just binding – but rather, binding in configurations that enable the formation of a productive transcriptional complex. We hypothesize that the larger, more complex tripartite systems are more sensitive to the spatial arrangement of these interactions and competitions between neighboring ADs.

To further support our model, our mass spec data (Figure 6d, shown below) shows that tripartite activators generally bind more proteins than their bipartite counterparts. This suggests that tripartite activators have no trouble binding proteins, but that their activation potential is more dependent on the specific configuration of these interactions.

Figure 6. (d) Volcano plots showing differential protein binding between MLLx3_HSF1 vs. MLLx3_HSF1_P65 and between VIRF2_HSF1 vs. CITED2x4_VIRF2_HSF1, where $\log_2(\text{fold-change})$ in protein binding is shown on the x-axis, and $-\log_{10}(\text{p-value})$ is shown on the y-axis.

Proteins significantly enriched in specific constructs (MLLx3_HSF1_P65 and CITED2x4_VIRF2_HSF1: left, MLLx3_HSF1 and VIRF2-HSF1: right) are highlighted in black.

With regards to the linker, we employed a 9 amino acid long linker of primarily glycines and serines between domains.

We do not believe the difference in behavior between bipartite and tripartite activators stems from lentiviral recombination. Prior to carrying out our high-throughput screen, we tested the frequency of lentiviral recombination using two barcoded lentiviral MCP activator constructs: one with an intact blasticidin resistance gene and another with a non-intact blasticidin resistance gene. For clarity, we have included a schematic of the constructs below.

We mixed the constructs together at equal ratios and made virus out of the mix. Then, we transduced HEK293T cells (the cell type ultimately used for our screen) at different multiplicities of infection (MOIs) and cultured the cells in 5 µg/mL blasticidin. After four passages, we extracted genomic DNA from the cells, amplified the barcode region using primers that bind to common regions flanking the barcode, and sequenced the barcodes. We hypothesized that by measuring the frequency that the barcode associated with the non-intact blasticidin resistance gene appeared in our sequencing, we could estimate the frequency of barcode swapping.

Our data show that, at low MOI (“1 µL virus” in the graph below), barcode swapping is infrequent (<5%). Our “high MOI controls,” in which multiple integration events (e.g., intact blasticidin barcode + non-intact blasticidin barcode integrated into the same cell) should have

allowed the barcode associated with the non-intact blasticidin resistance gene to persist in culture, behaved as expected, showing increasing frequencies of the non-intact blasticidin barcode with increasing MOI.

Furthermore, in this study, the "MCP-mutant" control, described in a response above, serves as a key indicator for assessing whether lentiviral recombination might be influencing activation scores. If lentiviral recombination were significantly occurring, the barcodes for the MCP-mutant (inactive) construct could potentially swap with those of active constructs, leading to artificially high or inconsistent performance scores for the MCP-mutant. However, since the MCP-mutant control consistently shows poor performance, even in the tripartite contexts with three barcodes, we can conclude that barcode swapping is not a major factor. This result indicates that activation scores reflect true functional differences rather than artifacts of recombination.

We hope this explanation addresses the reviewer's concerns regarding lentiviral recombination and provides clarity on the measures taken to ensure the robustness of our platform approach.

Minor comments

1. They could better demonstrate the impact of including UMIs, e.g. showing the data with or without UMI aware analysis.

We appreciate the reviewer's suggestion to better demonstrate the impact of including Unique Molecular Identifiers (UMIs) in our analysis. The main advantage of using UMIs lies in their ability to help us detect and eliminate anomalously enriched sequences, which are likely the result of PCR amplification biases or other technical artifacts rather than genuine biological

effects. This benefit is especially pronounced in our bipartite screen. To illustrate the impact of UMI-aware analysis, we have prepared a comparative figure (attached below) that shows the results of our EPCAM bipartite biological replicate analysis with and without UMI consideration. This figure clearly demonstrates the significant difference that UMI-aware analysis makes in our data interpretation. In the version without UMI analysis, we observe dramatic deviations in activation for a certain fraction of the library in one replicate. Upon closer examination, we found that these deviations stemmed from the anomalous enrichment of a single UMI in one bin of that replicate. Such enrichment is likely a technical artifact rather than a true biological signal (as different UMIs encoding the same activator did not show such dramatic enrichment). By incorporating UMI information into our analysis, we were able to identify and remove these outliers, resulting in a more accurate and reliable representation of activator performance across replicates. We believe that this additional analysis and visualization will provide readers with a clear understanding of the critical role that UMIs play in our experimental design and data interpretation and have included it as Supplementary Figure 35, shown below.

Supplementary Figure 35. UMIs enable detection and correction of noise. Comparison of biological replicates as scored from the pipeline as used (left) vs an alternative pipeline that does not incorporate UMIs (right) shows that UMI filtering removes noise that otherwise resulted in extreme outliers. UMI-aware pipeline filters out individual UMIs which are anomalously over-represented (See methods) and takes the average of the score across remaining UMIs. Axes are scaled to the same numerical values.

2. Supp Fig 24 has a floating “e” near “Off target”. I also think the schematic showing what MHV and MMH look like is very useful and could be moved into a main figure.

Editorial Note: Supplementary Figure 27 in this Peer Review File is reproduced with permission from BioRender, Created in BioRender. Giddins, M. (2025) <https://BioRender.com/Igdoc71>

Regarding Supplementary Figure 24, we thank the reviewer for pointing out the floating "e" near "Off target". We have included the updated Supp Fig 24 (now re-numbered as 27) without the "e" below.

Supplementary Figure 27. Schematic depicting validation experiments (multiple targets, multiple cell lines, off-targeting) intended to evaluate the potency, generalizability, and specificity of top hits, MHV and MMH, against a current state-of-the-art MCP activator, SAM.

We also appreciate the reviewer's insight regarding the schematic of MHV and MMH structures. We agree that this information is helpful for understanding the composition of these constructs. We have integrated the schematic into Figure 7 (shown below), where we present the performance data for MHV and MMH, as panel b.

Editorial Note: Figure 7b in this Peer Review File is reproduced with permission from BioRender, Created in BioRender. Giddins, M. (2025) <https://BioRender.com/lgdoc71>

Figure 7. MHV and MMH outperform SAM in multiple contexts. (a) Manual validation of best-performing bipartite hits. Activator-produced fold-changes for the top 10 performers against one of the four targets, EPCAM, are shown. Highly potent activators, MCP-HSF1-VIRF2 (MHV) and

MCP-MLLx3-HSF1 (MMH), are highlighted in green and blue, respectively. **(b) Schematic depicting composition of three activators taken forward for further workup: our new activators, MHV and MMH, and the gold-standard MCP activator, SAM.** (c-d) Testing MHV-, MMH-, and SAM-induced activation in HEK293T cells across a range of targets as quantified by flow cytometry (c) and RT-qPCR (d). (e) Multiplexed targeting of four surface-proteins in MHV-, MMH-, and SAM-expressing HEK293T cells. (f-h) Testing activation by MHV, MMH, and SAM within mouse N2A and human HeLa and HCT116 cells. (h) Quantifying off-target transcriptional perturbations induced by MHV-, MMH-, SAM-, and MCP-expressing HEK293T cells transfected with a *TTN*-targeting gRNA. On-target (*TTN*, red) and off-target (all other genes, gray) effects were evaluated. Transcriptional aberrations driven by MCP alone were evaluated by comparing gene expression in the MCP-expressing cell line to wildtype HEK293T cells transfected with a *TTN* gRNA. Data are shown as the mean of $n = 2$ independent transfections. Correlations were calculated using Pearson correlation coefficient (R). For panels a and c-h, data are normalized to a negative control line expressing MCP not fused to any protein. Black dots represent two independent transfection replicates. Significance was determined using one-way ANOVA with Sidak's multiple comparison test. For ****, ***, **, *, and ns = $p < 0.0001$, < 0.001 , < 0.01 , < 0.05 , and not significant, respectively.

3. Is it a stretch to call SAM “the gold standard CRISPR activator” (as in the Abstract), given the diversity of CRISPRa’s that are optimized for use in different contexts? Although it was highlighted in at least one published “bake off”, SAM is still not standardly used across all active laboratories in this field, in my experience. At one point in the manuscript the authors call SAM “the gold standard MCP activator” (emphasis my own), which seems more accurate to me.

We thank the reviewer for this observation. We agree that calling SAM "the gold standard CRISPR activator" overstates its universality. We will revise the manuscript to consistently describe SAM as "the gold standard MCP activator" to more accurately reflect its role in the field.

4. The key new component of MHV, VIRF2, was also identified in a recent preprint (Carosso et al., BioRxiv) that screened for activator domains to attempt to improve CRISPRa. I imagine this is an instance of independent co-discovery. Also, Ludwig et al, Cell Systems, 2023 reports a functional characterization of VIRF2 effectors and finds that overexpression of VIRF2 causes transcriptional changes in some ISGs. The present authors also cite that VIRF2 modulates human immune genes. Do these results conflict with the seemingly unperturbed RNA-seq upon MHV expression?

Thank you for your detailed and insightful comments regarding the identification and functional characterization of VIRF2. We note that our team designed our AD libraries in 2019-2021 and completed the initial screens in 2022. As such, the identification of VIRF2 in the recent preprint by Carosso et al. (BioRxiv) and the functional characterization reported by Ludwig et al. (Cell Systems, 2023), as the reviewer suggests, are instances of independent co-discovery. We acknowledge the importance of these studies and have cited them accordingly in our revised manuscript.

[LINE 605]

Furthermore, MHV and MMH offer several additional advantages over existing CRISPR activators. MHV, our most potent activator, is more compact than the MCP component of SAM by 210 bp (729 bp versus 939 bp). This reduced size could facilitate easier packaging into viral vectors for gene therapy applications. MHV also induced lower levels of toxicity than the MCP component of SAM, ranking in the bottom 9th vs. 17th percentile of toxicity, respectively, in our high-throughput screen (Supplementary Data 5) – potentially enhancing its suitability for long-term expression or use in sensitive cell types. **It is worth noting that VIRF2, a key component of MHV, has been recently uncovered in other studies, highlighting the convergent discovery of this potent AD.**

29. Ludwig, C. H. *et al.* High-throughput discovery and characterization of viral transcriptional effectors in human cells. *Cell Syst.* 14, 482-500.e8 (2023).

85. Carosso, G. A. *et al.* Discovery of hypercompact epigenetic modulators for persistent CRISPR-mediated gene activation. 2023.06.02.543492 Preprint at <https://doi.org/10.1101/2023.06.02.543492> (2024).

Ludwig *et al.* observed transcriptional changes in ISGs upon overexpression of the full VIRF2 protein, which includes its intact DNA-binding domain. However, in our study, we expressed only the AD of VIRF2, which we expect to localize specifically to target genes based on dCas9 targeting. This fundamental difference in experimental design likely explains the different cellular responses observed between our study and that of Ludwig *et al.*, as our use of VIRF2's AD alone would not be expected to broadly affect ISG transcription on its own.

5. Figure 1 legend could be more clear how the fold-change is calculated (MFI or % positive cells between two samples?). Is the normalization to dCas9-only negative control plasmid an additional step?

We agree that the description of the fold-change calculation could be more explicit. To address this:

1. We will specify that fold-change is calculated using median fluorescence intensity (MFI) of the target protein.
2. We'll clarify that normalization to the dCas9-only negative control is an additional step in the calculation.

The section of the revised legend is pasted below:

[LINE 1576]

For panels b-e, fold-change was calculated as the ratio of median fluorescence intensity (MFI) of cells transfected with each dCas9-AD fusion to the MFI of cells transfected with dCas9 alone (negative control).

6. Figure 1F – the term “specialist” sounds strong to me given the definition of being within the top 10. E.g. if an activator were top 10 in two screens, but ranked #11 in the third screen, it would seem incorrect to call it a “dual-target specialist”.

We appreciate the reviewer's observation regarding our terminology in Figure 1F. We agree that the term "specialist" may be too strong and potentially misleading given the arbitrary nature of our top 10 cutoff.

To address this, we propose the following changes:

1. Terminology revision: We will replace "dual-target specialist" with "Top ranker on two targets" and "single-target specialist" with "Top ranker on a single target". These terms better reflect the high performance/ranking of a given activator without implying strict specialization. See our changes in the text and figures pasted below.
2. Supplementary information: We will provide a supplementary file (Supplementary Data 2) with the actual rankings for each activator across all three targets, allowing readers to see the full performance spectrum.

[LINE 148]

The remaining top-performing non-VP16 domains ranked among the top 10 performers for either one or two target genes (**Figures 1b-d and 1f**).

Figure 1. (f) ADs were ranked by activity on each target, and those constituting the top 10 performers across all targets (generalist), **two targets (top rankers on two targets)**, or **one target (top rankers on a single target)** are shown. Blue, orange, and green circles designate performance on EPCAM, CXCR4, and the synthetic reporter, respectively.

[LINE 106]

To gauge activator generalizability across targets, we tested all fusions against two endogenous genes with different baseline expression levels, EPCAM and CXCR4, as well as a plasmid-borne synthetic reporter (Figure 1a, **Supplementary Data 2**).

7. Could the authors show how single domain lentiviral/MCP screen results correlate with the transfection/direct-fusion screen?

The reviewer asks about the correlation between the manual testing, performed by transfection of ADs directly fused to dCas9, and the single-domain screen, performed by lentiviral transduction of a mixed pool of MCP-fused domains. In the attached figure, we chart the L2FC from manual performance on the X axis, and the screen-score on the Y axis. We find a correlation of $R=0.26$, 0.43 , 0.46 for EPCAM, CXCR4 and the Reporter, respectively.

These correlations, while positive, indicate differences in domain performance between the two experimental setups that might be explained by the following factors. First, the manual testing was performed via transfection, which provides a much higher dose of DNA than transduction. Second, the constructs were fused to MCP instead of directly to dCas9. Third, the domains were scored in different ways, by median fluorescence in the first case and by differential enrichment in 4 bins in the second. Fourth, the transient transfection measurements were taken after 48 hours, while screens involved multiple cell passages, potentially allowing for long-term effects (e.g. cellular toxicity, transcriptional squelching, cellular adaptation to activator) to manifest.

8. For the Figure 3B schematic- Is there a 2A peptide between PuroR and MCP?

Editorial Note: Figure 3b in this Peer Review File is reproduced with permission from BioRender, Created in BioRender. Giddins, M. (2025) <https://BioRender.com/r7iqxqo>

Yes, there is a P2A peptide between the puromycin resistance gene (PuroR) and MCP in our construct. We intentionally kept the schematic high-level to ease interpretation of the overall design. However, we agree that including this detail would provide a more accurate representation of our construct. To address this, we have modified Figure 3B (shown below) to include the P2A peptide between PuroR and MCP.

Figure 3. High-throughput combinatorial screening of CRISPR activators. (a) Schematic depicting method for choosing 25 domains to combinatorially assemble and screen. (b) Schematic depicting approach for screening MCP activators. Domains were combinatorially fused downstream of MCP in all possible single, bipartite, and tripartite combinations (a representative tripartite construct, VP16-K11-MLL, is shown at top, in gray). Activators were then transduced into HEK293T cells, and cells were sorted based on their levels of target gene expression. All constructs within each sorted bin were sequenced, and these data were used to determine the potency of each activator. (c) EPCAM screen scores of all tripartite library members, along with examples of the abundances of select activators within bins 1-4. Data are shown as the mean (n = 2 independent screen replicates). (d) Correlations between EPCAM

screen scores across two biological replicates for single-domain, bipartite, and tripartite screens. Yellow dot (bipartite panel, middle) indicates performance of MCP-P65-HSF1 (MCP component of SAM). Correlations were calculated using Pearson correlation coefficient (R). (e) Correlations between EPCAM screen scores and EPCAM expression derived from manual testing of individual activators isolated from the single-domain, bipartite, and tripartite libraries. Experimental data (x-axis) are shown as the mean (n = 2 independent transduction replicates). Screening data (y-axis) are shown as the mean (n = 2 independent screen replicates). Correlations were calculated using Spearman correlation coefficient (ρ).

9. I don't think Figure 3 should be labeled "rapid" screening, as there is no information presented in the figure about the amount of time it takes to do this method. It appears to require the standard amount of time for such a combinatorial lentiviral FACS-based screen.

We thank the reviewer for their feedback regarding the labeling of Figure 3. We initially used the term "rapid" to highlight the increased efficiency of our workflow compared to traditional methods, which typically involve cloning and testing CRISPR activators one at a time. By pooling each of these steps, our approach significantly accelerates the characterization of thousands of CRISPR activators, even if the overall timeframe aligns with other lentiviral FACS-based screens. However, we agree that Figure 3 does not present a specific time metric. In response, we have revised the title to "High-throughput combinatorial screening of CRISPR activators," which more accurately reflects the essence of our approach.

10. Does the 4-bin analysis add useful information beyond a 2-bin (1 vs 4, or 1+2 vs 3+4) analysis? It could be very useful to readers if the authors can comment further on their binning strategy. Perhaps relatedly, I wonder if their library construction strategy makes the singles and doubles more highly abundant than the numerous triples, and that the triples then have lower cell coverage during the screen and readout, and this explains the higher noise in the measurements of triples. As such, it seems plausible that combining Bins1+2 and 3+4 to improve cell coverage would overall improve data quality, despite lowering bin resolution.

We appreciate the reviewer's thoughtful questions regarding our binning strategy and its impact on data quality, particularly for the tripartite constructs. Our 4-bin scoring method was designed to capture a more nuanced picture of activator performance than a simple 2-bin classification. This approach combines information about the abundance of constructs within each bin with the median fluorescence of cells sorted from those bins. By utilizing all four bins, we retain valuable information about the relationship between bins 1 and 3, or 1 and 4, which would be lost in a 2-bin analysis. To address the reviewer's suggestion, we compared our 4-bin method with a 2-bin analysis (combining bins 1+2 as low and 3+4 as high). Interestingly, we found that the 2-bin approach resulted in consistently lower replicate-to-replicate correlations across all screens. This difference was most pronounced in the bipartite and tripartite screens and most negligible in the single-domain screens (see data below). This result suggests that our 4-bin analysis is particularly beneficial for more complex, potentially noisier screens.

Regarding the concern about lower cell coverage for tripartite constructs, we acknowledge that this is a valid point. The tripartite library does indeed have lower coverage due to its larger size

and potential toxicity-driven skew. However, our analysis suggests that maintaining the 4-bin resolution provides better data quality even in this case. The improved performance of the 4-bin method for more complex screens indicates that the additional information gained outweighs the potential drawbacks of lower coverage by potentially splitting cells across more bins. That being said, we agree that this is an important consideration for future screen designs. For very large libraries where coverage is a significant concern, a 2-bin approach might be preferable.

Comparison of correlation between biological replicates upon using all four bins, as described in the paper, vs. an alternative two bin scoring method. In the two bin method, bin 2 is merged into bin 1 to produce a “low bin” and bin 3 is merged into bin 4 to produce a “high bin”. The reduced performance of this approach shows that our screens benefit from the information contained in the distribution between “high” bin 3 and the “very high” bin 4.

11. Figure 6H – Could *TTN* be labeled in Supp Fig 26A to show if CRISPRa is working? It just shows replicates, with no comparison between conditions or labeling of on/off-targets. Would MHV vs negative control be more informative about the transcriptional perturbations of overexpressing the MHV (which do sound minimal)?

We appreciate the reviewer's careful examination of Figure 6H (now Figure 7i) and Supplementary Figure 26A (now Supplementary Figure 34). We have now labeled *TTN* within Supplementary Figure 34, as suggested by the reviewer, and we have included the updated figure below. Of note, HBG1 expression is not marked in Supplementary Figure 34 (we added

this new target into our off-targeting analysis per the request of a previous reviewer) since it did meet our minimum read read cutoff.

Supplementary Figure 34. Transfection replicates show high correlations in gene expression.

(a-b) Correlation between *TTN* (a) or *HBG1* (b) gene expression ($\log_2(\text{TPM})$ (transcripts per million)) in transfection replicates of activator- or control-expressing lines transfected with a *TTN*- (a) or *HBG1*- (b) targeting gRNA. For all panels, genes showing <1 TPM (transcript per million) in either replicate of any construct in the given correlation were excluded before \log transformation. Correlations were calculated using Pearson correlation coefficient (R). **For panel A, red dot indicates gene expression of *TTN*. *HBG1* expression is not marked in panel B because it failed to meet the read cutoff.**

With regard to the reviewer's question concerning the comparison of MHV vs the negative control, we considered the condition in which cells were transfected with MCP without any fused activator to be the most appropriate negative control, since it would potentially control for any changes in transcription due to transfection and the MCP protein itself. We have added a figure (Supplementary Figure 33) displaying the comparison we believe the reviewer is suggesting: each activator vs the no MCP control ("HEK293T" on the x-axis), which shows a similar trend of minimal off-target effects of our MHV activators, should any of our readers be interested in this analysis.

Supplementary Figure 33. Activators exhibit minimal off-target activity as compared to cells not expressing MCP. Quantifying off-target transcriptional perturbations induced by MHV-, MMH-, SAM-expressing HEK293T cells transfected with (a) *TTN*- or (b) *HBG1*-targeting gRNA compared to cells absent of any MCP protein. For all panels, genes showing <1 TPM (transcript per million) in either replicate of any construct in the given correlation were excluded before log transformation. For panel b, *HBG1* gene expression was below this cut-off, preventing it from being plotted. For all panels, data are shown as the mean of n = 2 independent transfections. For all panels, correlations were calculated using Pearson correlation coefficient (R).

12. Is there a mistake in the title of Supp Figure 17 that gets the positive and negative reversed, relative to the plots?

The reviewer is correct that there is a discrepancy between the title of Supplementary Figure 17 (now Supplementary Figure 19) and the data presented in the plots. We thank the reviewer for bringing this to our attention and have changed the title of the figure to:

[LINE 2158]

Fraction of disorder-promoting residues and hydrophobicity show positive and negative correlations, respectively, with activator toxicity.

13. I believe PADDLE was exclusively trained on yeast data, which may be worth mentioning

when discussing how it performs on this dataset. Could they present a plot of PADDLE prediction vs measurement too?

As requested, we have prepared a plot comparing PADDLE predictions to our manual testing measurements, shown below. This plot shows the PADDLE score on the x-axis versus the log₂ fold-change over dCas9 control from our manual testing on the y-axis for each of our three targets (EPCAM, CXCR4 and Reporter).

Although there is a general positive trend between PADDLE scores and activation strength, the correlation is not strong, with many domains exhibiting high PADDLE scores despite showing low activation, as well as some exhibiting low PADDLE scores despite showing high activation.

The PADDLE algorithm was indeed trained entirely on data from yeast. However, the authors claim that it is fairly effective at predicting activation in other Eukaryotes. These findings reinforce the critical importance of experimental validation, especially when working across species.

PADDLE predictions vs manual testing results. PADDLE takes as input a sequence of 53 amino acids and produces as output a numerical value that is intended to correlate with the likelihood that the window contains an activation domain (AD). PADDLE was used to score domains according to the protocol laid out in Sanborn et al. Briefly, for each protein that we tested, PADDLE predictions were calculated for each 53-amino acid window, tiling the protein one amino acid at a time. These predictions were smoothed by applying a centered-rolling average to each position. For each paddle-prediction, the smoothed-prediction was calculated as the mean of up to 9 paddle-predictions, extending 4 positions in each direction, or fewer at the edges of the sequence. The score for the protein was defined as the highest value which was maintained across any 5 consecutive smoothed-predictions. Sanborn et al. recommend a score cut-off of 6 or 4 for high or medium strength ADs, respectively. Sequences shorter than 53 amino acids were handled by embedding in a sequence of “neutral” amino acids, as recommended by Sanborn.

14. Could the authors expand on the 'built in screening controls' from Note 1 more? They seem interesting and useful to readers. How do supernatant controls make it all the way through the cell sorting step into the final libraries? I am not familiar with supernatant controls; are they described elsewhere? Would the authors suggest not trusting screen data with a certain score from these controls?

We appreciate the reviewer's interest in our built-in screening controls. There were three controls we included. 1) an "MCP-mutant" control, which is in a lentiviral vector that has the puromycin resistance gene, but does not have any AD fused to MCP; 2) a "Puro-mutant" control, which is in a lentiviral vector but does not produce puromycin resistance; 3) a "Supernatant-mutant" control, which is not able to be packaged into lentivirus due to absence of viral packaging elements in the construct. The Puro-mutant control allows us to detect whether cells were either multiply infected with several viral constructs or whether there were crossovers between barcodes during packaging (a major concern when screening barcoded libraries). The supernatant control allows us to detect the presence of plasmid DNA that remains in the harvested lentiviral supernatant and can potentially confound our analysis with false signal. All controls were spiked into the viral barcoded library at an equal ratio at the time of transfecting our activators to generate lentivirus.

Across screens, Puro-mutant control and Supernatant-mutant barcodes showed significantly fewer reads than mutant MCP control barcodes. For the single-domain, bipartite, and tripartite screens they showed between 42.5%-90.1% and 80.8%-99.9% fewer reads than the mutant MCP control barcodes, respectively (data shown below). Given that Puro-mutant and Supernatant-mutant controls were spiked into each library at the same concentration as MCP mutant controls, these results suggest that only a small proportion of NGS reads derive from multi-copy + crossover events (Puro-mutant signal) and plasmid contamination from lentivirus preparation (Supernatant-mutant signal). Interestingly, the bipartite data seemed to show more signal from the Puro-mutant and Supernatant controls, but these reads did not significantly influence resultant data, based on downstream quality control (biological replat correlations and experimental validation of screen-derived activation). With regards to how the Supernatant-control barcodes make it to the final sequencing libraries, we were also surprised that any of these sequences make it to this point. It is unclear to us whether they derive from trace amounts of plasmid that remain in the system, rare crossovers that occur when the plasmid libraries are delivered to cells, or from amplicon contamination during library preparation.

The dramatic differences in the abundances of the various control barcodes in the unsorted pool of transduced cells for the tripartite screens are shown in the graphs below.

Abundance of negative control barcodes (“MCP-mutant” – blue, “Puro-mutant” – orange, “Supernatant-mutant – gray”) in tripartite screens.

Based on the reviewer’s suggestion, we have included the above graph in Supplementary Note 1 to improve clarity.

The MCP-mutant control constructs enabled us to perform an additional check, since these constructs contain an MCP protein not fused to any activator. Given this design, we expect the MCP-mutant controls to rank towards the bottom of all activators in the screen. When we look at their performance across the three targets in each of the screens, the MCP-mutant ranked, on average, 25.3 out of 26 in the single-domain screen, 622 out of 626 in the bipartite screen, and 817.33 out of 838 (of the set of constructs ranked across all three targets) in the tripartite screen. Our findings that this construct, which contains an inactive MCP, ranks towards the bottom of all constructs across screens highlights the accuracy of our high-throughput approach for scoring activators.

With regards to the reviewer’s question about control thresholds, there was no threshold for the abundance of our three spiked in controls that was used in isolation to determine a screen’s quality. Instead, in deciding whether a screen was properly performed, we looked at three main features of the screening data: 1) relative abundance of our three spiked in controls, 2) correlations between independent biological replicates, 3) correlation between manual testing of individual library members and their performance as determined by the screen. We believe that this holistic look at our data provides us with better insight into the success of our screens than only examining our spiked in controls.

REVIEWER COMMENTS

Color coding for this document: Reviewer comments in **BLACK**, our responses in **BLUE**, words and edits in the manuscript are in **PURPLE**. Of note, some responses were refined for clarity with the aid of ChatGPT-4o.

Reviewer #1 (Remarks to the Author):

The authors have reasonably addressed my comments.

Minor :

In the Supplementary Figures 32, 33, and 34, up/down-regulated genes should be labeled as distinguishing points, including *HBG1*.

We thank the reviewer for this helpful suggestion. In response, we have updated Supplementary Figures 32 and 33 to label differentially expressed genes in black. For these analyses, we define differential expression as genes with $|\log_2 \text{fold change}| > 2$ and an adjusted p-value of < 0.05 , and we apply a TPM threshold of > 1 to restrict the analysis to genes with adequate expression levels.

Supplementary Figure 32 compares gene expression between activator-expressing cells (MHV, MMH, SAM) and MCP alone upon targeting *HBG1*. *HBG1* is not labeled in this panel (and is not plotted at all) because its expression was below the TPM of 1 threshold in all conditions.

Supplementary Figure 33 includes two panels comparing activator-expressing cells to HEK293T cells expressing dCas9 and a guide but no MCP, using either a *TTN*- or *HBG1*-targeting guide (panels A and B, respectively). As in Supplementary Figure 32, *HBG1* is not shown in panel B due to low expression. In contrast, *TTN* is labeled in panel A.

We have also updated Figure 7h (which parallels Supplementary Figure 32 but targets *TTN*) to use the same differential expression labeling.

We did not apply differential expression labeling to Supplementary Figure 34, as these panels show replicate-to-replicate comparisons within the same activator condition and were intended to demonstrate technical reproducibility rather than highlight activator-induced transcriptional changes.

The updated panels and corresponding figure legends are pasted below.

h

Figure 7. (h) Quantifying off-target transcriptional perturbations in MHV-, MMH-, SAM-, and MCP-expressing HEK293T cells transfected with a *TTN*-targeting gRNA. Transcriptional aberrations driven by MCP alone were evaluated by comparing gene expression in the MCP-expressing cell line to wildtype HEK293T cells transfected with a *TTN* gRNA. Data are shown as the mean of $n = 2$ independent transfections. ***TTN* (on-target gene) is shown in red, differentially expressed genes are shown in black (defined as $|\log_2$ fold change > 2 and adjusted $p < 0.05$), and all other genes are shown in gray.** Correlations were calculated using Pearson correlation coefficient (R).

Supplementary Figure 32. Activators exhibit minimal off-target activity as compared to MCP not fused to any protein upon targeting *HBG1*. Quantifying off-target transcriptional perturbations in MHV-, MMH-, SAM-, and MCP-expressing HEK293T cells transfected with an *HBG1*-targeting gRNA. *HBG1* gene expression was below the < 1 TPM cutoff in all conditions and was therefore

excluded from the plots. Transcriptional aberrations driven by MCP alone were evaluated by comparing gene expression in the MCP-expressing cell line to wildtype HEK293T cells transfected with an *HBG1* gRNA. Data are shown as the mean of $n = 2$ independent transfections. **Differentially expressed genes are shown in black (defined as $|\log_2 \text{fold change}| > 2$ and adjusted $p < 0.05$), and all other genes are shown in gray.** Correlations were calculated using Pearson correlation coefficient (R).

Supplementary Figure 33. Activators exhibit minimal off-target activity as compared to cells not expressing MCP. Quantifying off-target transcriptional perturbations in MHV-, MMH-, SAM-expressing HEK293T cells transfected with a (a) *TTN*- or (b) *HBG1*-targeting gRNA compared to cells absent of any MCP protein. For all panels, genes showing < 1 TPM (transcript per million) in either replicate of any construct in the given correlation were excluded before log transformation. For panel b, *HBG1* gene expression was below this cut-off and was therefore excluded from the plots. For all panels, data are shown as the mean of $n = 2$ independent transfections. **For panel A, *TTN* (on-target gene) is shown in red. For all panels, differentially expressed genes are shown in black (defined as $|\log_2 \text{fold change}| > 2$ and adjusted $p < 0.05$), and all other genes are shown in gray. No genes met differential expression criteria in panel B.** Correlations were calculated using Pearson correlation coefficient (R).

Referring to the recent Nature Methods-published tool CRISPR DREAM (Mahuta et al. 2023), the author used its data to justify the current analysis. But I haven't seen similar data, such as TBX5 et al. Please indicate which figure in CRISPR DREAM these data came from. Also, it'd be better if the author could make a statistical graph to clearly show the activity comparison with SAM.

We appreciate the request for clarification on the sources of our comparative data from the CRISPR DREAM (Mahuta et al. 2023) paper. The specific figure from which we extracted TBX5 data is Supplementary Figure 11, panel e (pasted below). In this figure, CRISPR DREAM's activation levels for TBX5, alongside other targets, were presented on a logarithmic scale (10^0 - 10^1 - 10^2 range). To maintain consistency with our own activation data format, which is plotted on a linear scale, we de-logged the CRISPR DREAM values before making direct comparisons (also re-pasted below).

More broadly, the comparative data we selected from the CRISPR DREAM work – including TBX5 – was chosen from both the main and supplementary figures to represent a range of differences with SAM, including cases where CRISPR DREAM was much better, moderately better, and not better than SAM. This selection was meant to be representative of the larger dataset across the entire Mahuta et al. 2023 paper.

Supplementary Figure 11 from Mahuta et al. 2023:

[Figure Redacted]

Our comparison figure (from last round of reviews):

We also appreciate the reviewer's suggestion to provide a graph comparing the activity of our new activators with SAM across all targets. In response, we have included a new panel (Figure 7g, pasted below) summarizing the comparative outcomes across all targets and conditions. We have also generated summary visualizations for all tested protein and RNA targets across all cell lines used (added to Supplementary Figures 28, 29, 30, and 31, pasted below). These graphs allow for a side-by-side comparison of MHV, MMH, and SAM across all targets.

Figure 7. (g) Summary of activator performance across targets and cell types. Each circle represents the performance of MHV or MMH relative to SAM for a specific target gene in a given cell line, based on either protein or RNA expression. Green circles indicate targets where the activator is significantly stronger than SAM, orange circles indicate targets where the activator is significantly weaker than SAM, and grey circles indicate no significant difference.

Supplementary Figure 28. (c-d) Summary visualization of all tested protein (c) or RNA (d) targets across MHV-, MMH-, and SAM-expressing HEK293T cells. Grouped data are shown for visualization, but statistical comparisons for each individual target gene were performed using one-way ANOVA.

Supplementary Figure 29. (c-d) Summary visualization of all tested protein (c) or RNA (d) targets across MHV-, MMH-, and SAM-expressing HeLa cells. Grouped data are shown for visualization, but statistical comparisons for each individual target gene were performed using one-way ANOVA.

Supplementary Figure 30. (b) Summary visualization of all tested targets across MHV-, MMH-, and SAM-expressing N2A cells. Grouped data are shown for visualization, but statistical comparisons for each individual target gene were performed using one-way ANOVA.

Supplementary Figure 31. (b) Summary visualization of all tested targets across MHV-, MMH-, and SAM-expressing HCT116 cells. Grouped data are shown for visualization, but statistical comparisons for each individual target gene were performed using one-way ANOVA.

Reviewer #2 (Remarks to the Author):

The authors have responded with experiments, analysis and text changes to my review. My concerns related to lentiviral recombination are fully satisfied which was a major concern. My concern related to the explanation of validation data in Figure 3E remains a concern although this can be addressed as noted below. The expansion of text and data on toxicity of activators as well as AD order and context are strengths of the manuscript and will be highly useful.

We thank the reviewer for their thoughtful analysis of our responses to the first round of comments. We are pleased to hear that our experiments and explanation addressing the issue of lentiviral recombination were satisfactory. We have incorporated updates to address the reviewer's remaining concerns about the validation data in Figure 3E. Finally, we appreciate the reviewer's recognition of the expanded analysis on activator toxicity, activation domain order, and domain context. We're encouraged that these findings are seen as strengths and potentially useful to the field.

I remain unconvinced that MHV and MMH (or DREAM) are a sufficiently big advance (especially across cell types as further discussed below) to convince the community to move away from SAM, VPR or other more widely use activator constructs. Specifically, MMH is similar or sometimes worse than SAM. MHV is overall not significantly more active in N2A or HCT116 across comparisons. MHV is more active than SAM in HEK293 and Hela at about half of tested examples (many of which may not be biologically meaningful differences) so there remains unexplained activator cell type specificity and overall MHV is more active than SAM at only 19

out of 48 comparisons across cell types. If the authors really want to show MHV is better than SAM they should do a genome-scale screen and show they learn something new with MHV that is missed by the SAM approach. However that said it is interesting that such a large effort has not yielded an activator that is remarkably better than SAM.

We agree that MHV is not universally superior to SAM in every scenario tested, and we appreciate the opportunity to clarify. We have revised the text to more accurately reflect its context-dependent performance and to avoid overstating its generalizability. That said, our data indicate that MHV is rarely outperformed by SAM. Across 48 direct comparisons spanning multiple gene targets and cell types, MHV showed higher activity than SAM in 18 instances, while SAM was more active in only 1 instance. The remaining comparisons showed comparable activity or a trend favoring MHV.

To better visualize this overall trend, we have now included a new panel (Figure 7g, pasted below) summarizing the comparative outcomes across all targets and conditions. This figure shows that while MHV does not consistently outperform SAM across the board, it is almost never worse, and sometimes provides an improvement. We believe this outcome supports the value of MHV as a robust alternative.

We have also updated the Discussion section (pasted below) accordingly to reflect these points.

Figure 7. (g) Summary of activator performance across targets and cell types. Each circle represents the performance of MHV or MMH relative to SAM for a specific target gene in a given cell line, based on either protein or RNA expression. Green circles indicate targets where the activator is significantly stronger than SAM, orange circles indicate targets where the activator is significantly weaker than SAM, and gray circles indicate no significant difference.

[LINE 606]

“Although previous CRISPR activators are deployed today in broad-ranging in vitro and in vivo applications, they display inconsistent performance across targets and cellular contexts, limiting their utility. Our high-throughput screens elucidated two novel activators, MHV and MMH. **While MHV is not universally superior to SAM, it outperformed SAM in 18 out of 48**

comparisons across multiple targets and cell types, was outperformed in only one case, and showed comparable or trending improvements in the remainder. Importantly, our data point to a context-dependent component to activator performance, where gains by MHV and MMH are often substantial but not uniform across all targets or cell types. Understanding the determinants of this variability remains an important direction for future work.”

In addition, it is informative that even after exhaustive optimization efforts, the best new tool (MHV) is only marginally better than the state-of-the-art (SAM) rather than overwhelmingly superior. These results align with recent tool development efforts and underscore the challenges in this space given the current high baseline. We believe this insight is itself meaningful, as it sets realistic expectations and informs future tool development. We have revised the Discussion to emphasize this perspective:

[LINE 629]

“These features, combined with their enhanced activation potential, position MHV and MMH as promising tools for a wide range of applications. **That said, the incremental nature of these gains – despite extensive screening – highlights the inherent difficulty of surpassing well-optimized systems like SAM. Our results indicate that progress in this space may come through steady, context-aware refinements. We view this as a valuable insight that helps recalibrate expectations and inform future tool development.**”

Finally, beyond performance metrics, our study offers several other important contributions that strengthen its significance. In particular:

1. We performed an extensive screen of candidate activator designs that represents one of the most comprehensive surveys of CRISPR activation domain variants to date, yielding a wealth of data on the design features that enhance gene activation. Importantly, this screen included many domains that would not be covered by traditional fixed-length tiling approaches, due to their length exceeding 80 AA.
2. We conducted comprehensive toxicity assessments to ensure that increased activation did not come at the cost of cellular fitness. These experiments showed that MHV produces lower toxicity than the MCP component of SAM. More broadly, our findings reveal that activator-induced toxicity is both more pervasive and context-independent than previously appreciated, underscoring the importance of incorporating toxicity screening into future activator design workflows.
3. We developed a user-friendly cloning system to rapidly construct and functionally evaluate chimeric proteins. This method is broadly adaptable and can be easily applied to the design and optimization of other multi-domain fusion proteins, extending its utility beyond CRISPR activators.

- In addition to developing new tools, our study yielded valuable biological insights into CRISPR activation and toxicity. For example, we found that activation is governed by domain spatial arrangement and the 3D configuration of bound cofactors, while toxicity manifests largely independent of these features. These findings help disentangle the mechanisms underlying activation versus toxicity and provide a framework for future activator design.

Related to Figure 1:

Please show a histogram of domain amino acid length for the 230 domain library to illustrate how the authors library would compare against more the more common approach of using 80AA tiles. Please color also

We thank the reviewer for this suggestion. To illustrate how our domain library compares to more traditional tiling approaches, we have added a new panel (Figure 1b) showing a histogram of the amino acid lengths for all 230 domains tested. The plot includes a reference line at 80 amino acids and is colored by whether each domain was classified as a "hit" or not.

Supplementary Figure 1. Diversity of individually tested domains. (a) All individually tested parts were clustered at 50% sequence identity using uclust. Families of related activators which were clustered into multiple sets were grouped and colored similarly. Clusters are labeled with their centroid member's name followed by the size of the cluster. (b) **Distribution of lengths of tested domains, separated between hits and non-hits. Line at 80 amino acids shows the length associated with commonly used 80-AA tiling methods.**

Related to Figure 3:

In Figure 3D the authors now highlight that the MCP components of the SAM system (MCP-P65-HSF1) are a very top hit in the screen (top 2) which undermines the notion that this manuscript's discovery effort will lead to new useful tools.

We appreciate the reviewer's point. While it is true that MPH performed well in the bipartite EPCAM screen, ranking in the top 1% of constructs, it proved less dominant on other targets, ranking in the 90.4th percentile on CXCR4 and 89.1st percentile on the synthetic reporter (see below figure). We interpret these results as consistent with the fact that MPH is a well-characterized and highly optimized activator. Its strong – but not uniformly top – performance across targets serves as a positive control that validates the screen, while also underscoring the value of our screening approach in identifying activators like MHV and MMH that may offer improvements. Finally, our manual validation data in Figure 7 demonstrate that MHV frequently outperforms SAM across a range of targets and cell types, supporting its utility as a high-performing alternative.

Bipartite screen scores across replicates for EPCAM, CXCR4, and the synthetic reporter. The performance of MCP-P65-HSF1, the MCP component of the SAM system, is highlighted in yellow.

Figure 3E as commented on in my first round of comments it remains confusing that the authors picked random hits from the screens. The authors state “To experimentally validate our screening results, we isolated 20-30 library members from each of our single-domain, bipartite, and tripartite libraries and tested them independently via transduction against our screening targets.” Perhaps stating “To experimentally validate our screening results, we randomly isolated 20-30 library members from each of our single-domain, bipartite, and tripartite libraries only a subset of which are strong hits and tested them independently via transduction against our screening targets.” Please clarify.

In my second round of comments I reviewed the manuscript without first reading the response to review and wrote the whole comment below on Figure 3E so this section is clearly confusing.

In Figure 3E the authors validate a number of hit constructs from their screens. The authors show a fairly good correlation between screen and validation (0.54-0.83) but closer inspection of the data shows that many of the hits are inactive in the validation experiments. This is especially true for the bipartite hit validation with a stripe of hits at zero on the x-axis. Please plot the x-axis as log₂ fold change over a negative control construct for all validation data instead of or in addition to the current plots (Figure 3E and Supplementary Figure 7). In Figure 1 the authors define hits as producing “at least two-fold activation on EPCAM, CXCR4, and the synthetic reporter, respectively”. It seems to me that any bi partite or tri partite construct that doesn’t meet this criteria is a false positive hit from the screen despite the apparent correlations. Please define your false positive rate given these issues raised.

As you can see I remained confused. Perhaps noting with different colors which tested constructs would have been called hits will clarify this figure?

We appreciate the reviewer’s continued attention to Figure 3E and the opportunity to clarify its purpose more explicitly.

These experiments were **not intended to validate screen-identified hits**. Instead, we designed this analysis to assess how well the screen ranked constructs **across the full range of observed activities**. To do this, we selected 20–30 constructs from each of our single-domain, bipartite, and tripartite libraries. Since the single-domain library contains only 25 members, we tested all of them. For the larger bipartite and tripartite libraries, we randomly selected constructs spanning low, intermediate, and high screen scores. If we had only selected top hits for follow-up, we would not have been able to evaluate the screen’s ability to **accurately rank activators from weakest to strongest** – which was the purpose of this experiment.

This distinction is visualized in the below scatter plot showing EPCAM bipartite screen scores across replicates, where constructs used in the validation set (orange) are plotted alongside screen-defined top hits (green). While hits are clustered in the high range, most validation constructs fall in the mid-to-lower range of the screen distribution and were not expected to show ultra-strong activation when tested individually.

EPCAM scores for the bipartite screen across replicates, where top hits taken forward for additional workup are shown in green, and constructs used to assess the efficacy of our screening approach are shown in orange.

The ≥ 2 -fold activation threshold referenced in Figure 1 was used during our initial one-by-one testing of 230 individual activation domains, specifically to define a set of “active” domains for downstream analysis of biochemical features associated with activation (e.g., charge, disorder, hydrophobicity). This threshold is not applicable to interpreting screen scores, which reflect relative barcode enrichment rather than direct fold-changes in expression.

Regarding the “stripe” of low-performing constructs in the bipartite validation experiment (shown below): while many of these constructs were drawn from the lower end of the screen activity distribution and were not expected to show strong activation, we acknowledge that some had moderate screen scores but still performed poorly in individual validation. These discrepancies likely arise from inherent differences between the screening and validation formats, including pooled versus one-by-one transduction, variation in construct expression levels, or differences in assay dynamic range and sensitivity. Importantly, the overall correlation between screen score and experimental activity remains strong (e.g., $R = 0.79$, $p < 0.0001$ for the bipartite EPCAM-targeting screen), indicating that the screen effectively ranks activators by performance, even if exact activity levels do not always match between formats.

Figure 3. (e) Correlations between EPCAM screen scores and EPCAM expression derived from manual testing of individual activators isolated from the bipartite library. Experimental data (x-axis) are shown as the mean ($n = 2$ independent transduction replicates). Screening data (y-axis) are shown as the mean ($n = 2$ independent screen replicates). Correlations were calculated using Spearman correlation coefficient (ρ).

Finally, for a more direct evaluation of false positive rate, one might rely on Supplementary Figure 26, pasted below, where we individually validated every bipartite and tripartite “hit” from the screen. In the case of bipartite hits tested on CXCR4, for example, all constructs tested showed at least a 2.5-fold increase in target expression over MCP alone, and 17 out of 25 constructs exceeded a 6-fold increase – demonstrating the strong predictive power of the screen for identifying high-performing constructs.

Supplementary Figure 26. (a) Manual validation of 25 bipartite hits. Constructs were stably integrated into HEK293T cells and transfected with a mixture of gRNAs targeting four surface-protein genes, CD45, EPCAM, EGFR, and CXCR4, followed by flow cytometry to measure activation. Each activator is annotated based on the species it is derived from or its functionality as a PF, human ADs (white dots), viral ADs (gray dots), and PFs (black dots). Positions within the MCP fusion (P1-P3) are shown, where P1 represents the position most proximal to MCP.

We have revised the text to better distinguish the intent of Figure 3E from that of a hit validation experiment.

[LINE 248]

“To evaluate the fidelity of our screen in ranking activators across a broad range of activity levels, we tested all 25 constructs from our single-domain library, and randomly selected 20–30 constructs from each of our bipartite and tripartite libraries – spanning low, intermediate, and high screen scores – via one-by-one transduction against the original screening targets.”

Related to Figure 6:

Did the authors include a dCas9 alone, MCP alone or dCas9 + MCP controls or any control here to help with interpretation of which proteins are binding to dCas9 + MCP + linkers and which are binding to various activation domains? I ask because if the vast majority of these human proteins were to bind to dCas9 + MCP + linker protein interfaces then this would cause the results to look more correlated than expected across different activator fusion proteins. If the authors don't have such controls this experiment is very hard to interpret especially as protein-protein interactions can stabilize or destabilize fusion proteins. Please include such controls.

We appreciate the reviewer's comment regarding the importance of appropriate controls in the interpretation of our AP-MS data. We agree that distinguishing interactions specific to activation domains from those associated with the dCas9-MCP scaffold is critical for meaningful analysis.

To that end, we included an MCP-only control (not fused to any activation domains) in all AP-MS experiments and analyses. Furthermore, all conditions, including this control condition, were performed in cells expressing dCas9 and a CXCR4-targeting gRNA. All reported protein

enrichments are calculated relative to the MCP-only control condition, which accounts for non-specific interactions with the MCP scaffold, and general components of the dCas9–MCP system. This normalization enables us to focus specifically on proteins that are enriched due to the presence and composition of the fused activation domains.

We have clarified this point in the Methods, pasted below.

[LINE 1014]

“Twelve constructs exhibiting a range of activation and toxicity scores, along with an unfused MCP plasmid, were each appended with a 3X FLAG-tag. Each FLAG-tagged construct was transfected into HEK293T cells **expressing dCas9 and a CXCR-targeting gRNA** in quadruplicate using FuGENE HD (Promega) in 100 mm dishes.”

Do the results in Figure 6 relate to this experiment being a transient transfection assay? Transient transfection assays result in vastly higher protein levels than are generally relevant to cellular proteins with natural activation domains.

We agree with the reviewer that transient transfection can result in higher protein expression levels than what is typically observed for endogenous proteins with natural activation domains, and we acknowledge that overexpression may promote interactions that would not occur under physiological conditions. However, we believe that the experimental context is appropriate given that these are synthetic transcriptional activators, which are inherently designed to be used as exogenous tools – often delivered by transient transfection, viral vectors, or other high-expression systems. We agree with the reviewer that expression level is an important variable that future work could explore in greater detail – particularly in relation to dose-dependent toxicity or tuning of activity.

I would recommend that the authors validate their mass-spec data with co-IP western data for constructs expressed via transient transfection and lentiviral expression so the community has a sense of validation and some more constructive context for this toxicity mass-spec analysis.

We appreciate the reviewer’s suggestion and agree that validating mass spectrometry findings is an important consideration. However, our AP-MS experiments were not designed to assess specific activator–cofactor interactions, but rather to identify broad, reproducible binding patterns associated with activation and toxicity. Our conclusions focus on the principle that cofactor binding alone is sufficient for toxicity, whereas activation requires that cofactors bind in precise 3D configurations – rather than on any individual protein–protein interaction. In this context, we believe that validating a small number of specific interactions via co-IP would not strengthen the broader mechanistic claims we are making.

Several internal controls support the reliability of the AP-MS results. In particular, we see a strong correlation in binding between constructs that share the same domains (see Supplementary Figure 24 below). Bipartite and tripartite constructs containing the same domains exhibit correlations ranging from $R=0.92$ to $R=0.99$ across their cofactor binding

profiles. In contrast, the average correlation between activators that do not contain the same domains is roughly $R=0.75$. These results strongly suggest that our AP-MS data reflect reproducible, composition-driven interaction profiles, not noise or artifacts of expression.

Supplementary Figure 24. Activators composed of the same domains or that display similar behavior in cells show high correlations in binding partner enrichment. Pairwise comparisons of protein binding profiles across all bipartite and tripartite activator constructs. The grid layout shows comparisons between constructs labeled on the diagonal. Each off-diagonal plot (left of diagonal) and R value (right of diagonal) compares the construct named in its row (x-axis) against the construct named in its column (y-axis). Dots represent individual proteins' \log_2 (enrichment in binding over the unfused MCP control) for the two constructs being compared. Only proteins detected in all conditions are shown. Histograms along the diagonal show the distribution of \log_2 (enrichment in binding over the unfused MCP control) for each construct. Correlations were calculated using Pearson correlation coefficient (R).

What proteins do SAM, MHV and MMH bind? I would add a figure and additional discussion on this to the manuscript. It will scare the community to highlight co-factor binding toxicity without discussing this in detail for these key highlighted constructs. Validating that SAM which is widely used and also MHV and MMH does or does not bind to proteins that are associated with toxicity is a really key part of this new data section.

We appreciate the reviewer's interest in the AP-MS data and agree that understanding specific cofactor binding patterns will be an important direction for future studies. In this work, our goal

was to identify general trends in activator behavior across a large number of constructs, rather than to deeply profile individual tools.

Our AP-MS analysis revealed a general trend wherein binding to certain proteins (no matter the 3D configuration) – spanning diverse cellular pathways – is associated with toxicity but not activation, suggesting that these phenotypes are governed by distinct molecular mechanisms. To our knowledge, this represents the first systematic investigation of CRISPRa-associated cellular toxicity, and we believe it lays important groundwork for future efforts to understand this topic.

To reassure the community, we note that MHV and MMH were included in our AP-MS studies and fall within the non-toxic group in Figure 6e, which shows significantly lower binding to proteins associated with toxicity. We have now clarified this in the figure legend, pasted below.

[LINE 1715]

“e) Log₂(fold-change) in protein binding for toxic (red) and non-toxic (blue) activators. Proteins are grouped by cellular function. The significance of the difference in values between toxic and non-toxic activators was assessed via unpaired two-tailed t-test for each protein. Only interactions with $p < 0.0001$ are shown. **Two top-performing, low-toxicity activators, MHV and MMV (described below), are included in the non-toxic group.**”

In addition, we have provided a re-worked version of Figure 6e for the reviewer in which MHV and MMH are highlighted separately. We are not including this revised version in the final manuscript, as MHV and MMH have not yet been introduced at this point.

$\text{Log}_2(\text{fold-change})$ in protein binding for toxic activators (red), non-toxic activators (blue), and MHV and MMH (yellow). Proteins are grouped by cellular function. The significance of the difference in values between toxic and non-toxic activators was assessed via unpaired two-tailed t-test for each protein. Only interactions with $p < 0.0001$ are shown. Two top-performing, low-toxicity activators, MHV and MMV (described below), are included in the non-toxic group.”

Furthermore, within our high-throughput screen-based toxicity assessment (which was subsequently validated using one-by-one experimental testing of cellular fitness levels), both MHV and MMH exhibited relatively low toxicity scores – particularly MHV, which ranked among the least toxic activators tested (bottom 9th percentile). While we did not include SAM in our AP-MS panel, its MCP component (MPH) also yielded a relatively low toxicity score (bottom 17th percentile) in our high-throughput screen-based assessment.

Finally, we have made the full AP-MS dataset available in the supplementary materials to support further exploration of individual cofactor interactions.

How many of the proteins enriched in this mass-spec analysis are canonically localized within an lipid membrane enclosed organelle? I ask because it is surprising to see such an enrichment of mitochondrial proteins. If proteins that are localized inside the ER, golgi or mitochondria are enriched in these mass spec data it would argue that some fraction of these results are artifacts. Understanding this is important for interpretation of the authors’s claims. I understand that some of the proteins identified in the mass-spec have regions of the protein exposed to the cytosol and thus could be reasonably expected to interact with activation domains localized to the cytoplasm/nucleus.

The reviewer raises a valid concern: most mitochondrial proteins are localized within membrane-enclosed compartments and are not typically accessible to synthetic activators. However, we believe the enrichment of mitochondrial proteins observed in our AP-MS data is more likely to reflect the biological effects of toxicity, rather than technical artifact. Mitochondrial membrane integrity is known to be compromised during cellular stress or early stages of cell death, which could expose proteins normally contained within the mitochondria to the cytoplasm

and allow for interactions that would not occur under normal conditions. Importantly, this enrichment is observed specifically in toxic constructs, not across all activators, and our AP-MS experiments were performed across four biological replicates, making systematic artifacts unlikely. Taken together, we interpret the mitochondrial enrichment as a biologically meaningful consequence of toxicity, rather than an artifact of the experimental workflow.

Related to Figure 7

I believe the authors have a mistake in the text related to Figure 7c-d and Supplemental Figure 28. They state MHV is significantly more active than SAM for 11/17 genes but it is actually 10/17 as 6 genes show no significant difference between MHV and SAM and 1 gene (CXCR4) shows SAM is more active.

We thank the reviewer for catching this mistake. The reviewer is correct – CXCR4 is the one target for which SAM outperforms MHV, and should not have been counted among the targets with improved or comparable activation. We have updated the text accordingly.

[LINE 480]

“Each activator was tested against 17 targets in HEK293T cells – five surface proteins (CD2, CD45, EPCAM, EGFR, and CXCR4) using a protein-based (flow cytometry) readout and 12 non-surface proteins (*TTN*, *HBG1*, *RHOXF2*, *NEUROD1*, *MIAT*, *ACTC1*, *ASCL1*, *IL1RN*, *IL1B*, *ZFP42*, *LIN28A*, and *IL1R2*) using an RNA-based (RT-qPCR) readout. **MHV showed up to 11.6-fold increased activation over SAM against 10 of the 17 target genes** (Figures 7c-d, Supplementary Figure 28; $p < 0.05$).”

In Supplemental Figure 29, for ITGAV it appears each of the activators is actually a repressor with around 50% knockdown? Please clarify. If all constructs tested are repressive this should be noted or removed from the claims that call MHV as more active than SAM as an activator so that comparison would be 9 out of 31 rather than 10 out of 32.

We thank the reviewer for pointing this out. The reviewer is correct – all constructs tested at the ITGAV locus resulted in reduced expression, including those fused to activators. We suspect this reflects a case where targeting dCas9 to the locus is inherently repressive, and the activators are able to partially overcome this repression to varying degrees. However, as this does not constitute activation in the strict sense, we agree that it should not be included in our target panel.

We have removed ITGAV from Supplementary Figure 30 (pasted below) and have updated the relevant text to reflect these corrected counts.

[LINE 506]

“Like in HEK293T cells, MHV outperformed SAM in HeLa cells across several instances, showing significant improvement in 5 out of 12 comparisons, while MMH showed improved activity in 1 out of 12 comparisons (**Figure 7g and Supplementary Figure 29**; $p < 0.05$ for all increases). In

contrast, in HCT116 and N2A cells, both MHV and MMH performed more similarly to SAM, with MHV outperforming SAM against only 2 out of 12 and 1 out of 7 targets, respectively (**Figure 7f-g and Supplementary Figure 30 and 31**; $p < 0.05$ for all increases). Several additional targets across all cell lines exhibited trends of improved activation by MHV or MMH relative to SAM that approached, but did not reach, statistical significance.”

Supplementary Figure 30. MHV performs similarly to or better than SAM against multiple targets in N2A cells. (a) MHV-, MMH-, and SAM-expressing N2A cells were tested against additional non-surface protein targets. Constructs were integrated into mouse N2A cells and transfected with a species-specific gRNA. Activation was quantified via RT-qPCR. (b) Summary visualization of all tested targets across MHV-, MMH-, and SAM-expressing N2A cells. Grouped data are shown for visualization, but statistical comparisons for each individual target gene were performed using one-way ANOVA. For all panels, activator-produced fold-changes in target expression were normalized to a negative control line expressing MCP not fused to any protein. Black dots depict performance of each of two independent transfection replicates. Significance was determined

using one-way ANOVA with Sidak's multiple comparison test. For **, *, and ns = $p < 0.01$, < 0.05 , and not significant, respectively.

Related to Figure 7 and Supplementary Figures 28-31: I believe a more accurate way to describe the data is MHV and SAM are roughly equivalently active in N2A cells (MHV is more active in 1 out of 7 comparisons once ITGAV is excluded) and HCT116 cells (MHV is more active in 2 out of 12 comparisons) while MHV is somewhat more active than SAM in HEK293 (10 of 17 comparisons) and HeLa cells (6 out of 12 comparisons) which highlights that there is a cell type specific aspect of activators that is not fully explored here. This could be a future direction commented on in the discussion.

We thank the reviewer for this helpful suggestion. We agree that MHV's performance varies across cell types and that its relative activity compared to SAM is most pronounced in HEK293T and HeLa cells, while in N2A and HCT116 cells, MHV and SAM show more comparable activity. We have revised the text accordingly to more accurately describe these trends. We also appreciate the reviewer's suggestion to highlight this cell type-specific aspect of activator performance as a potential direction for future work, and we have now added this point to the Discussion.

[LINE 506]

"Like in HEK293T cells, MHV outperformed SAM in HeLa cells across several instances, showing significant improvement in 5 out of 12 comparisons, while MMH showed improved activity in 1 out of 12 comparisons (**Figure 7g and Supplementary Figure 29**; $p < 0.05$ for all increases). In contrast, in HCT116 and N2A cells, both MHV and MMH performed more similarly to SAM, with MHV outperforming SAM against only 2 out of 12 and 1 out of 7 targets, respectively (**Figure 7f-g and Supplementary Figure 30 and 31**; $p < 0.05$ for all increases). Several additional targets across all cell lines exhibited trends of improved activation by MHV or MMH relative to SAM that approached, but did not reach, statistical significance."

[LINE 608]

"While MHV is not universally superior to SAM, it outperformed SAM in 18 out of 48 comparisons across multiple targets and cell types, was outperformed in only one case, and showed comparable or trending improvements in the remainder. Importantly, our data point to a context-dependent component to activator performance, where gains by MHV and MMH are often substantial but not uniform across all targets or cell types. Understanding the determinants of this variability remains an important direction for future work."

Overall I would temper expectations for MHV and MMH. For MHV which is the more active construct the authors show that only 19 out of 48 comparisons show MHV is somewhat more active than SAM across genes and cell types.

We thank the reviewer for this suggestion. We believe the above changes have effectively achieved this tempering.

Reviewer #3 (Remarks to the Author):

I commend the authors on a thorough revision, wherein they performed further analysis of domain order, performed a new mini screen that shows toxicity is similar across cell types, showed MHV>SAM across more target genes, analyzed PADDLE scores, clarified figure labels, better explained UMIs and spike in controls, and made other improvements. Most notably, the new AP-MS characterization of protein binders of their activator combos adds to the impact of the paper, showing clearly how domain order does not majorly affect AP-MS interactors.

I expect many readers will be especially interested in MHV as a useful new activator that sometimes outperforms SAM and is rarely worse (while dropping the VP64 component). The authors did a good job of thoroughly demonstrating this across several contexts, while not over-promising on its performance. I think MHV's advantage is sufficiently consistent, and the need for better activators so pressing, that many readers will want to try this tool. By one measure, it seems MHV performs better than SAM at ~43% of targets and similarly at most other targets. MMH is almost always equivalent to SAM, but is notable because it is all human activators, unlike SAM which includes viral components. Further studies will be needed to assess how the set of recently published next generation activators (e.g. DREAM, NFZ, MHV) perform head-to-head and whether certain ones are better used in certain biological, protein fusion, or delivery contexts.

We thank the reviewer for their thoughtful evaluation of our revised manuscript. We appreciate the reviewer's recognition of our comprehensive additions, including domain order analysis, cross-cell toxicity analysis, expanded MHV activation data, and improved methodological clarity. The reviewer's emphasis on the significance of our AP-MS characterization, which provides molecular insights and confirms domain order's minimal impact on interactions, is particularly gratifying. We're pleased that the reviewer sees MHV as a valuable addition to the gene activation toolkit. MMH's all-human design offers unique benefits for immunogenicity-sensitive applications. We agree that head-to-head comparisons of activators like DREAM, NFZ, and MHV are crucial for the field and hope our work lays a foundation for such studies.

Major comment:

The authors responded convincingly to many of the questions about toxicity. The AP-MS Figure 6 is entirely new and raises some questions. Could they better show what are the top interactions for these activators (e.g. labeling dots in D)? It is unclear if they pulled down any co-activators (e.g. p300, Med). It is definitely of interest to the field to see which co-activators are recruited by which activator, including the new MHV. Perhaps co-activators could be included in C,D,E. If no co-activators are enriched, that would suggest these datasets are incomplete, and should be more caveated in the writing, especially given the lack of validation experiments.

We thank the reviewer for their insightful comments regarding our AP-MS data. We appreciate the opportunity to provide additional details on co-activator interactions and the interpretation of Figure 6.

To directly address the question of co-activator binding: we note that 74 proteins were identified as uniquely enriched with top-performing activators (MHV, MMH, and related constructs), and not with poorly performing ones, based on MSstats analyses comparing each activator

individually to the MCP-only control. This comparison – each activator versus MCP-empty – forms the basis of all binding enrichment analyses presented in the manuscript. It provides a stringent baseline for identifying interactions gained as a result of activation domain composition rather than general background binding to MCP or the scaffold. Among these 74 proteins, we identified numerous transcriptional co-activators, such as KDM6A, SSRP1, TAF15, HMGB3, and MACROD1.

These results are further visualized in the Venn diagram below.

Venn diagram of protein binding in high- and low-activation constructs. Each circle represents proteins bound (as determined by MSstats) in constructs with either high activation (orange) or low activation (green). Shared proteins (center) are found in both groups, while unique proteins are enriched only in one set.

With regard to p300, we note that this protein does not appear in the MCP-empty-based enrichment analysis used for most binding analyses in the manuscript, as it was not detected in the MCP-only condition. As a result, no \log_2FC could be calculated for p300 using MSstats. However, because p300 is robustly detected in both strong and weak activators, we were able to compare its abundance across those conditions. In comparisons between strong activators and weak activators, we observe clear enrichment of p300: for example, \log_2 fold changes of 1.9 for MHV and 2.0 for MMH, respectively, over MCP_BEL1_VP16x2 (weak activator). These data are included in Supplementary Data 6.

In addition to MSstats, we also performed an internal SAINT (Significance Analysis of INTERactome) analysis, which evaluates interaction specificity across all samples while modeling reproducibility and background. According to our SAINT results, all strong activators – and 1 to 2 weak ones – were confidently enriched for multiple transcriptional coactivators, including p300, CCAR1, and others.

With respect to the suggestion to label specific proteins in Figure 6D – we believe this might be difficult to achieve within the small panel, and might detract from its main purpose. This figure was designed to test whether adding a third domain to bipartite constructs results in inter-domain interference that reduces binding breadth. To do so, we compared bipartite and tripartite constructs that differ by only one domain (e.g., AD1–AD2 vs. AD1–AD2–AD3). As shown in Figure 6D, tripartite constructs consistently bind more proteins than their corresponding bipartite counterparts. Because this analysis is focused on quantifying total binding, rather than identifying specific interactions, labeling individual points would not clarify the result and could detract from the overall message.

Finally, we emphasize that our AP-MS experiments were not designed to assess individual activator–cofactor interactions, but rather to uncover broad, reproducible patterns of cofactor engagement associated with activation and toxicity. Our conclusions focus on the principle that cofactor binding alone is sufficient for toxicity, whereas activation requires that cofactors bind in precise 3D configurations – not on any single protein–protein interaction.

In E there are some proteins that are more enriched with the toxic activators – is that due to more binding or more expression of those proteins in the cells experiencing toxicity? Why does E not include any interactors that are more specific to the non-toxic activators? (e.g. wouldn't it change interpretations if more protein folding pathway interactors are actually higher in the non-toxic activators?) Is it valid to use ChatGPT to group interactors into functional categories (vaguely described in Methods), as opposed to supervised and reproducible gene ontology/gene set enrichment methods? Are those functional categories in E statistically enriched among toxic AD interactors vs non toxic AD interactors?

We appreciate the reviewer's comments regarding Figure 6e and the interpretation of protein enrichment in our AP-MS data. We cannot rule out the possibility that toxic activators trigger upregulation of certain endogenous proteins, which could in turn increase their detection in AP-MS. That said, our data shows that toxicity is highly additive across domains – more consistent with domain-intrinsic binding behavior than with secondary expression effects. Additionally, many of the proteins enriched in toxic constructs are not canonical stress-induced genes, further suggesting that the signal reflects specific interactions, not just broad transcriptional changes. Finally, even if increased expression of certain genes in response to particular activators contributes in part to the AP-MS signal, we believe these results remain biologically meaningful, as they identify cellular pathways consistently *associated* with toxic activators.

Our model, as presented in the manuscript, is based on the idea that toxicity arises from what activators bind, rather than what they fail to bind. This interpretation is supported by the finding that toxicity is additive, consistent with domain-intrinsic effects; and that enriched proteins are involved in cellular systems – such as the cytoskeleton and mitochondria – that are plausibly disrupted by misdirected recruitment. Based on this, we focused our analysis in Figure 6e on proteins enriched in toxic activators.

While proteins in Figure 6e are grouped into broad functional categories to aid interpretability, these categories were not derived from formal enrichment analysis.

Finally, the reviewer inquires about our use of ChatGPT to group proteins into functional categories. These groupings were manually reviewed and intended solely to provide visual orientation for readers. As recent benchmarking studies have shown (e.g., Hou & Ji, *Nature Methods* 2024), GPT-4 performs comparably to human experts in complex biological classification tasks, including cell type annotation from gene markers. Our use of GPT was limited to an arguably simpler task – mapping individual proteins to general functional categories. This approach, combined with manual curation, provided a reliable framework for organizing the data.

We have added additional detail to the Methods section to clarify how GPT-4 was used for functional grouping. The updated text is pasted below.

[LINE 1076]

“ChatGPT (GPT-4, June 2023 version) was used to assist in grouping proteins significantly enriched in toxic activators into broad functional categories (e.g., mitochondrial, cytoskeletal, chaperone). Protein names were provided as input, and functional characterizations were based on literature-derived annotations. These outputs were manually reviewed and curated.”

Minor comments, which are not necessary to resolve for publication in my view:

1. I valued the QC of lentiviral recombination with their triple AD combos, and would have liked to see that data added to the paper, as I think it would be useful to see for others working on related methods. Hopefully it will at least be included in an available peer review file.

We are happy to hear that the reviewer found the lentiviral recombination QC useful. In response, we've added this data as a Supplementary Note and included a brief cross-reference in the Methods section, pasted below.

[LINE 842]

“Assessment of lentiviral recombination

To evaluate the potential impact of lentiviral recombination on our screening results, we performed an experiment using two barcoded MCP activator constructs – one with an intact blasticidin resistance gene and one with a non-functional version. These constructs were mixed, packaged into lentivirus, and transduced into HEK293T cells at various MOIs. Barcode abundance was quantified after selection and passaging. Full experimental details and results are provided in Supplementary Note 3.”

2. It was helpful to see RNA-seq measurement of off-targets with appropriate controls and two target genes, but I wondered if more information could be retrieved from this data. The RNA-seq plots supp 33 do seem to show some off targets, and the writing mentions up to 5 genes that are significantly upregulated. What are these genes, and are they related to the sgRNA (e.g. have a mismatched sgRNA binding site near the TaSS) or not? Are they induced by expression

of MHV/SAM regardless of target gene? Of course, overall the tools look quite specific, so this is not a major problem, but it could be useful information.

The reviewer requests clarification on the identity and potential origin of the few genes that were upregulated in our RNA-seq off-target analysis. The maximum number of significantly upregulated genes observed in any activator condition was actually four, not five, and we have updated the manuscript accordingly.

[LINE 521]

“After filtering away genes with low baseline expression (<1 TPM), no activator induced significant upregulation (≥ 2 -fold change in expression) in more than **four** genes for either target, suggesting that, overall, these tools are highly specific.”

Specifically, when comparing MHV-expressing cells to MCP controls, we observed significant upregulation ($\geq \log_2(\text{fold change})$ of 2 and adjusted $p < 0.05$) of the following genes: TCEAL7, SLC16A2, KLF8, and HMOX1.

TCEAL7 and SLC16A2 were consistently upregulated or trended toward significance for all three activators (MHV, MMH, and SAM). KLF8 showed significant or near-significant upregulation for MHV and MMH. HMOX1 was uniquely upregulated in cells expressing MHV.

Importantly, these trends were observed across both the *TTN*- and *HBG1*-targeting conditions, suggesting that they do not stem from off-target effects of the guide RNAs but rather reflect the inherent activity of the activators themselves. We did not identify any clear guide RNA sequence homology near the TSS of these genes, supporting the interpretation that these are not guide-directed off-target events.

Despite these findings, the overall number of differentially expressed genes remained low, supporting the notion that our activators are highly specific.

3. The text does a better job than Fig 7 of accurately portraying how often MHV outperforms SAM. It may be helpful to add a panel to 7 that summarizes the many barplots that are currently left in the supplement, perhaps a heatmap or a count of the frequency with which each activator is best.

We thank the reviewer for the helpful suggestion. To better summarize how frequently MHV and MMH outperform SAM across targets and cell types, we have added a new panel (Figure 7g) to consolidate this information. This panel displays a categorical summary of significant differences relative to SAM, showing whether each activator performed significantly better, worse, or not significantly different across all tested targets. We believe this addition complements the detailed bar plots and provides a broader overview of comparative activator performance. The updated panel and legend are pasted below.

Figure 7. (g) Summary of activator performance across targets and cell types. Each circle represents the performance of MHV or MMH relative to SAM for a specific target gene in a given cell line, based on either protein or RNA expression. Green circles indicate targets where the activator is significantly stronger than SAM, orange circles indicate targets where the activator is significantly weaker than SAM, and gray circles indicate no significant difference.

4. Double check author names – is it Max Staller?

We thank the reviewer for pointing this out. The author's name was indeed incorrect – it has now been corrected to “Max Staller” in the revised manuscript.

5. The “floating e” issue is still there in Supp Fig 27. Also, the Off-targeting panel seems to show “idealized data” of what the RNA-seq could look like, but with stronger activation of the targeted gene than what was actually achieved in the real data in Figure 7 and supp 32-34. This scatter plot depiction could instead just be a clearly cartoonized schematic of RNA-seq.

We thank the reviewer for these observations. We have removed the “floating e” from Supplementary Figure 27 and updated the Off-targeting panel to more clearly reflect that it is a schematic representation of the RNA-seq workflow, rather than a depiction of actual data. The revised figure is pasted below.

Editorial Note: Supplementary Figure 27 in this Peer Review File is reproduced with permission from BioRender, Created in BioRender. Giddins, M. (2025) <https://BioRender.com/Igdoc71>

Supplementary Figure 27. Schematic depicting validation experiments (multiple targets, multiple cell lines, off-targeting) intended to evaluate the potency, generalizability, and specificity of top hits, MHV and MMH, against a current state-of-the-art MCP activator, SAM.

Reviewer #3 (Remarks on code availability):

n/a

Reviewer #4 (Remarks to the Author):

REVIEWER COMMENTS

Color coding for this document: Reviewer comments in **BLACK**, our responses in **BLUE**, words and edits in the manuscript are in **PURPLE**. Of note, some responses were refined for clarity with the aid of ChatGPT-4o.

We thank the reviewers for their thoughtful feedback and constructive suggestions. Before addressing specific comments, we note that we have updated the title of our manuscript to **“Combinatorial Protein Engineering Identifies Potent CRISPR Activators with Reduced Toxicity.”** This change reflects the reviewers’ emphasis on the significance of our findings regarding activator toxicity, and better captures the dual focus of our study on both tool potency and safety.

Reviewer #2 (Remarks to the Author):

The authors have responded to most of my concerns with revised figures, text, analysis or new data.

I remain concerned by the observation that mass-spec results show “a high correlation in binding partner enrichment, even among those activators composed of entirely different domains” (e.g. Figure 6C: VP16x2-BEL1 and VIRF2-ICP411-MLLx3 = 0.89). How do the authors explain this result given the data is normalized to an MCP control?

Is there some structural or domain similarity driving this for specific domains? Or perhaps it is more likely the explanation could come from the features of their MCP library construct design such as mislocalization or aggregation for toxic MCP fusions? The authors never examine the degree of P2A skipping in their Puro-P2A-NLS-MCP library design. It is known that P2A skipping can be context specific (PMID:). Additionally they do not examine the degree of nuclear localization for a representative toxic and non-toxic MCP activator fusion. It is known that NLS biology is also context specific. I realize the variable domains of each protein (the AD domains are on the C terminus but is it possible that much of this toxicity phenotype seen for the library could be due to fusion proteins that are Puro-P2A-NLS-MCP fusions and/or cytosolic mislocalized proteins? Or is it possible the toxic proteins are aggregating despite the inclusion of PF domains in some constructs? Western blotting to examine protein size could address P2A skipping efficiency. Immunofluorescence or cell fractionation followed by western blotting could address mislocalization concerns. This concern is re-enforced by my prior concern that it is quite surprising that the authors see so many mitochondrial, cytoskeletal and vesicular trafficking proteins associated with MCP fusions in their mass-spec results.

We thank the reviewer for raising important questions regarding the mass spectrometry data. We are grateful for the opportunity to clarify our interpretation of the high correlation in binding profiles among certain activators and to address the reviewer’s concerns about P2A skipping, nuclear localization, and the presence of mitochondrial and trafficking-related proteins within the AP/MS data.

Regarding the high correlation in binding partner enrichment among activators composed of entirely different domains (e.g., VP16x2-BEL1 and VIRF2-ICP411-MLLx3, $r = 0.89$; Figure 6c), we interpret this as a biologically meaningful result rather than an experimental artifact. The same strong correlation is not observed across all activators with different domain compositions, but specifically across those identified as toxic. That is, even when composed of entirely different activation domains, toxic constructs exhibit highly similar binding profiles. This convergence suggests that toxicity arises not from specific domain identity, but from shared effects on binding. We believe this supports a mechanistic link between toxicity and cofactor engagement, rather than reflecting nonspecific or stochastic interactions. These findings reinforce our conclusion that toxicity is a reproducible, biologically grounded phenotype, characterized by a distinct cofactor binding signature.

Regarding the concern about P2A skipping: P2A causes ribosomal stalling, leading to the release of the upstream polypeptide and reinitiation of translation (Sharma et al., *Nucleic Acids Res*, 2011). Since all of our constructs share the same upstream sequence (the PuroR resistance gene) – it is not clear to us how the downstream sequences, all of which begin with MCP and have not been translated at the point when the P2A peptide facilitates ribosomal stalling and release – would influence the rate of “P2A skipping”. For this reason, we expect all constructs to display a consistent rate of “unskipping” between puroR and the rest of the fusion, rather than variability driven by the identity of downstream domains. As such, we do not consider variable P2A skipping a likely explanation for the patterns observed in our AP/MS data.

We agree that NLS biology can be context-sensitive, and we cannot rule out the possibility that some activation domain fusions alter nuclear localization efficiency. If certain activation domains do modulate NLS efficiency, leading to mislocalization or cytosolic accumulation, that too would represent a mechanistic contributor to toxicity – not a confounder, and is something that future work, beyond the scope of this manuscript, can explore. The key result is that toxic constructs converge on a coherent cofactor binding signature, regardless of the specific route by which this signature arises.

We also appreciate the reviewer's observation that many enriched proteins in toxic constructs are mitochondrial, cytoskeletal, or vesicular trafficking components. Rather than interpreting these as technical noise or aggregation artifacts, we view them as markers of stress or dysregulation associated with the toxic state. These proteins are not observed in the non-toxic constructs, further supporting the idea that they are associated with toxicity-specific cellular responses.

Finally, we reiterate that our AP-MS experiments were not designed to probe the molecular basis of each individual interaction, but rather to identify reproducible patterns of cofactor engagement associated with toxicity and activation. We agree that follow-up experiments, such as immunofluorescence to monitor localization, could shed light on these underlying mechanisms. That said, to our knowledge, this work represents the most comprehensive and systematic effort to date to examine toxicity as an emergent property of CRISPR activators. We hope it will serve

as a starting point for further mechanistic investigations and elevate the importance of toxicity as a design consideration in future studies.

In response to the reviewer's comments, we have added the following to the discussion:

[LINE 580]

“These observations, paired with the distinct clustering of toxic activator interactions within our PCA analysis and absence of positive correlation between activation and toxicity, challenge the presumed link between these properties. **Future studies are needed to understand the basis of activator toxicity and how these effects might be reduced to improve tool safety.**”

Reviewer #3 (Remarks to the Author):

1. I appreciate the new summary figure and would only suggest changing the color scheme to be a bit more colorblind friendly (e.g. red, gray, blue).

We thank the reviewer for this helpful suggestion and agree that updating the color scheme will improve accessibility. In response, we have revised the summary figure to use a colorblind-friendly palette: blue, gray, and red indicate constructs that are significantly stronger, not significantly different, and significantly weaker than SAM, respectively. The updated panel is included below.

Figure 7. MHV and MMH outperform SAM in multiple contexts. (g) Summary of activator performance across targets and cell types. Each circle represents the performance of MHV or MMH relative to SAM for a specific target gene in a given cell line, based on either protein or RNA expression. **Blue circles indicate targets where the activator is significantly stronger than SAM, red circles indicate targets where the activator is significantly weaker than SAM, and gray circles indicate no significant difference.**

2. I note they chose not to revise the AP-MS section, but did provide more nuanced interpretations in the rebuttal to both reviews. I think more future work is needed to firmly establish what causes CRISPRa toxicity (including if toxicity requires an AD interaction with any of these proteins).

We thank the reviewer for their comment. We agree that further work is needed to pinpoint the precise mechanisms underlying CRISPRa toxicity, including whether direct interactions between activation domains and specific cofactors are required to elicit toxic effects. While we did not add new experimental data to the AP-MS section, we have incorporated this nuance into the revised discussion, pasted below.

[LINE 580]

“These observations, paired with the distinct clustering of toxic activator interactions within our PCA analysis and absence of positive correlation between activation and toxicity, challenge the presumed link between these properties. **Future studies are needed to understand the basis of activator toxicity and how these effects might be reduced to improve tool safety.**”

3. I appreciate their point about clarifying for the field where the baseline for activators is and their setting expectations for where to find future gains. I suggest this revised manuscript should be accepted for publication and commend the authors on this work.

We thank the reviewer for their thoughtful comments and positive assessment of our revised manuscript. We are grateful for their recognition of our efforts to contextualize current CRISPR activator performance and to help set realistic expectations for future development in the field.

Reviewer #4 (Remarks to the Author):

Reviewer #2 (Remarks to the Author):

The authors have responded to my comments on AP-MS results for toxic constructs that have highly correlated pull down profiles without common MCP fusion domains with text and a slight modification to the manuscript. The authors now state this toxicity mechanism issue is beyond the scope of the current manuscript in the text of the manuscript. I think it is reasonable for this manuscript to leave open mechanistic questions around toxicity but the results do remain somewhat confusing to me which is still a bit problematic as a big message of the paper is around avoiding toxicity.

With respect to P2A/T2A biology. There is quite a bit known about this for example:

<https://www.nature.com/articles/s41598-017-02460-2>

Our response:

We thank the reviewer for their comment and agree that understanding the mechanisms driving toxicity remains an important area for future work. We also appreciate the reference provided, which describes context-specific effects when multiple 2As are used in tandem. In our design, the single P2A lies between puroR and MCP. While different “unskipped” puroR-MCP-AD fusions may vary in stability or behavior, the ADs themselves are not directly downstream of the P2A (and are not yet translated at the point of skipping). As such, we believe they are unlikely to influence skipping efficiency. That said, we agree that potential P2A effects are important considerations and could be studied in depth in future work.